# Mithramycin alters EWS::FLI1 DNA binding and RNA polymerase II processivity to inhibit nascent transcription

Rebecca Kaufman [1,5], Guillermo Flores[1,2,3,5], Elissa A. Boguslawski [1,2,4,5], Seneca Kinn-Gurzo[1], Maggie Chassé[1,2], Ian Beddows [2], Marie Adams [2], Matthew C. Stout[1], Lauren Gaetano[1], Raphael Lopez[1], Sridhar Veluvolu[1], Andrew Fuller [4], Susan M. Kitchen-Goosen[2], Zachary P. Tolstyka[4], Jenna M. Gedminas[1,2] & Patrick J. Grohar [1,2,4] ✉

Although many DNA binding natural products exert their effects through non-specific mechanisms, a therapeutic opportunity exists for a subset of these compounds that alter the expression or activity of specific driver oncogenes in specific cell contexts. In this study, we integrate CUT&Tag with Global Run-On Sequencing (CUT, Tag, and GRO) to show that the minor groove binding compound, mithramycin (MMA), inhibits the Ewing sarcoma oncogenic driver, the EWS::FLI1 transcription factor. MMA causes either an increase or decrease in EWS::FLI1 binding to chromatin at downstream target response elements to poison nascent transcription. The reversal of EWS::FLI1 activity is limited by non-specific effects of the drug on RNAPII processivity but can be optimized by continuous administration at low concentration to cause more precise reversal of the oncogenic transcriptome and striking Ewing sarcoma xenograft regressions. The activity in vivo is further improved with a less-toxic second-generation analog, AIT-102.

DNA binding natural products are often considered non-specific "chemotherapy" due to broad mechanisms of action that generate DNA damage and/or target essential cell processes such as cell division. However, many natural products demonstrate heightened potency in subsets of tumors. Many of these compounds, such as mithramycin (MMA), trabectedin, dactinomycin, and irinotecan, alter the expression or activity of specific oncogenes in specific cellular contexts[1–3]. The contribution of these context-specific effects to the overall cytotoxicity profile, relative to non-specific mechanisms, is not well understood for many compounds. Modern genomics provides an unprecedented opportunity to understand and optimize the context-specific effects of broadly active compounds for specific tumors to improve activity.

The ETS (E-twenty-six specific sequence) family of transcription factors (TFs) is frequently mutated or overexpressed in a range of tumors including prostate cancer, breast carcinoma, GBM, and sarcoma[4]. These proteins play critical roles in these tumors with the best example being the clear dependence of Ewing sarcoma on the FET-ETS fusion protein, EWS::FLI1[5–10]. The challenge targeting these oncogenic transcription factors is modulating activity with specificity. All ETS factors bind DNA at a core GGA(A/T) binding motif. However, every ETS factor drives a unique transcriptome and as many as 15–20 different ETS factors are co-expressed at the same time in a cell[11–13]. To achieve specificity, these TFs rely on networks of interacting proteins including androgen receptor, PARP, TP53, AP-1, and DNA dependent protein kinases as well as context specific gene expression changes in upstream and downstream modulators[12,13]. Nevertheless, the common molecular process for activity is binding to the core GGA(A/T) DNA response element. Therefore, compounds such as MMA that bind GG/

[1]Children's Hospital of Philadelphia, Philadelphia, PA, USA. [2]Van Andel Research Institute, Grand Rapids, MI, USA. [3]Michigan State University, College of Human Medicine, Grand Rapids, MI, USA. [4]University of Michigan Division of Pediatric Hematology/Oncology, Ann Arbor, MI, USA. [5]These authors contributed equally: Rebecca Kaufman, Guillermo Flores, Elissa A. Boguslawski. ✉e-mail: grohar@med.umich.edu

GC rich DNA at the same sequence have the potential to find broad utility modulating ETS factor activity in specific tumors to impact patient outcomes.

We identified MMA as an inhibitor of the EWS::FLI1 transcription factor in an unbiased High Throughput Screen (HTS) of over 50,000 compounds[14]. We showed selective suppression of a highly specific downstream target NR0B1 luciferase reporter construct, reversal of expression of well-established EWS::FLI1 downstream targets, and of the EWS::FLI1 transcriptome. We also demonstrated comparable reversal of EWS::FLI1 with a second-generation MMA analog with an improved toxicity profile, AIT-102 (formerly EC8042), as well as a combination therapy with CDK9 inhibitors[15,16]. However, the mechanism of EWS::FLI1 blockade has remained elusive. MMA is known to bind the minor groove of DNA at GC/GG rich sequences that are similar to the GGAA binding domain of ETS factors[17–19]. Competitive inhibition by MMA has been described for proteins such as SP1 that bind GC sequences as well as proteins and complexes that bind the minor groove such as MBP1, TBP, and the SMARCB1-deficient mutated SWI/SNF chromatin remodeling complex (an effect also described with a related aureolic acid in vitro)[17,20–23]. X-ray crystallography has provided structural data to support the ability of MMA to either stabilize or destabilize binding of the FLI1 binding domain to DNA due to alterations in DNA conformation or via direct interaction between the position-3 side chain of MMA and three different residues of the EWS::FLI1 binding partner, RUNX2[24–28]. These observations were supported by fluorescent anisotropy data both with the RUNX2 complex and a hexameric GGAA repeat sequence intended to model the GGAA microsatellite EWS::FLI1 enhancer[26,29]. Nevertheless, the impact of MMA on EWS::FLI1 DNA binding and activation of RNA Polymerase II (RNAPII), genome-wide, in Ewing sarcoma cells, at physiologically relevant concentrations, has not been described.

A published report from the 1960s demonstrated a complete response of an Ewing sarcoma patient to MMA when given as a seven-day continuous infusion (long before EWS::FLI1 was discovered)[30]. Importantly, when we re-introduced the drug to Ewing sarcoma patients in 2012, we administered the drug as an intermittent bolus dose, and only one patient demonstrated brief and transient disease stabilization[31]. Therefore, the goal of the current study is to revisit the mechanism of EWS::FLI1 suppression by MMA through the lens of modern genomic technologies and a novel integration of CUT&Tag with GROseq. The approach examines how the drug alters EWS::FLI1 binding and activity to determine if the difference in schedule explains the difference in activity in patients in the 1960s relative to our more recent study. This understanding will provide the opportunity to optimize the schedule of administration of the drug and facilitate the clinical development of a less-toxic second-generation MMA analog, AIT-102.

## Results

### Modeling the impact of mithramycin schedule on EWS::FLI1 activity

To determine how MMA exposure (exposure = drug concentration x time) impacts EWS::FLI1 activity, we identified two different drug exposures that caused comparable suppression of EWS::FLI1 but reflected intermittent bolus dosing relative to continuous infusion. The goal was not to recapitulate the exact exposures of the drug achievable in patients which is influenced by many factors. Instead, we wanted to model the impact of MMA on EWS::FLI1 binding and activity at higher concentrations intermittently relative to continuous exposure at lower concentrations to determine if molecular mechanisms favor a particular schedule.

MMA suppressed expression of the EWS::FLI1 induced targets NR0B1 and EZH2 but not RPB1 or GAPDH in TC32 and TC252 cells at concentrations between 25 and 100 nmol/L for 18 h (Fig. 1a, b). To identify similar suppression at lower concentrations of MMA, TC32

and TC252 lysates were collected at intervals during a continuous 20 nmol/L exposure. Similar suppression of NR0B1 and EZH2 expression occurred between 54 and 72 h in TC32 cells or 48 h in TC252 cells (Fig. 1c, d). Therefore, we selected relatively similar exposures in TC32 cells of 18 h at 100 nmol/L (High Concentration Intermittent Exposure = 1800 nmol/L*hour; HCE) and 72 h at 20 nmol/L (Low Concentration Continuous Exposure = 1440 nmol/L*hour; LCE) for further study (Fig. 1e). To confirm similar suppression using a quantitative assay, we used qPCR to determine the impact of LCE vs. HCE on expression of NR0B1, a well-established EWS::FLI1 induced target, and PHLDA1, a well-established EWS::FLI1 repressed target. NR0B1 is a downstream target with an EWS::FLI1 GGAA microsatellite response element in the promoter of the gene. CRISPR-Cas9 deletion of this microsatellite eliminates EWS::FLI1 responsiveness[32]. Further, while both FLI1 and EWS::FLI1 can bind this element, only EWS::FLI1 can transactivate at the locus[33–35]. PHLDA1 also has a well-established link between EWS::FLI1 binding and suppression of expression[36]. LCE and HCE showed similar suppression of NR0B1 (LCE, $Log_2FC = 0.14 +/− 0.072$ SD vs HCE, $Log_2FC = 0.1 +/− 0.03$ SD; 1 way ANOVA $P < 0.0001$) and induction of PHLDA1 expression (LCE, $Log_2FC = 6.49 +/− 1.4$ SD; vs HCE, $Log_2FC = 8.6 +/− 2.0$ SD; 1 way ANOVA $P < 0.0001$) (Fig. 1f and g). We chose to match suppression instead of exact drug exposure, therefore the HCE does have a slightly higher exposure than LCE.

### Mithramycin causes a global increase in transcription and impairs RNAPII processivity at higher concentrations

To understand how these drug exposures impact nascent transcription, we performed Global Run-On sequencing (GROseq)[37]. Following incubation with MMA, RNAPII was paused and then released in the presence of brominated uridine (BrdU), immunoprecipitated using an anti-BrdU antibody, and aligned to the genome to determine the impact of LCE vs. HCE on nascent transcription genome-wide[37]. Critically, this approach differs from the steady state evaluation of EWS::FLI1 downstream mRNA expression levels provided by RNA sequencing. MMA has often been referred to as a "general transcription inhibitor" due to old in vitro data at supraphysiologic concentrations (800 nmol/L) showing inhibition of *MYC* gene transcriptional elongation[31,38,39]. Strikingly, we found exactly the opposite and a fair amount of non-specific transcriptional activation, particularly with HCE to MMA (Fig. 2a).

To determine if there is an impact of MMA on transcription elongation that may limit productive transcription as previously reported for MYC, we plotted the $Log_2FC$ ratio of counts (treated to solvent) of all protein coding genes from transcriptional start site (TSS) to transcription end site (TES) +/− 3 kb (Fig. 2b). We observed a diagonal line from the TSS to TES indicative of an impairment in the ability of RNAPII to transcribe across the gene body that was more prominent with HCE and more evident on the profile plot (Fig. 2b).

Next, we reasoned that if there was indeed an impact on transcription elongation, it should be more prominent at larger genes. Therefore, we partitioned the transcriptome into tertiles representing small (< 13,932 base pairs), medium (13932 to 50308 base pairs) or large genes (> 50,308 base pairs). We found more prominent impact on RNAPII processivity at the large genes with higher concentrations of MMA; an effect most visible in the corresponding profile plots (Fig. 2c, d). LCE to MMA had minimal effects on RNAPII processivity (Fig. 2d). These effects were not confounded by apoptosis as there was limited apoptosis above background at these drug exposures (Fig. 2e). Importantly, the maximum concentration (Cmax) of mithramycin in patients was approximately 14−18 nmol/L suggesting that these non-specific effects on transcription that occur between 20 nmol/L and 100 nmol/L may be the dose limiting molecular mechanism(s) that causes toxicity[31]. Nevertheless, these data demonstrate that at concentrations that are achievable in patients, MMA is not a general transcription inhibitor.

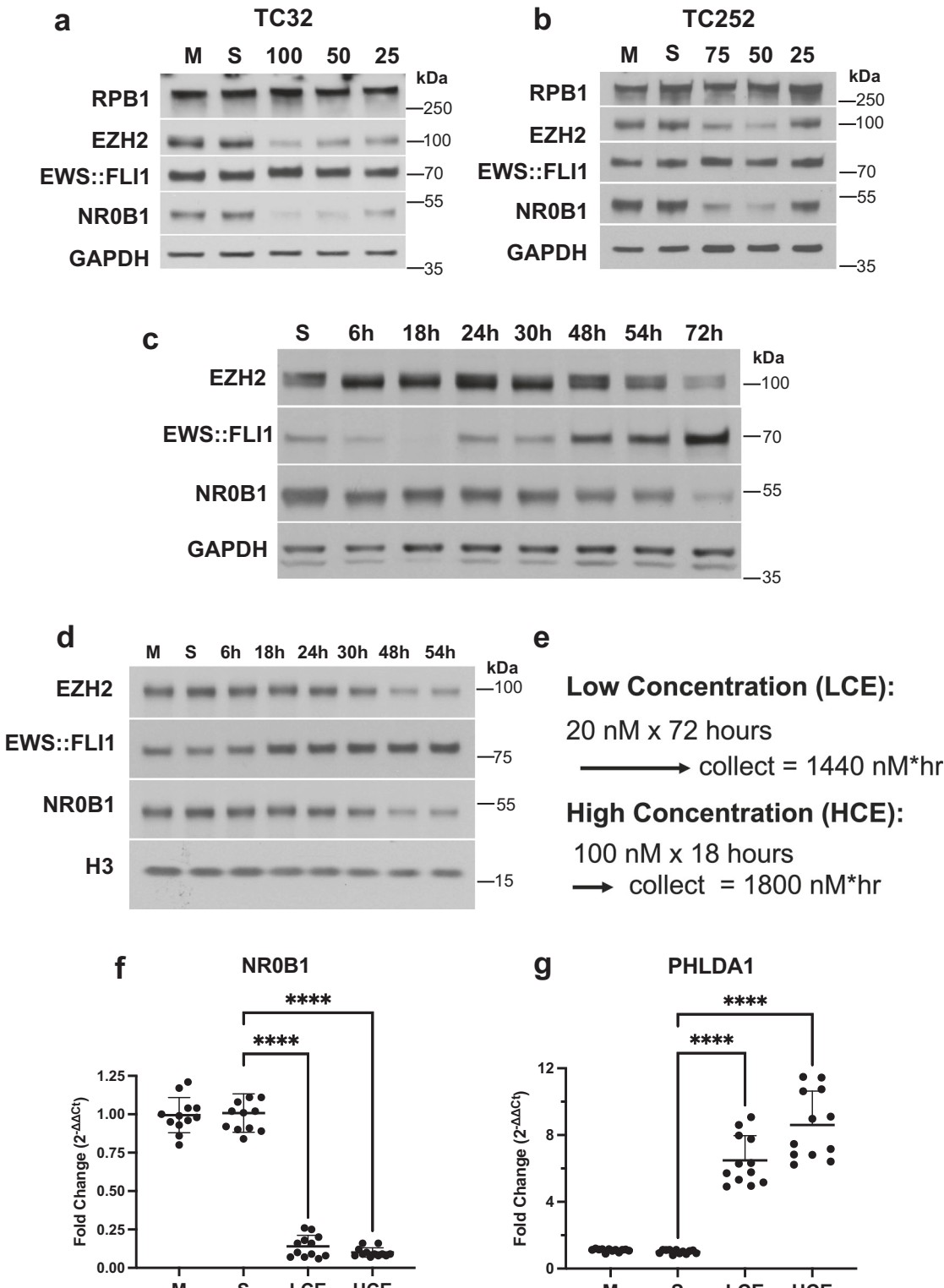

**Fig. 1 | Modeling continuous mithramycin exposure relative to intermittent bolus dosing.** Western blot demonstrating reduction in expression of the EWS::-FLI1 downstream targets, NR0B1 and EZH2, but not EWS::FLI1, RPB1 or GAPDH following exposure of **a** TC32 or **b** TC252 cells to 25, 50, 75, or 100 nmol/L MMA for 18 hours relative to medium (M) or solvent (S). Similar suppression of NR0B1 and EZH2 is obtained with 20 nmol/L MMA relative to solvent (S) in **c** TC32 cells at 72 h and **d** TC252 cells at 48 h. **e** Drug exposures (exposure = concentration x time); High Concentration (100 nM for 18 hours = (HCE); 20 nM for 72 h = Low Concentration

(LCE) selected *for* in vitro modeling in TC32 cells. Quantitative comparison of the impact of drug treatment on expression of the EWS::FLI1 induced target **f** NR0B1 or repressed target **g** PHLDA1. Data presented as mean +/− SD fold change ($2^{-\Delta\Delta CT}$) relative to GAPDH ($n = 12$). Significance determined by one-way ANOVA followed by Dunnett's multiple comparison test compared to solvent (**** $P < 0.0001$) of 3 independent experiments each with 2 biological and 2 technical replicates (all data points are shown). See source data for full statistics.

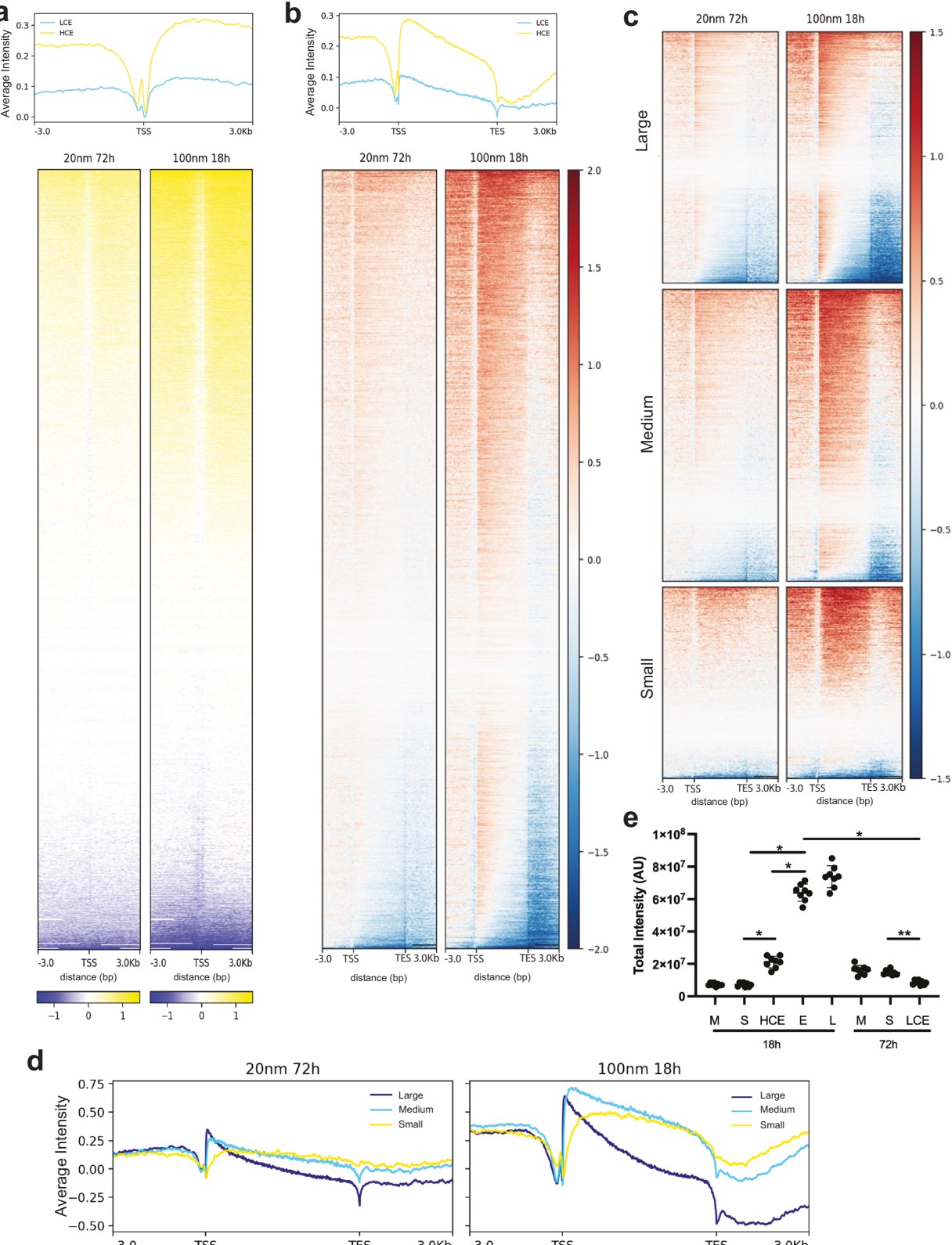

## Mithramycin impairs RNAPII processivity at higher concentrations to limit EWS::FLI1 blockade

To determine how this effect on RNAPII elongation influences EWS::FLI1 driven transcription, we used an established approach and highly specific siRNA targeting the breakpoint of EWS::FLI1[8,40] to silence the fusion protein in TC32 cells and generate a list of EWS::FLI1 induced and repressed targets with a Log$_2$FC +/−1 and P < 0.01. We then plotted

the ratio of counts (treated to control) of these EWS::FLI1 induced (induced by the fusion protein and repressed with silencing) vs. repressed targets (repressed by the fusion protein and induced with silencing) from TSS to TES +/− 3 kb (Fig. 3a).

Consistent with our prior publications[14,15], we found clear suppression of EWS::FLI1 induced targets that was highly significant at the TES with both LCE (P = 1.1e-48) and HCE (P = 1.5e-33)(Fig. 3b). The

**Fig. 2 | Mithramycin induces transcription but impairs RNAPII processivity.**
**a** Profile plot of average GROseq signal intensity centered on the Transcription
Start Site (TSS; top) and heat map of Log$_2$FC of counts (treated to solvent) of
GROseq signal centered on TSS genome-wide demonstrating few suppressed
transcripts (blue) and limited induction of transcription (yellow) favoring HCE (HCE
= yellow line; LCE = blue line). **b** Profile plot of average GROseq signal intensity and
heat map of Log$_2$FC counts (treated to solvent; bottom) aligned genome-wide from
TSS to Transcription End Site (TES) showing impaired elongation as indicated by
the downward slope in the profile plot and the change from increased counts (red)
at the TSS relative to no change (white) or decreased (blue) at the TES favoring HCE.
**c** More prominent impairment in RNAPII processivity evident in large ( > 50308

base pairs) genes relative to medium (13932 to 50308 base pairs) or small ( < 13,932
base pairs) genes as demonstrated by Log$_2$FC of GROseq signal aligned genome-
wide from TSS to TES favoring HCE. **d** Profile plot of average GROseq signal
intensity separated by gene size (large (blue), medium (light blue), small (yellow))
demonstrating impaired elongation favoring large genes and HCE. **e** Cleaved cas-
pase 3,7 assay measuring apoptosis after HCE or LCE relative to medium (M),
solvent (S), or the positive controls (E) 5 micromol/L etoposide for 18 h or (L) 25
nmol/L for 18 h of lurbinectedin. Each point is a biological replicate (line indicates
mean +/− SD, $n$ = 8 per condition; 1 way ANOVA ($P$ < 0.0001) with Tukey's multiple
comparisons tests *$P$ < 0.0001; **$P$ < 0.0175); overall data is representative of 3
independent experiments. See source data for full descriptive statistics.

suppression of EWS::FLI1 with LCE was due to altered transcription
initiation leading to uniform suppression of EWS::FLI1 induced targets
and induction of repressed targets (Fig. 3a). Uniform suppression was
visible in the profile plot and quantitated using violin plots of relative
counts in the TSS, gene body, and TES (Fig. 3c; Supplementary Figs. S1,
S2). In contrast, the suppression of EWS::FLI1 with HCE resulted from a
combination of effects on initiation and impaired transcriptional
elongation (Fig. 3a, d). The generalized non-specific induction of
transcription described above (Fig. 2a) was evident with HCE but was
rescued by effects of the drug that restricted transcriptional elonga-
tion to aid in the suppression of EWS::FLI1 induced targets to cause
statistically significant suppression at the TES (Figs. Supplementary
Figs. S1, 3b, 3d). However, the impairment of RNAPII elongation limited
the induction of EWS::FLI1 suppressed targets. Statistically significant
induction of these repressed targets was only seen with LCE and not
HCE at the TES (Fig. 3e, Supplementary Fig. S2).

These differences were more prominent as a function of gene size.
At large EWS::FLI1 target genes, HCE demonstrated clear interference
with RNAPII processivity to suppress expression of induced targets but
was unable to induce large EWS::FLI1 repressed targets (Fig. 3f, g).
Additionally, medium and small EWS::FLI1 induced targets did not have
enough genomic distance to allow the effect of RNAPII processivity to
repress expression with HCE and they were further induced with this
exposure (Fig. 3f, g). In contrast, LCE reversed expression of both
induced and repressed targets uniformly throughout the gene body
independent of gene size consistent with alterations in transcription
initiation as the driving mechanism (Fig. 3g). PRKCB[41] is a well-
established EWS::FLI1 target and an example locus that clearly shows
these effects: suppression of transcriptional initiation with LCE but a
reliance on elongation to suppress expression with HCE (Supplemen-
tary Fig. S3). Together these data point to mechanisms that show that
MMA LCE is more specific and effective at suppressing EWS::FLI1-dri-
ven nascent transcription than HCE.

## Mithramycin alters EWS::FLI1 binding to chromatin
Because MMA influenced transcriptional initiation of EWS::FLI1 target
genes, we next examined how these different exposures impacted
EWS::FLI1 binding to chromatin. Chromatin fractionation demon-
strated similar global binding of EWS::FLI1 to chromatin with LCE and
HCE. However, there was a slight increase in chromatin occupancy of
EWS::FLI1 with LCE (Fig. 4a). This increase likely reflected the combi-
nation of increased expression (see Fig. 1c) and stabilization of
EWS::FLI1 binding at the GGAA response element. Stabilization of
binding by MMA has previously been demonstrated in vitro for the FLI1
DNA binding domain at a GGAA hexameric repeat oligonucleotide
using fluorescent anisotropy[26,29]. As controls for the fractionation, we
probed for GAPDH for the soluble fractions and for three different
SWI/SNF chromatin remodeling complex subunits that are known to
be chromatin bound[42], DPF2, BRD9, PBRM1 (Fig. 4a).

To gain a deeper understanding of the impact of drug exposure
on EWS::FLI1 binding and activity at downstream target genes, we
integrated our GROseq data with highly precise mapping of EWS::FLI1

binding to chromatin generated using Cleavage Under Targets &
Tagmentation (CUT&Tag)[43] to develop the novel CUT, Tag & GRO
assay. The impact of drug exposure on EWS::FLI1 binding was eval-
uated in a quantitative manner per cell by standardizing the cell
number input, isolating nuclei, and then again standardizing the input
of 100,000 nuclei to perform CUT&Tag. We used a FLI1 antibody to
bind to EWS::FLI1 since FLI1 is not expressed in this cell type (as is
commonly done in the field)[44,45]. We then used an anti-rabbit second-
ary antibody to link to the pA-Tn5 to tagment the DNA.

MMA both increased and decreased EWS::FLI1 binding to chro-
matin (Fig. 4b). Increased binding favored LCE relative to HCE, con-
sistent with the chromatin fractionation data, and included highly
significant changes in occupancy at multiple well-established GGAA
microsatellite driven EWS::FLI1 target genes including: *NROB1, FEZF1,
FOCAD, IL1RAP, RCOR1, GSTM4, CACNB2, FCGRT, EZH2, and AKAP7*
(Fig. 4b)[14,33,46–48]. Importantly, the integration of GROseq with CUT&-
Tag proved to be a powerful way to demonstrate the impact of the
drug on EWS::FLI1 binding and activity at every locus in the genome.
Both increased and decreased EWS::FLI1 binding reversed activity
of the fusion protein as shown at the well-established GGAA
microsatellite driven target genes, *NROB1* and *RCOR1* (Fig. 4c–f). At
the *NROB1* locus, loss of binding occurred at peak 3 just upstream of
the GGAA microsatellite with both LCE and HCE in the promoter (on
the (−) strand)(Fig. 4c, Supplementary Fig. S4a). This loss of binding
was demonstrated quantitatively using fragment counts on the
IGV and associated with a loss of nascent transcription at this locus
(Fig. 4d, e, Supplementary Fig. S4b). This peak showed a reduction in
fragment count with both [LCE (46200 (SEM +/− 7396) to 15586
(SEM +/− 1016); $P$ = 0.01) and HCE (5100 (SEM +/− 597) to 1034
(SEM +/− 597.3); $P$ = 0.01)](Fig. 4d, Supplementary Fig. S4b). Impor-
tantly, the replicates were extremely consistent thus validating the
nuclei standardization approach; a result validated with a correlation
matrix (Fig. 4c, Supplementary Fig. S5).

In contrast, the *RCOR1* locus showed a gain in binding and loss of
transcription with fragment counts increasing with both LCE [(Peak 1
22,967 (SEM +/−10170) to 80,278 (SEM +/− 10170); $P$ = 0.006), (Peak 2
(168221 SEM +/− 38713) to 313200 (SEM +/− 17481); $P$ = 0.02); HCE
(Peak 1 (2314 (SEM +/− 765) to 6751 (SEM +/− 2195); $P$ = 0.052), Peak2
(6522 (SEM +/− 1602) to 12687 (SEM +/− 620); $P$ = 0.007)] (Fig. 4d and
Supplementary Fig. S6). This increased binding was significant at
multiple other well established GGAA microsatellites: *FCGRT, IL1RAP,
EZH2, CACNB2, GSTM4, FEZF1, FOCAD, AKAP7* (Fig. 4b, Supplementary
Fig. S7)[33,46–48]. This is a clear demonstration using CUT, Tag, and GRO of
the proposed "Goldilocks hypothesis"[49] that just the right transcription
factor dosage of EWS::FLI1 is required at response elements for pro-
ductive transcription.

## Mithramycin alters EWS::FLI1 binding at GGAA microsatellites and isolated GGAA response elements to inhibit the fusion protein
Having demonstrated altered EWS::FLI1 binding at well described
GGAA microsatellite loci and targets, we next expanded our analysis

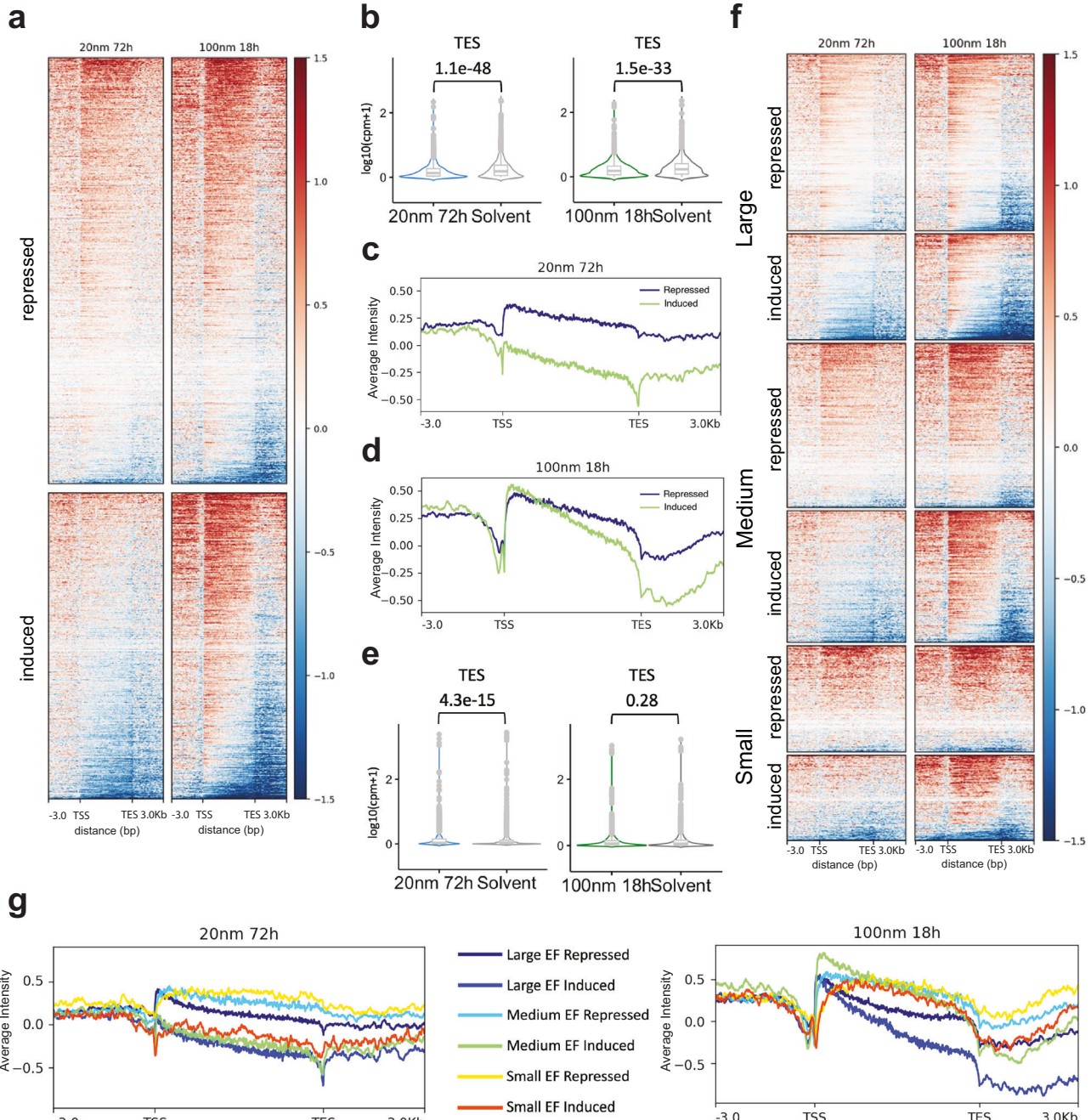

**Fig. 3 | Low-dose continuous mithramycin inhibits EWS::FLI1 with high specificity. a** Heat map of Log$_2$FC of counts (treated to control) of GROseq signal aligned from the TSS to TES (kb = kilobases) of EWS::FLI1 downstream target genes demonstrating MMA induction of repressed targets (top) and repression of induced targets (bottom). The impairment of RNAPII processivity with HCE aids in the repression of induced targets but limits repressed target expression. **b** MMA represses expression of EWS::FLI1 induced targets with both LCE (*P* = 1.1e-48) and HCE (*P* = 1.5e-33). Violin plot of normalized counts (Log$_{10}$(CPM + 1)) for EWS::FLI1 induced targets at the TES. A two-sided Wilcoxon rank-sum test comparing biological replicates of solvent control (*n* = 3) to LCE (*n* = 2)(*P* = 1.1e-48) or different solvent controls (*n* = 3) to HCE (*n* = 3)(*P* = 1.5e-33). Center lines show medians, box limits indicate 25$^{th}$ to 75$^{th}$ percentile (IQR), and whiskers extend 1.5 times the IQR. **c** LCE MMA alters transcription initiation to uniformly repress EWS::FLI1 induced targets (induced, green) or induce repressed targets (repressed, blue) as demonstrated in a profile plot of average Log$_2$FC (treated to control) in GROseq signal intensity aligned to targets from TSS to TES. **d** HCE MMA impairs RNAPII processivity to limit the induction of EWS::FLI1 repressed targets (repressed, blue) but

aids in the repression of EWS::FLI1 induced targets (induced, green). Data is a profile plot of average Log$_2$FC (treated to control) GROseq signal intensity aligned to targets from TSS to TES. **e** Violin plot of normalized counts (Log$_{10}$(CPM + 1)) at the TES showing MMA induction of EWS::FLI1 repressed targets with LCE but not HCE. A two-sided Wilcoxon rank-sum test comparing biological replicates of solvent control (*n* = 3) to LCE (*n* = 2)(*P* = 4.3e-15) or different solvent controls (*n* = 3) to HCE (*n* = 3)(*P* = 0.28). Center lines show medians, box limits indicate IQR, and whiskers extend 1.5 times the IQR. **f** Heat map of Log$_2$FC of counts (treated to control) of GROseq signal and **g** corresponding profile plots of average Log$_2$FC (treated to control) GROseq signal intensity aligned from the TSS to TES of EWS::FLI1 repressed or induced targets partitioned into large, medium, or small target genes. LCE uniformly induces repressed targets and uniformly represses induced targets (heat map and left profile plot) independent of gene size. In contrast, HCE is unable to induce large EWS::FLI1 repressed targets (heat map; right profile purple tracing) or repress medium or small induced targets (heat maps; right profile green and red tracings).

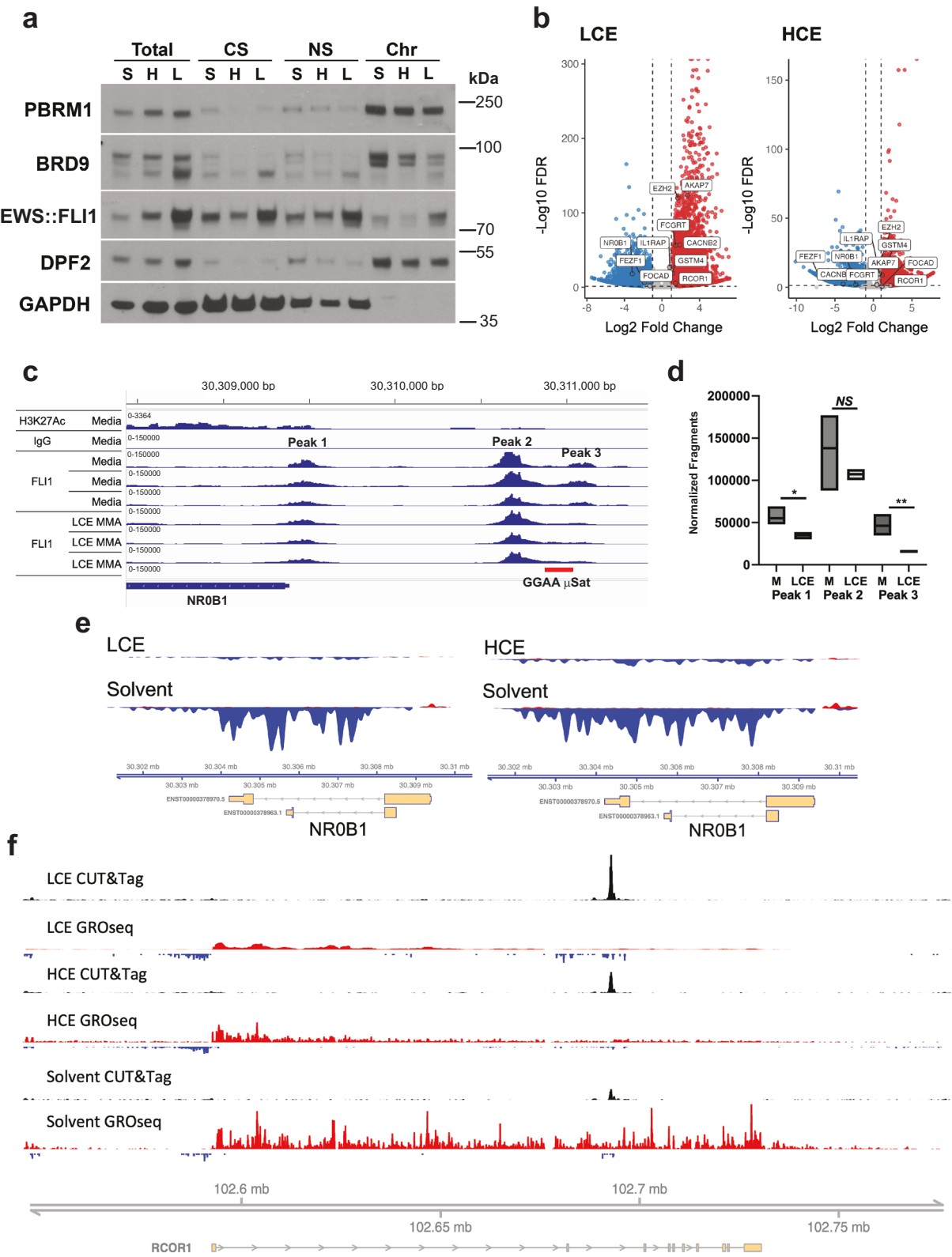

genome-wide to examine the impact of MMA exposure on EWS::FLI1 binding and activity. We took all differentially bound peaks and performed Homer motif analysis. We found a striking, highly significant, enrichment of ETS factor GGAA motifs genome-wide following MMA exposure with both LCE and HCE. The top three differentially bound and 17 of the top 20 highly significant known motifs were GGAA containing ETS motifs for LCE, FLI1 (*P* = 1e-3556), ETV2 (*P* = 1e-3319), and EWS::FLI (*P* = 1e-3244)(Fig. 5a, Supplementary Fig. S8a). Further,

unbiased de novo motif discovery identified the ETS (Elk1) TTCC/GGAA (*P* = 1e-3722) and the GGAA microsatellite (*P* = 1e-2317) as the top two most significant enriched motifs (Fig. 5b, Supplementary Fig. S8b). Importantly, the ELK1 motif was the complement of the known "high affinity" EWS::FLI1 target motif, ACCGGAAG/aT/c[50].

Similar results were observed with HCE, where the same top motifs were identified in the known motif analysis (FLI1 (*P* = 1e-903), EWS::FLI (*P* = 1e-879), ETV2 (*P* = 1e-864)) and an alternative ETS

**Fig. 4 | Mithramycin alters EWS::FLI1 occupancy on chromatin. a** Chromatin fractionation demonstrating increased occupancy of EWS::FLI1 with LCE (L) but reduced occupancy with HCE (H) relative to solvent control (S) in the chromatin bound fraction (Chr) relative to the total lysate (Total), cytoplasmic soluble (CS), nuclear soluble (NS) fractions. SWI/SNF subunits (DPF2, PBRM1, BRD9) and GAPDH are chromatin and soluble fraction controls. Data reflects three independent experiments. **b** MMA alters EWS::FLI1 binding to chromatin as demonstrated by CUT&Tag favoring increased binding with LCE. Volcano plot showing statistical significance (-Log$_{10}$FDR (False Discovery Rate)) as a function of Log$_2$FC change in binding (treated to control; red (increased), blue (decreased)); callouts are well-established microsatellite driven EWS::FLI1 downstream targets. **c** Reduction of EWS::FLI1 binding at peak 3 of the *NR0B1* locus as measured by CUT&Tag using a FLI1 antibody compared to H3K27ac positive and IgG negative control following exposure to LCE. **d** Quantitation of binding on the IGV. Box plots show minimum to maximum value and the mean for 3 independent biological replicates for each condition (two tailed *T*-test: *$P = 0.058$; **$P = 0.01$; *NS* = Not Significant)(See source data for full statistics). **e** Reduction in EWS::FLI1 driven NR0B1 nascent transcription as demonstrated by GROseq with either LCE or HCE. **f** Enrichment of EWS::FLI1 at the *RCOR1* locus leads to a loss of nascent transcription with LCE and HCE as demonstrated by CUT, Tag, and GRO (see text for fragment quantitation and statistics). GROseq tracks are group scaled to the highest peak in control; bp = base pairs; mb = megabase pairs.

(ETV1,TTCC/GGAA ($P = 1e\text{-}3722$)) and the GGAA microsatellite ($P = 1e\text{-}624$) in the de novo analysis (Supplementary Fig. S9a, S9b). Further, the top motifs with either MMA exposure or either analysis (all other top 30 known motifs and number three in the de novo analysis) were AP-1 motifs which are known to be associated with EWS::FLI1[51–53]. The identified motifs were more statistically significant in all analyses with LCE than with HCE. Additionally, a motif analysis of LCE vs HCE identified the GGAA microsatellite as the most significant differentially enriched motif with LCE. These data suggest a more specific effect on EWS::FLI1 binding with this lower concentration continuous MMA (LCE)(Supplementary Fig. S9c).

To understand how MMA influences EWS::FLI1 binding and activity, we analyzed groups of genes associated with binding at specific response elements using CUT, Tag and GRO. We first analyzed EWS::FLI1 binding and activity at GGAA microsatellites. We identified 20,765 GGAA microsatellites in the genome. These GGAA motifs were annotated to genes using Homer and then filtered and viewed separately for EWS::FLI1 induced targets (Log$_2$FC < 1, $P < 0.01$) or EWS::FLI1 repressed targets (Log$_2$FC > 1, $P < 0.01$) identified using siRNA silencing of EWS::FLI1 and RNAseq. Impressively, we were able to clearly show enrichment in binding of EWS::FLI1 at GGAA microsatellites for both LCE and HCE. Aggregate signal profiles centered on GGAA repeat loci (+/− 1.5 kb) revealed markedly enriched EWS::FLI1 binding in LCE and HCE compared to media for both gene sets, but with notable differences in magnitude (Fig. 5c, Supplementary Fig. S10a (top see scales)). Induced genes exhibited higher signal intensity than repressed genes suggesting preferential activity at regulatory repeats associated with EWS::FLI1 induced targets favoring LCE (Fig. 5c, Supplementary Fig. S10a). This resulted in loss or gain of transcription initiation for these EWS::FLI1 induced or repressed targets, with LCE, as shown in the profile plots (Fig. 5d, Supplementary Fig. S10b). In contrast, there was increased transcription initiation with HCE and the limitation of RNAPII processivity described above aided the suppression of EWS::FLI1 induced targets but limited productive transcription of EWS::FLI1 repressed targets (Fig. 5d, Supplementary Fig. S10b). At isolated single GGAA response elements, MMA acted analogously as at microsatellites but with substantially smaller magnitude to enhance EWS::FLI1 binding to both induced and repressed targets to alter transcription (Fig. 5e, f, Supplementary Figs. S10c, S11 a-d; note scales in 5c, 5e).

**MMA alters EWS::FLI1 binding and activity heterotypic response elements**

Next, we focused on how mithramycin impairs EWS::FLI1 activity at tandem sites bound with established cofactors, E2F, RUNX2 and AP-1[51,54,55]. It has been shown by X-ray crystallography that MMA disrupts binding of a complex of the FLI1 DNA binding domain and RUNX2 via a direct interaction between the side chain at the 3-position of MMA and RUNX2[28] but not FLI1 itself. GGAA-RUNX2 and GGAA-E2F cofactor sites proved challenging to evaluate genome-wide because these motifs were under-represented in the CUT&Tag data and found in only 2–6% of EWS::FLI1 peaks (see motif analysis supplementary

Figs. S8a, S9a). Therefore, we first focused on AP-1 tandem sites identified in 10% of differentially bound peaks and combined the analysis with an evaluation of all of the limited number of established EWS::FLI1 tandem binding sites in the literature for all three tandem motifs.

MMA disrupted EWS::FLI1 binding at the tandem EWS::FLI1-AP-1 motif (Fig. 6a, c). This effect favored repressed targets and LCE (Fig. 6a, c, Supplementary Fig. S12a, S12c). Similarly, consistent with X-ray crystallography data, disruption in binding was also seen at EWS::FLI1-RUNX2 sites; an effect that could only be demonstrated by visualizing differentially bound peaks of repressed targets and only with HCE likely due to the small number of peaks (Supplementary Fig. S13a, S13b). Similarly, at tandem EWS::FLI1-E2F sites, reduced occupancy was modestly different than untreated media control (Supplementary Fig. 14a, 14c). Nevertheless, all of the tandem sites established in the literature demonstrated reduced binding of EWS::FLI1 favoring LCE as shown for EWS::FLI1-RUNX2 target genes (*SPP1* and *CNN1*[55]), EWS::FLI1-E2F sites (*ID2, ATAD2, E2F3, RFC2, GEMIN4, RAD51*) and the EWS::FLI1 AP-1 sites *(UPP1, MMP9, HBEGF)* (Fig. 6e–h, Supplementary Figs. S15, S16). The exception was the tandem EWS::FLI1-E2F site at *VRK1* which trended in the opposite direction (Supplementary Fig. S16c)[45]. Careful examination of this binding locus showed that it contained a GGAA microsatellite (consistent with a prior publication[56]). Importantly, at these individual loci, HCE was less specific than LCE causing either a gain in binding (RUNX2 sites), a reduction in only 1 of 3 binding sites (AP-1 sites) or was difficult to ascertain above background (E2F and AP-1 sites). Clear GROseq tracks were not evident for all of these targets but those that were observed did indeed show altered nascent transcription with reduced EWS::FLI1 binding as shown in representative tracks (Supplementary Fig. S17a–e).

These data suggest that MMA stabilizes binding at isolated GGAA motifs and GGAA microsatellites to alter transcription initiation, particularly with LCE. In addition, MMA causes a loss of binding of EWS::FLI1 containing complexes to chromatin to alter transcription, consistent with published fluorescent anisotropy and X-ray crystallography data[28]. Specificity favors LCE relative to HCE in the magnitude change in binding, the effect on nascent transcription, the specificity for the enhancer and the limitation of the non-specific effect on elongation. The limitation in this data is the relatively small number of cofactor sites that have been definitively linked to cooperative activity of EWS::FLI1 containing complexes.

Finally, at subsets of target genes, enrichment was anecdotally associated with the initiation of antisense transcription on the opposite strand at the EWS::FLI1 peak as shown for *RCOR1* which could, in theory, alter transcription either by antisense mechanisms, competition between forward and reverse transcription, or perhaps due to conflict between RNAPII moving on opposite strands in the opposite direction (Fig. 4f). Similar opposite strand transcription initiation and conflict was observed at other loci including *PRKCB, GRK5, CACNB2, GSTM4, AKAP7* (Supplementary Fig. S18a–e). One last mechanism of EWS::FLI1 modulation by MMA was observed at sites where ETV6 has

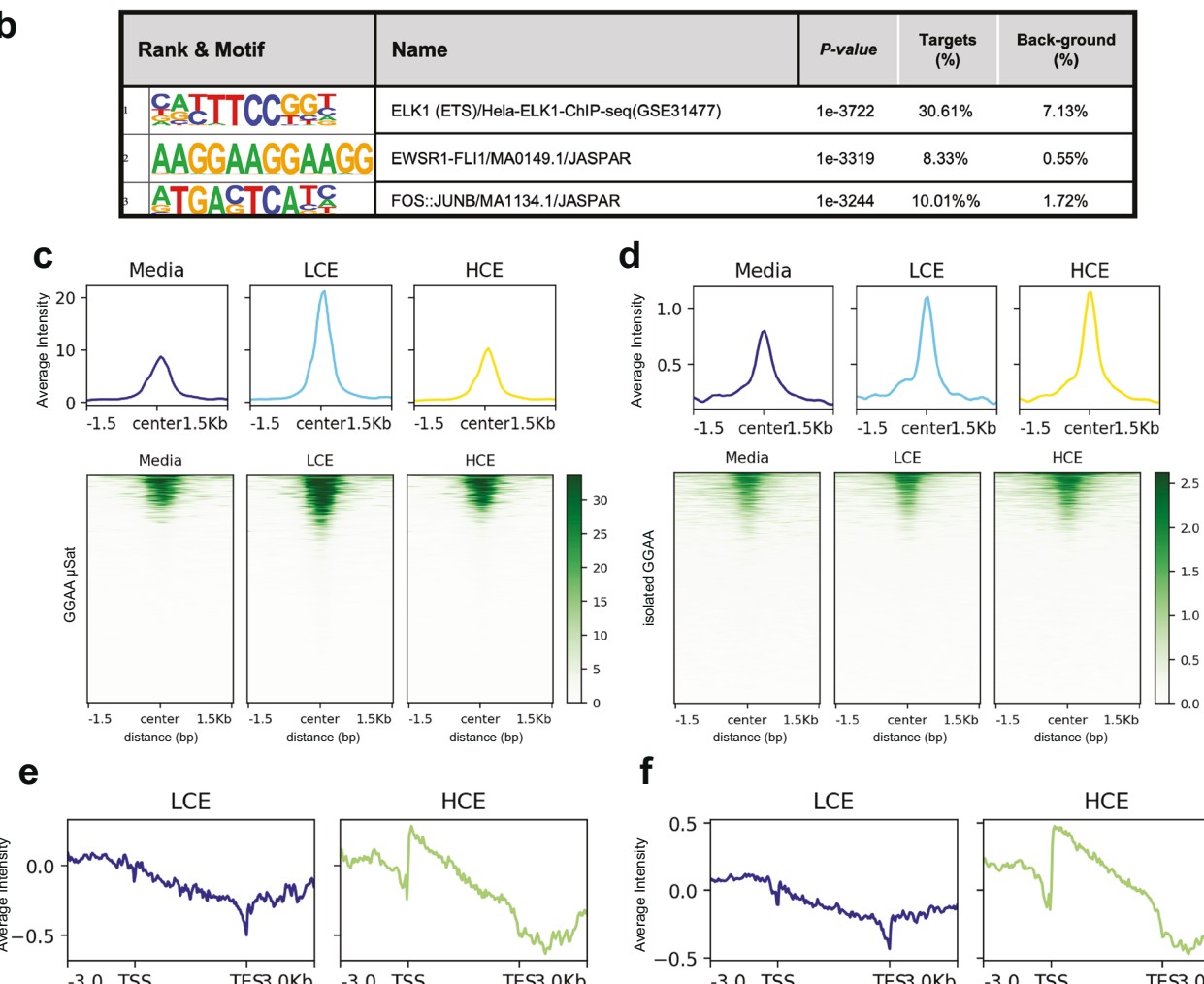

**Fig. 5 | Mithramycin alters EWS::FLI1 binding at multiple different ETS elements.** Homer **a** known or **b** de novo motif analysis of CUT&Tag sequencing data obtained after exposure to LCE. (The enrichment *P*-value was calculated using the cumulative hypergeometric distribution of motif vs. background). **c** Profile plot (top) of average normalized EWS::FLI1 CUT&Tag signal intensity and corresponding heat map (bottom) of EWS::FLI1 CUT&Tag signal centered on all GGAA microsatellites annotated by Homer to EWS::FLI1 induced targets (*n* = 445) following exposure to media, LCE, or HCE. Data demonstrates enrichment with LCE > HCE. **d** Corresponding profile plot (bottom) of average Log₂FC (treated to control) GROseq signal intensity aligned from TSS to TES to these GGAA-microsatellite

annotated EWS::FLI1 induced targets showing loss of nascent transcription initiation and productive transcription favoring LCE. **e** Profile plot (top) of average normalized EWS::FLI1 CUT&Tag signal intensity and corresponding heat map (bottom) of EWS::FLI1 CUT&Tag signal centered on isolated GGAA motifs annotated by Homer to EWS::FLI1 induced targets (*n* = 857) following exposure to media, LCE, or HCE. Data demonstrates enrichment with LCE = HCE. **f** Corresponding profile plot (bottom) of average Log₂FC (treated to control) GROseq signal intensity aligned from TSS to TES at these isolated GGAA-motifs annotated to EWS::FLI1 induced targets showing loss of nascent transcription initiation and productive transcription with LCE.

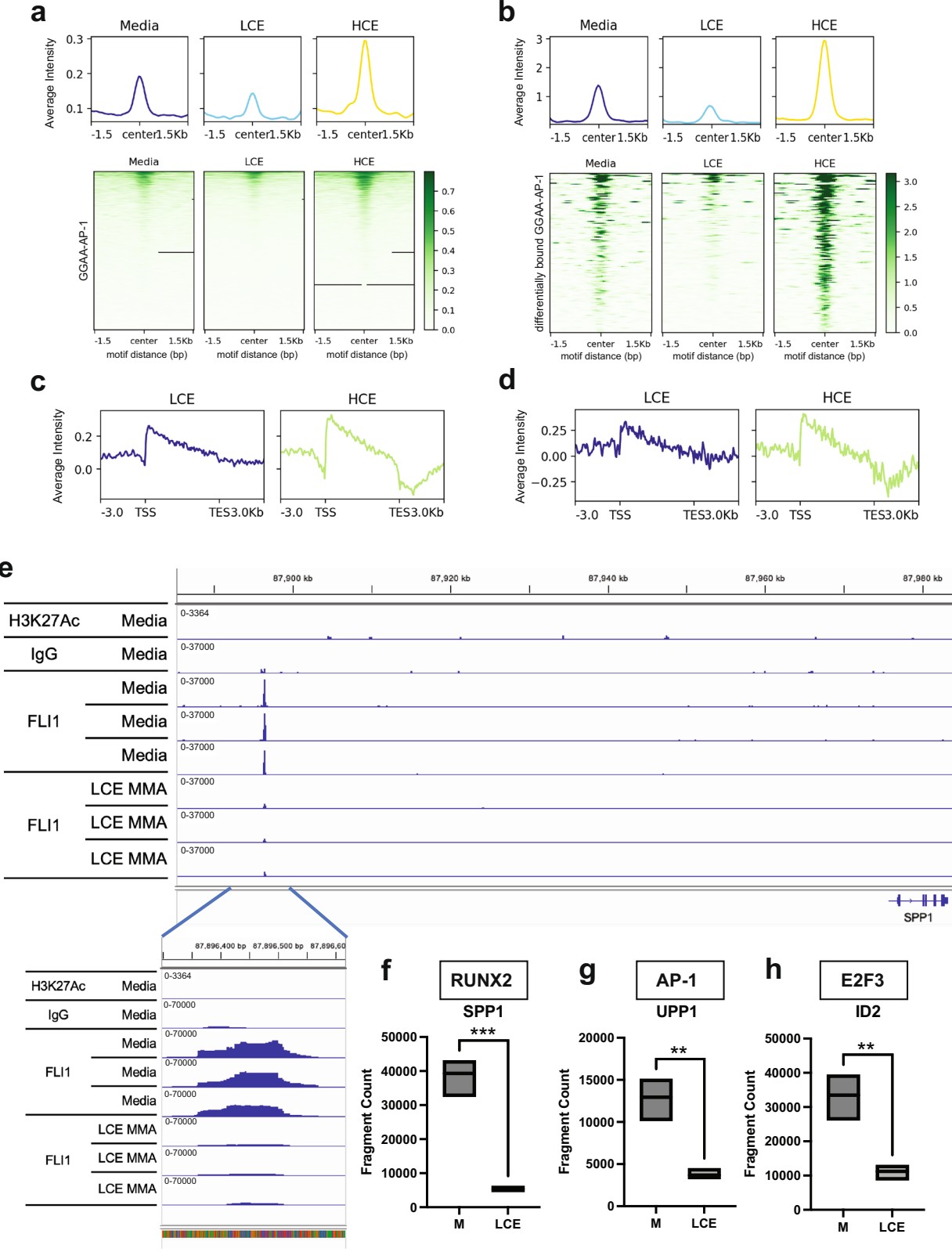

been shown to attenuate EWS::FLI1 driven transcription[57,58]. Here, MMA increased occupancy of EWS::FLI1 at the well-described *SEMA5B, FAS, ACTA, SOX11* loci and overcame ETV6 suppression to induce transcription. In contrast, reduced EWS::FLI1 binding at the *TRIBL* locus paradoxically increased EWS::FLI1 driven transcription presumably because both EWS::FLI1 and ETV6 are ETS factors (Supplementary Fig. S19a–d).

**Mithramycin is more effective in preclinical models when given as a continuous infusion**

Next, we utilized an in vivo model to determine if these molecular differences observed with different drug exposures translate to improved activity and reduced non-specific effects in a xenograft model. We directly compared MMA continuous infusion administered using an Alzet osmotic infusion pump to intermittent bolus dosing in a

**Fig. 6 | Mithramycin disrupts binding at EWS::FLI1 heterotypic complexes.**
**a** Profile plot (top) of average normalized EWS::FLI1 CUT&Tag signal intensity and corresponding heat map (bottom) of EWS::FLI1 CUT&Tag signal centered on tandem GGAA-AP-1 motifs annotated by Homer to EWS::FLI1 repressed targets ($n = 1106$) following exposure to media, LCE, or HCE. Data demonstrates enrichment with HCE but loss of binding with LCE. **b** Corresponding profile plot (bottom) of average Log$_2$FC (treated to control) GROseq signal intensity aligned from TSS to TES to these tandem GGAA-AP-1 motifs annotated EWS::FLI1 repressed targets showing increased nascent transcription initiation and productive transcription with both LCE and HCE. **c** Identical profile plot (top) and heatmap (bottom) visualizing only differentially bound tandem GGAA-AP-1 motifs annotated to repressed targets highlighting impaired EWS::FLI1 binding with LCE and increased occupancy with HCE. **d** Corresponding profile plot (bottom) of average Log$_2$FC (treated to control) GROseq signal intensity aligned from TSS to TES of only differentially bound GGAA-AP-1 annotated EWS::FLI1 repressed targets showing increased nascent transcription initiation and productive transcription with both LCE and HCE. **e** IGV view showing CUT&Tag data at the *SPP1* locus generated using H3K27ac, FLI1, or IgG control antibodies to tagment DNA following exposure to media (top) or LCE for all biological replicates. Call out showing an expanded view of the reduction in occupancy following exposure to LCE. Quantitation of occupancy of FLI1 at tandem FLI1 (GGAA) **f** RUNX2 (*SPP1*) **g** AP-1 (*UPP1*) or **h**. E2F3 *(ID2)* loci. Box plots show minimum to maximum value and the mean for 3 independent biological replicates for each condition. (two tailed t-test ***$P = 0.0006$; **$P < 0.006$)(see source data for full statistics).

TC32 Ewing sarcoma xenograft. We previously reported excellent activity and near tumor elimination in vivo when MMA was given as 1 mg/kg IP QOD * 7 for a total dose of 7 mg/kg[14]. Therefore, we reduced the total dose of drug 40% from 1 mg/kg daily to 0.6 mg/kg daily IP X 7 daily for a total dose of 4.2 mg/kg in an attempt to draw out differences in schedule (Fig. 7a–c). Although one tumor briefly regressed with the intermittent bolus schedule, most of the tumors grew through this schedule. In contrast, striking tumor regressions were observed in most mice who received the identical total dose of drug (4.2 mg/kg) continuously over the same 7-day period via infusion pump. Importantly, while a second infusion would likely further extend survival and would be possible in the clinic, we were unable to retreat the animals due to appropriate IACUC limitations on additional surgery.

Finally, to exclude non-specific DNA damage as the cause of the tumor response, we collected tissue on day 2 (see arrow in Fig. 7c), just prior to tumor regression, and stained for γH2AX, a marker of DNA double strand breaks. There was no appreciable increase in DNA damage in MMA treated tumors relative to vehicle with either schedule (Fig. 7d, e, Supplementary Fig. S20). Further, it is known that MMA achieves far higher concentrations in the mouse than in patients[15]. Therefore, the drug does not derive activity by non-specific DNA damage in this model or at doses that are achievable in patients.

Importantly, there was no difference in toxicity between the two treatment schedules. Both treatment cohorts experienced a transient, but statistically significant, weight loss relative to saline vehicle control ($P < 0.01$)(Supplementary Fig. S21). However, all animals recovered immediately after drug cessation, and blood chemistries (see ref. 59 for reference values) did not show evidence of kidney or liver impairment (Supplementary Fig. S22). Attempts to quantify the pharmacokinetics of the drug in serum were unsuccessful. Nevertheless, the data clearly demonstrated that MMA is more effective as a continuous infusion at the same total dose of drug. This improvement in activity is not due to non-specific DNA damage and suggests an on-target difference between the two schedules consistent with the preclinical modeling. Although appropriate IACUC restrictions on additional surgeries limited the ability to retreat the animal which limited the overall response, the limit in the response was due to the model, not the drug.

### The less toxic mithramycin analog AIT-102 is more effective as a continuous infusion

Finally, to increase the likelihood that these effects could be realized in patients, we evaluated the impact of continuous infusion of the MMA analog AIT-102 on the growth of Ewing sarcoma xenografts. AIT-102 is a second-generation analog of MMA that is between 20 and 40 times less toxic than MMA in multiple species and currently under development for the clinic[15]. We previously reported that the compound inhibits EWS::FLI1 at the same drug exposure as MMA in vitro. In our previous publication, TC71 xenograft tumors recurred immediately after cessation of treatment when AIT-102 was administered using an intermittent bolus schedule[15]. Therefore, with the new mechanistic information in hand, we administered AIT-102 as a continuous infusion in the TC32 xenograft model at 40 or 50 mg/kg, which is far lower than the 192 mg/kg total dose that was given by intermittent bolus dosing in the previous publication[15]. Every mouse in both cohorts demonstrated striking regressions of their tumors (Fig. 7f–h). These were sustained responses with one mouse in both cohorts demonstrating complete tumor elimination. Additional retreatment could not be performed to extend these responses in a larger number of animals due to appropriate IACUC limitations on additional surgery. But this could be done in the clinic.

In summary, mithramycin is a minor groove DNA binding compound that inhibits EWS::FLI1 by either stabilizing DNA binding at GGAA microsatellites or interfering with binding at tandem GGAA EWS::FLI1 response elements. These effects are specific for EWS::FLI1-driven transcription and this specificity favors lower concentration continuous exposure and translates into increased activity as a continuous infusion in vivo. The mechanism that causes toxicity is likely non-specific limitation of transcription elongation[16].

## Discussion

Natural products are compounds that have evolved in nature for a purpose and as a class are the most effective anti-cancer agents in the clinic. These drugs frequently have broad cytotoxic profiles due to non-specific mechanisms of action. However, subsets of compounds may derive their broad activity profile because they modulate the expression or activity of proteins, complexes, or protein families that are important to numerous cancer types. This presents an opportunity to pair specific compounds to specific tumors with a heightened sensitivity and optimize dose, schedule, and synthetic lethal combinations. Therefore, the identification of compounds that fit these characteristics is a promising area of study and perhaps these drugs should be thought of as a class called transcription factor modulators.

MMA may be the paradigm for this class of molecules. This compound was one of the first identified anti-cancer agents that found utility in testicular cancer along with a broad range of other tumors. Importantly, the drug was active in Ewing sarcoma as a 7-day infusion where it induced a complete response in a patient with metastatic Ewing sarcoma[30]. It was ultimately felt to be too toxic; an observation attributed to non-specific transcription inhibition based on in vitro work done at roughly 40 times the achievable serum concentration[38,39]. We rediscovered the drug as a compound that inhibits the driver oncogene of Ewing sarcoma, the EWS::FLI1 transcription factor[14]. We reintroduced it to patients but administered it as a daily infusion instead of a continuous infusion. In contrast to the activity seen in the 1960s, only one patient had brief disease stabilization[31].

In the current study, we sought to model these exposures and determine, using CUT, Tag and GRO, if there is a molecular rationale to administer the drug at a lower concentration for longer time. Indeed, we showed more specific blockade of EWS::FLI1 binding, less non-specific impact on transcription elongation, and clear superiority as a continuous infusion in preclinical models compared to bolus dosing. The continuous exposure drew out the overlap between the preferred

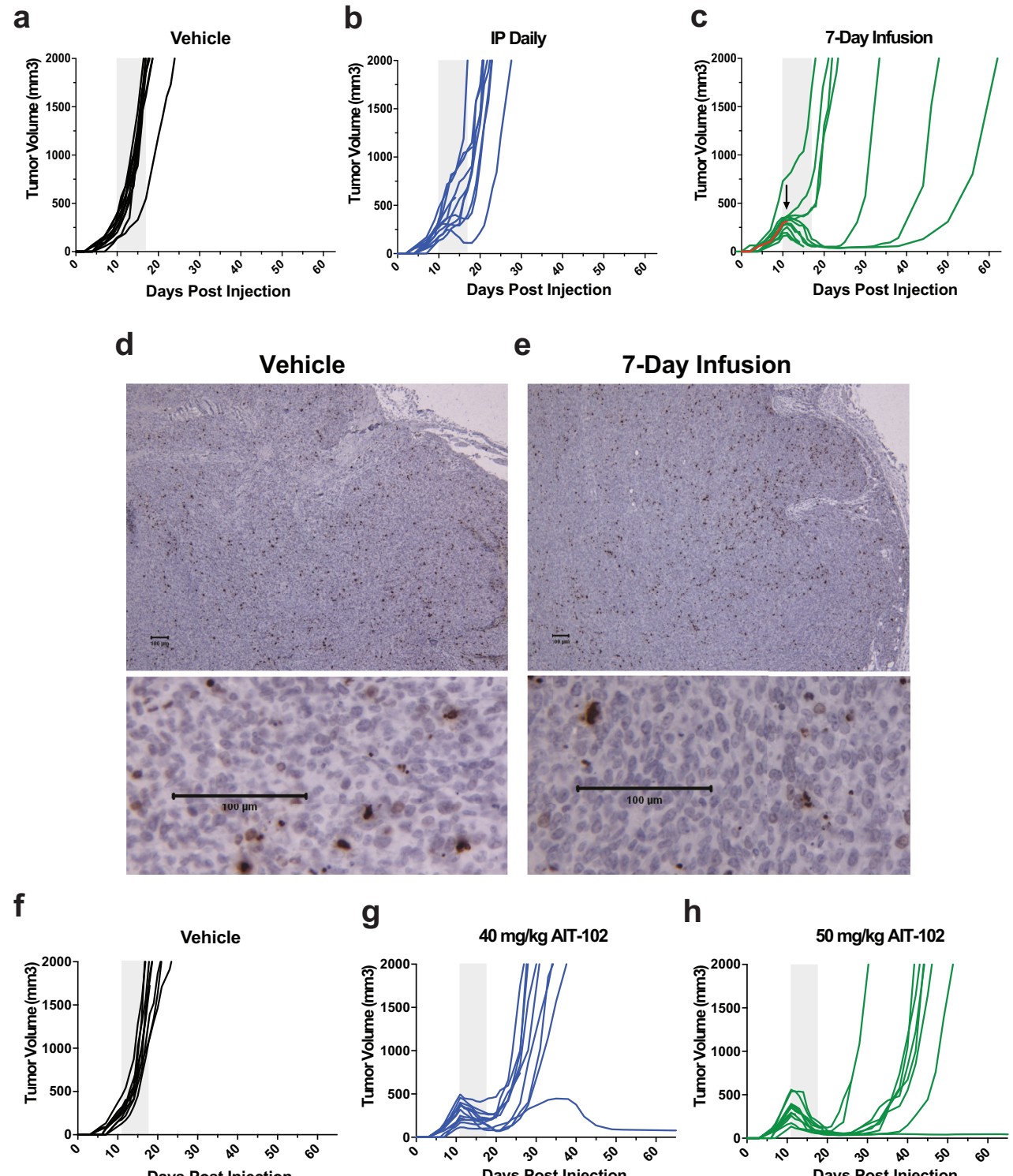

**Fig. 7 | Mithramycin is more effective in vivo as a continuous infusion.**
Mice bearing a TC32 xenograft were treated with **a.** Vehicle (V, black, *n* = 11)
**b**. 0.6 mg/kg/day Intraperitoneal X 7 daily (IP, blue; *n* = 12) or **c** 0.6 mg/kg/day
continuously (C, green; *n* = 12) and the effect on tumor growth was measured daily.
Each line represents an individual mouse, the gray box indicates duration of
treatment, and the arrow and red line indicates the mouse and day of tissue col-
lection. Comparisons by mixed effects model with Tukey post-hoc correction; V vs
IP (*P* = 0.0007), V. vs. C (*P* < 0.0001), IP vs C (*P* = 0.0087). **d**, **e** There is no increase in
DNA damage as measured by γH2AX staining of control or treated (7-day infusion)

tissue collected from the highlighted animal (red line, arrow) on day +2 of treat-
ment at the inflection point of tumor regression. Spaghetti plots showing the
growth of individual tumors in animals bearing TC32 xenografts treated with
**f.** Vehicle (black; *n* = 10) **g** 40 mg/kg AIT-102 IP continuously (40 C, blue; *n* = 12)
over 7 days **h** 50 mg/kg AIT-102 IP continuously over 7 days (50 C, green; *n* = 12).
Comparisons by mixed effects model with Tukey post-hoc correction; V vs 40 C
(*P* = 0.0024), V. vs. 50 C (*P* < 0.0031), 40 C vs 50 C (*Not Significant*). See source
data for full statistics.

binding of GG/GC of MMA and the ETS factor GGAA binding sequence with highly significant enrichment of this motif. These data clearly show a window of activity where EWS::FLI1 is blocked without inducing either DNA damage or general transcription blockade thus paving the way for the second-generation, less toxic analog, AIT-102. AIT-102 has tremendous therapeutic potential for Ewing sarcoma and as an ETS factor modulator which is known to be mutated in prostate cancer, GBM, AML, and ALL[60]. In addition, the drug could find utility as a competitive inhibitor for minor groove binding transcription factors and complexes such as mutated SWI/SNF since the effects on mutated SWI/SNF occurred at similar concentrations as described in the current report[20].

This study provides significant insight into transcription factor drug development and advances our understanding of the biology of transcription. First, this study provides a new gold standard assay for transcription factor drug targets which traditionally lack the equivalent of a kinase assay to determine if a compound is an inhibitor. CUT, Tag, and GRO definitively measures the impact of drug exposure on the DNA binding of a transcription factor and the associated impact on nascent transcription.

Second, this study does indeed confirm the "Goldilocks hypothesis" that states that either more or less DNA binding of a transcription factor is sufficient to poison transcription. In Ewing sarcoma, groups have sought to either inhibit EWS::FLI1 or block degradation and the associated ubiquitin ligase[49,61]. Here, we definitively show, using one compound in the same context, that both approaches should be effective. We show that either an increase or decrease of EWS::FLI1 occupancy at downstream target response elements is sufficient to poison transcription. We also make the anecdotal observation that enrichment of EWS::FLI1 on chromatin in subsets of targets causes transcription initiation on the opposite strand which either competes for initiation, provides antisense transcripts, or may cause transcriptional conflict between RNAPII moving in opposite directions on opposite strands.

This study also points to the complexity of transcription as a challenge to drug development. EWS::FLI1 shows tremendous flexibility to modulate the expression of important downstream target genes. It can bind at isolated GGAA sequences in complex with other proteins such as RUNX2, AP-1 family members or E2F, or phase transition at GGAA microsatellites to both induce or suppress expression of target genes. The peaks at these isolated GGAAs show fragment counts 10 to 100 times lower than those at microsatellites and yet these small magnitude binding events regulate critical cell processes such as the cell cycle (E2F), inflammatory response (AP-1) and extracellular matrix interactions (AP-1)[45,51–54,62]. It is not clear which one, or how many, of these small magnitude binding events are druggable or how a molecule could target these interactions with so much more EWS::FLI1 deposited at microsatellites. In addition, it is not clear if a molecule could be designed that binds to EWS::FLI1 to interfere with all of these different heterotypic interactions.

Finally, this study serves as compelling proof of principle that schedule matters particularly for this class of compounds and that more drug is not necessarily better. We have shown this in a series of studies of a natural product transcription factor modulator, trabectedin; observations that were recently supported in the clinic. The current study provides a similar hope for the class of mithramycins particularly for the second-generation analog AIT-102 with a far superior toxicity profile, that is advancing in the NCI-NExT program. It is likely that the traditional phase I approach of dose escalating to toxicity may be flawed for AIT-102. Nevertheless, recognizing the value of continuous dosing schedules will be pivotal in unlocking the full therapeutic potential of this promising class of drugs.

## Methods
All work has been performed consistently with institutional guidelines. Approval from the relevant IACUC and IBC committees at Van Andel Institute, Children's Hospital of Philadelphia, and University of Michigan was maintained throughout the study. Reagents info to be found in Supplementary Data 1.

### Cell lines
TC32 cells were obtained from Dr. Lee Helman (Osteosarcoma Institute), TC252 cells were a gift from Dr. Tim Triche (Children's Hospital of Los Angeles, Los Angeles, CA). Cell lines were routinely screened for mycoplasma and by short tandem repeat profiling approximately annually (DDC Medical and Penn Genomic and Sequencing Core). The cells were cultured at 37 °C with 5% $CO_2$ using RPMI-1640 (Gibco) with 10% FBS (R&D Systems, bio-techne), 2 mmol/L L-glutamine, and 100 U/mL penicillin with 100 µg/mL streptomycin (Gibco).

### Quantitative PCR
RNA was collected (RNeasy-Kit; Qiagen), reverse transcribed with a High-Capacity cDNA Reverse Transcription Kit (Applied Biosystems) at 25 °C for 10 min, 37 °C for 120 min, 85 °C for 5 min. qPCR was performed using SYBR green (BioRad) at 95 °C for 10 min, 95 °C for 30 seconds, 55 °C for 30 s, and 72 °C for 30 s for 40 cycles. Expression of NR0B1 and PHLDA1 was determined relative to GAPDH and a solvent control using standard $\Delta\Delta C_t$ methods and a one-way ANOVA with Dunnett's multiple comparison test.

### Western blot analysis
Protein was collected following MMA incubation and PBS washing, lysed in 4% lithium dodecyl sulfate (LDS), quantitated after diluting detergent using a bicinchoninic acid assay (BSA) (Pierce, Thermo Scientific), and 30 µg was resolved on a NuPage 4–12% BisTris gradient gel (Invitrogen) in 1x MOPS running buffer (Invitrogen). The protein was transferred to nitrocellulose at 20 V in 1x Tris Tris-Glycine SDS Buffer (Novex) for 18 h, blocked in 5% milk in and probed (see key resources table). All westerns in the manuscript performed between 3 and 7 times with no failed experiments although some bands that did not transfer well in a limited number of replicates. See RRID table for method of antibody validation.

### Chromatin fractionation
TC32 nuclei were collected by dounce homogenization, and isolated in 320 mM sucrose, 8 mM Tris-HCl (pH 7.5), 4 mM MgCl2, and 0.8% Triton-X as described[63]. The cytoplasmic fraction was isolated at $1500 \times g$ for 5 min at 4 °C. The fractionation of the nuclear soluble and chromatin bound fractions were performed as previously described[63].

### Caspase 3,7 apoptosis assay
TC32 cells were exposed to MMA for either 18 or 72 h relative to Medium, Solvent, Etoposide or Lurbinectedin, incubated with 1:1000 diluted Caspase 3/7 detection reagent (ThermoFisher Scientific). Apoptosis was determined by automated counting of green fluorescence using the CellCyte real time cell imaging system and quantitated at assay time end-points with a one way anova and tukey's multiple comparison test. Presented data representative of 3 independent experiments with no experimental failures.

### Global run-on sequencing
$4 \times 10^6$ TC32 cells from 3 biological replicates for each condition were cooled and maintained at 4 °C, collected and incubated in RNase-free buffer containing 10 mmol/L Tris-HCl (pH 7.5), 200 µmol/L MgCl2, and 300 µM CaCl2, resuspended in buffer plus 10% glycerol and lysed by incrementally adding NP-40 (1% final). $10^7$ nuclei per 100 µL was frozen in aliquots in 40% glycerol, 50 mmol/L Tris-HCl (pH 8), 5 mmol/L MgCl2, and 50 µmol/L EDTA and stored at −80 °C. Nuclear run-on occurred at 30 °C for 5 min and 850 rpm in a thermomixer (Eppendorf) in a 1:1 solution with 500 µmol/L N-laurylsarcosine, 10 mmol/L Tris-HCl (pH 8), 5 mmol/L MgCl2, 300 mmol/L KCl, 1 mmol/L DTT, 500 µmol/L

ATP, 500 μmol/L GTP, 500 μmol/L 5′-BrUTP, 2 μmol/L CTP, and 200 U/mL SUPERase In RNase Inhibitor (Invitrogen). RNA was isolated using Trizol LS Reagent (Invitrogen) and a Direct-zol RNA Kit (Zymo Research), fragmented using 1X Magnesium RNA Fragmentation Module (NEB) incubated at 94 °C and 850 rpm to obtain fragment lengths between 100 and 150 nucleotides. RNA fragment length was analyzed using an RNA Nano Chip (Agilent Technologies) on a 2100 Bioanalyzer system (Agilent Technologies). 50 μL of BrdU Antibody (IIB5) Agarose Conjugated Beads (Santa Cruz Biotechnology) per sample were pre-blocked washed, and resuspended in a binding buffer of 0.5X UltraPure SSPE, 1 mmol/L EDTA, 0.05% Tween 20, and 1 μL/mL SUPERase at room temperature for 1 h. Bead-RNA conjugates were then centrifuged and washed with the following sequence of buffers; a low salt buffer containing 0.2X UltraPure SSPE, 1 mmol/L EDTA, 0.05% Tween 20, and 1 μL/mL SUPERase In; a high salt buffer containing 0.2X UltraPure SSPE, 1 mmol/L EDTA, 0.05% Tween 20, 150 mmol/L NaCl, and 1 μL/mL SUPERase In; and TE Buffer (pH 7.4) with 0.05% Tween 20 and 1 μL/mL SUPERase. RNA was eluted in buffer containing 50 mmol/L Tris (pH 7.5), 150 mmol/L NaCl, 0.1% SDS, 20 mmol/L DTT, 1 mmol/L EDTA, and 10 μL SUPERase In at 90 °C and 1000 rpm. RNA was purified using a Qiagen RNeasy kit. cDNA libraries were generated using a NEBNext Small RNA Library Prep Set for Illumina kit (New England Biolabs) and sequenced on a NextSeq500 with 75 bp, single end reads, to a depth of 25 M reads/sample. Base calling was done by Illumina NextSeq Control Software (NCS) v2.0 and output of NCS was demultiplexed and converted to FastQ format with Illumina Bcl2fastq v1.9.0. All biological replicates represent separate dishes one LCE replicate failed during library prep. Data compared to solvent which is <0.01% PBS.

## CUT&Tag

CUT&Tag was performed using a CUT&Tag-IT anti rabbit assay kit (Active Motif). $1.5 \times 10^6$ cells TC32 cells were incubated with MMA with either LCE or HCE and counted, nuclei and collected from 3 separate $10 \text{ cm}^2$ plates per condition, isolated by dounce homogenization in 320 mM sucrose, 8 mM Tris-HCl (pH 7.5), 4 mM MgCl2, and 8% Triton-X with Protease inhibitor cocktail (PIC; Active Motif), recounted and 100,000 nuclei were resuspended in 400 μL binding buffer + PIC (Active Motif) and immobilized on 20 μL of activated Concavalin A beads in binding buffer + PIC (Active Motif). Nuclei-ConA beads were resuspended in 40 μL antibody binding buffer + PIC + 5% Digitonin (DIG) (Active Motif), combined with 10 μL antibody binding buffer + PIC + DIG containing 1 μL of either H3K27Ac (Active Motif), total rabbit IgG (Cell Signaling), or FLI1 (Abcam) at 4 °C with gentle agitation for 18 h. Primary was removed, washed in Dig Wash buffer, and incubated in1 μL secondary antibody + PIC + DIG (Guinea Pig Anti-Rabbit (Active Motif)) for 1 h at RT with agitation. The samples were washed, incubated with pA-TN5-Transpososomes (Active Motif) in DIG-300 buffer + PIC + DIG, washed, incubated in tagmentation buffer + PIC + DIG (Active Motif) at 37 °C 1 h. The reaction was stopped with proteinase K stop solution (Active Motif) at 55 °C for 1 hour and column purified on the DNA Purification Column with binding buffer added (Active Motif). The tagmented DNA was next qPCR amplified, indexed with one i5 and one i7 primer per manufacturer's instructions and the following PCR program: 72 °C for 5 minutes, 98 °C for 30 s, 14 cycles of 98 °C for 10 s and 63 °C for 10 s, 72 °C for 1 min then purified at 1.1x SPRI beads to purification volume, quantitated on the Qubit. Pre-made libraries were sent to Novogene for sequencing on the Illumina NovaSeq X Plus Series PE150 platform. Data generated using two medium control and one solvent control ( < 0.01 PBS) with correlation matrix establishing no impact of solvent but referred throughout manuscript as medium control.

## Genomics data analysis: GROseq

Following adapter trimming and low quality sequence removal by cutadapt v2.10[64], GROseq reads were aligned to hg38 using STAR

v2.7.1a[65]. Sequence quality was additionally confirmed by fastqc. Additional quality control for GROseq was done using NRSA tools[66]. Reads mapping to rRNA loci and reads with mapping quality less than 60 were removed. After read mapping and filtration, reads around the TSS ( + 30 bp,-300bp),TES (TES-30bp, +300 bp) and Gene Body (TSS + 30 bp,TES-30bp) of genes were quantified using bedtools coverage v2.29.2[67]. Heatmaps were generated with deeptools v3.5.1[68]. For visualization, log2ratio GRO-seq signal of treatment vs. media (or solvent) was calculated using deeptools bamCompare. All tracks were group scaled to the highest peak in control.

## CUT&Tag bioinformatics and motif analysis

Sequence quality was first assessed with FastQC (v0.12.1). Sequencing reads were trimmed to remove adapter sequences and low-quality bases using Trim Galore (v0.6.6). Trimmed reads were then aligned to hg38 with bowtie2 v2.3.4.2 with parameters "--local --very-sensitive --no-mixed --no-discordant --phred33 -I 10 -X 700" Aligned read pairs on the same chromosome and fragment length less than 1000 bp were kept and filtered on minQualityScore = 2 with samtools v1.15. Peaks were called with macs2 v2.2.4 callpeak and peaks in blacklist regions were removed with bedtools intersect. Only uniquely mapping reads were retained. Reproducibility between replicates was assessed by computing pairwise Spearman correlations of signal intensities using R corrplot. Peaks were called using MACS2 with a $q$-value threshold of 0.05, then Fraction of Reads in Peaks (FRiP) was calculated using bedtools intersect to quantify signal enrichment. DiffBind v 3.16.0 was used to identify differentially bound peaks (padj <0.05) in LCE,HCE,and LCE vs HCE. (HCE vs Media $n = 10475$, LCE vs Media $n = 35119$, HCE vs LCE $n = 15952$). Differentially bound peaks were then further characterized for their associated motifs using Homer for both LCE and HCE.

## Identification of EWS::FLI1 induced or repressed targets by RNAseq

EWS::FLI1 was silenced using an siRNA spanning the fusion breakpoint in biological replicates of 3 per condition using a previously published approach[40] and confirmed by qPCR and a 3-cycle $(2^3)$ change in expression. siEWS::FLI RNA-seq reads were mapped to hg38 with STAR v2.7.1a[65] and transcript expression was quantified with RSEM v1.3.1[69]. Following quantification, induced and repressed targets were identified with DESeq2 v1.32.0[70]. using a log fold change cutoff of +/− 1 and adjusted $P$-value cutoff of 0.01. These lists of induced and repressed targets were used to annotate CUT&Tag peaks below.

## EWS::FLI1 differential binding analysis

*Tandem motif identification:* Genome wide motif locations for AP-1 ($n = 78183$) RUNX2 ($n = 72469$), E2F3 ($n = 82248$), and FLI1 ($n = 73629$) were extracted using fimo v 5.5.8. Isolated GGAAs not within 20 bp of another GGAA were found using BioStrings ($n = 18377396$). Isolated GGAAs intersecting with a differentially bound peak were retained ($n = 137254$). The resulting GGAAs were filtered within 200 bp of cofactor motifs AP-1 ($n = 7737$), RUNX2 ($n = 2207$), E2F3 ($n = 4130$) or FLI1 ($n = 9855$). *Peak annotation of tandem motifs or FLI1 to EWS::FLI1 induced or repressed targets:* Homer annotatePeaks.pl was used to annotate motifs and peaks and were filtered to include GGAAs for EWS::FLI1 induced targets (Log2FC < 1; $P < 0.01$)(GGAA-AP-1 $n = 578$, GGAA-RUNX2 $n = 197$; GGAA-E2F3 $n = 519$, FLI1 $n = 857$) or EWS::FLI1 repressed targets (Log2FC > 1; $P < 0.01$)(GGAA-AP-1 $n = 1106$, GGAA-RUNX2 $n = 307$; GGAA-E2F3 $n = 487$, FLI1 $n = 1060$). *Microsatellite identification and annotation to EWS::FLI1 induced or repressed targets:* GGAA microsatellites were identified using fimo v 5.5.8 ($n = 20,765$), annotated to genes using Homer annotatePeaks.pl, then filtered on EWS::FLI1 induced (GGAA μSat $n = 445$) and EWS::FLI1 repressed (GGAA μSat $n = 487$) targets. *Visualization of EWS::FLI1 binding:* CUT&Tag signal at these motifs and GRO-seq Log2ratio signal at annotated genes were visualized with deeptools computeMatrix, plotHeatmap and

plotProfile (v 3.5.6). For the diff_bind graphs only differentially bound peaks were visualized.

## Ewing sarcoma xenograft studies

$2 \times 10^6$ TC32 cells were injected into the gastrocnemius of 8-10-week old female homozygous nude mice, (Crl; Nu-$Foxn1^{Nu}$)(Charles River Laboratories), housed in a SPF facility, and tumors were established to ~0.3 cm. Animals were group housed in a specific pathogen-free environment with 12 h light cycles at an average room temperature of 74 F and 30% humidity. All experiments were done in female mice because we wanted to reference the previous work with this drug in our lab. That work was done in females to match the gender of the cell lines. Cohort size ($n = 12$) was determined assuming the smallest reduction tumor growth being 60% relative to control, a power of >80% and an alpha of 0.05. Mice were randomized into treatment groups: Mithramycin, 0.6 mg/kg/day was administered in DPBS with magnesium and calcium (Gibco) via intraperitoneal bolus injection for 7 days ($n = 12$), 0.6 mg/kg/day continuously infused for 7 days via an Alzet osmotic pump (DURECT 1007D) implanted within the peritoneum ($n = 12$), or vehicle alone ($n = 11$). The AIT-102 cohorts received 40 mg/kg ($n = 12$) or 50 mg/kg ($n = 12$) in a 7-day infusion pump in DPBS with magnesium and calcium, vehicle alone ($n = 10$), or no treatment ($n = 5$). Pumps were removed within 7 days following treatment. Tumor volume was measured daily and determined using the equation (D x $d^2$)/6 × 3.12 (where D is the maximum diameter and d is the minimum diameter) and body weights recorded. Mice were sacrificed when the tumor diameter reached 2 cm total volume. Tissue was collected and fixed in 10% formalin for immunohistochemical analysis. All experiments were performed in accordance with and the approval of the Van Andel Institute or University of Michigan Institutional Animal Care and Use Committee. Investigators were not blinded to the treatment groups due to the need for surgery.

## Tissue staining and immunohistochemistry

5-micrometer sections of FFPE tissue were mounted and stained with hematoxylin and eosin (Ventana Symphony instrument or manually (Abcam)). Antigen retrieval for IHC was performed using EDTA Decloaking buffer (Biocare Medical), slides were blocked then incubated with the γH2AX antibody (Cell Signaling), secondary antibody (Agilent), counterstained with hematoxylin, and visualized (DAB + ; Agilent).

## Statistics and reproducibility

All experiments are performed with at least 3 experimental replicates as indicted above. Biological and technical replicates are specified in the figure legends and/or in the methods. Binding analysis box plots, In vivo experiments. Quantitative PCR uses a one way ANOVA with Dunnett's post-hoc analysis for multiple comparisons to the same control to minimize type I error. Multiple comparison corrections are used throughout and the exact test is specified in the legend. In vivo comparisons are made using a mixed effects model two-way repeated measures ANOVA to account for missing measurements due to animals reaching end point criteria. Post-hoc multiple comparisons were corrected using Tukey test.

## Reporting summary

Further information on research design is available in the Nature Portfolio Reporting Summary linked to this article.

## Data availability

All materials are commercially available with the exception of AIT-102 which may be available upon completion of material transfer agreement with OrphAI therapeutics. The GROseq, CUT&Tag, and RNAseq data generated in this study have been deposited in GEO with the accession numbers: GSE30554, GSE316977, GSE316978. All western blots, caspase assay data, IGV analysis, supplementary data, and extended statistics including multiple comparison testing, box plot, median, IQR and whiskers are available in the source data files. All data and resources are available from correspondent author Patrick J. Grohar at grohar@med.umich.edu. Source data are provided with this paper.

## Code availability

Standard code was used throughout the manuscript and will be shared if needed upon request.

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

## Acknowledgements

The authors would like to gratefully acknowledge the support of the Hyundai Hope on Wheels Grant (JMG), the Alex's Lemonade Stand Foundation Innovation Award (Northwestern Mutual Grant), F31CA236300 (MHC), F99CA253749 (MHC), Gold in September (G9), and the NCI NExT Program, along with intramural funds from the Van Andel Institute and Children's Hospital of Philadelphia. We would like to thank Chad Punchard and Family for additional support of this manuscript. We would also like to thank Francisco Morís and EntreChem for providing AIT-102 (EC8042) and Brigette Roberts, Paul Boni, Keith Fandrick, Patrick Sarmiere (OrphAI) for continued support of this work.

## Author contributions

Conceptualization: R.K., G.F., E.A.B., M.H.C., M.A., P.J.G. Methodology: R.K., G.F., E.A.B., I.B., M.A., S.M.V., A.J.F., P.J.G. Validation: G.F., E.A.B., M.H.C., M.C.S., L.M.G., R.D.L., A.J.F., S.M.K.G., Z.P.T., J.M.G., P.J.G. Formal Analysis: R.K., G.F., E.A.B., I.B., P.J.G.; Investigation: R.K., G.F., E.A.B., S.S.K.G., M.H.C., M.C.S., L.M.G., R.D.L., A.J.F., S.M.K.G., Z.P.T., J.M.G., P.J.G.; Resources: M.A., P.J.G.; Data Curation: R.K., G.F., E.A.B., P.J.G.; Writing—Original Draft: R.K., E.A.B., Z.P.T., P.J.G.; Writing—Review and Editing: R.K., G.F., E.A.B., S.S.K.G., M.H.C., M.A., L.M.G., R.D.L., A.J.F., Z.P.T., P.J.G. Visualization: R.K., G.F., E.A.B., P.J.G.; Supervision: P.J.G. Funding Acquisition: M.H.C., J.M.G., P.J.G.

## Competing interests

P.J.G. is a consultant for OrphAI Therapeutics. P.J.G., M.H.C. and E.A.B. have a patent for the method of use of mithramycin for solid tumors characterized by mutant SWI/SNF which is not relevant. P.J.G. receives grant support from Janssen Pharmaceutical and in the last three years served on the Ad Board of Jazz Pharmaceutical. S.S.K.G. has stock in Panosome. The remaining authors have no competing interests.

### Declaration of generative AI and AI-assisted technologies

The authors declare that no artificial intelligence or AI-assisted technologies were used to formulate, conceive, write or present the above manuscript.
