## [Transparent Peer Review file · Nature Communications]

Mithramycin Alters EWS::FLI1 DNA Binding and RNA Polymerase II Processivity to Inhibit Nascent Transcription

Corresponding Author: Professor Patrick Grohar

Version 0:

Reviewer comments:

Reviewer #1

(Remarks to the Author)

Kaufmann and colleagues explore the mechanism of anti-neoplastic activity of mithramycin (MMA) and a 2nd generation minor groove binder AT102 in Ewing sarcoma cells. They show that MMA inhibits expression of key genes driven by the EWS::FLI1 oncogenic transcription factor in Ewing sarcoma. By integrating CUT&Tag with Global Run-On Sequencing (CUT, Tag, and GRO), the authors demonstrate how MMA impacts TF binding including the EWS-FLI1 fusion, affecting RNA Polymerase II (RNAPII) processivity. The findings suggest that MMA alters transcription factor occupancy at specific response elements. Additionally, the study highlights AIT-102, a less-toxic second-generation analog, as a promising therapeutic alternative. There is tremendous rigor with regard to the attempt to correlate in vitro drug exposures with in animo PK studies, and the xenograft results are striking. The integration of genomics datasets with an effective, clinically viable agent in a lethal disease is intriguing and warrants publication in Nature Communications. However, several improvements can be made to the manuscript that will enhance its appeal and perhaps make it even more compelling.

Major Revisions:

1. Show average plots and heat maps for CUT&Tag with FLI1, AP1, RUNX2 and E2F3 in the various drug treatment conditions to go along with figures 2 and 3. These plots would help better demonstrate global effects of MMA on these factors.
2. The authors should better integrate their data with DepMap, which TFs among the set the authors focus on are the biggest dependencies? Loss of those TFs is likely to be more consequential.
3. While the specific examples shown/chosen for TF binding sites do support the authors' conclusions, it would be interesting to show heatmaps for all TFs and identify subsets of sites that either retain or lose signal together. These can be teased out using a k-means clustering approach. It might be the case that all changes are concordant, but the finding that EWS-FLI1 fusion binding increases while other factors are lost suggests that different sites might be differentially impacted by MMA.
4. Why were only TC32 and TC252 included in this study? In the EWS field there is increasing appreciation that different cell lines display different sensitivities to fusion knockdown/loss. For example, a quick perusal of DepMap shows that some EWS lines have scores of 0 for FLI1 and EWSR1 while others have very strong dependency scores. The authors should report IC50s for minimum 5 and closer to 10 cell lines across this spectrum that display different sensitivities to the EWS-FLI1 fusion. TC32 is also TP53 WT, is MMA similarly potent in TP53 mutant lines? This is an important point and the authors even mention TP53 in the text.
5. One model is that MMA increases FLI1 binding, which outcompetes upstream adjacent nucleosome binding to reduce the nucleosomal barrier to anti-sense transcription. The authors should plot GRO-seq signal onto a list of bidirectional promoters (~1kb between up and downstream TSSs) to determine whether MMA promotes bidirectional transcription. This would allow the authors to comment more rigorously on anti-sense transcription than the somewhat anecdotal examples provided.

Minor point: Line 159 Error "We next examined"

Also no need to show triplicates in IGV browser track shots, please put a correlation matrix in the supplemental figures with

all replicates contained within it. This should sufficiently convince readers about the high concordance and obviate the need to show more than 1 replicate per condition/ab.

Reviewer #2

(Remarks to the Author)

In this study, the authors revisit the mechanism of action of mithramycin (MMA) on EWSR1::FLI1 (EF1) activity in two distinct administration modes of the drug. By integrating Cut&Tag and GRO-seq data, they characterize the molecular effects of two drug administration strategies, identifying a specific effect of the drug on EF1 oncogenic properties, as well as a non-specific effect on global RNA-Pol II processivity. Based on these findings, they propose recommendations for the future treatment of Ewing patients.

The authors have pioneered the study of the mechanism of action of mithramycin as an inhibitor of EF1 for the treatment of Ewing sarcoma. The present manuscript uses various high throughput methods to further document this inhibitory effect. However, as they stand, the data fail to be fully convincing. Indeed, the authors do not show global analyses, neither for transcriptomic nor Cut&Tag data. They only point out at specific genes or loci which is not sufficiently convincing to support their general conclusions. These conclusions therefore frequently appear as overstatements. Finally, the authors convincingly demonstrate through in vivo xenograft experiments that long-term, low-dose treatment is more efficient in inducing regression of their Ewing model.

Major Points

- 1- Showing how MMA impact on the expression level of genes, and how it impacts on known gene signatures for EF1 activity is essential. On Figure 1, the authors focus only on two EF1 target genes, which is not sufficient to document the global effect of MMA on EF1 targets.
- 2- On Figure 1, EF1 levels fluctuate significantly, with decrease at early time points then increase at later time points. Is this fluctuation linked to a transcriptional regulation? Does LCE or HCE treatment regulate EF1 expression at the RNA level?
- 3- MMA Effect on Chromatin: The chromatin-level effects of MMA are presented only for selected target genes. However, a broader, unbiased analysis is needed to support the conclusions. Showing a global heat map of the variation of EF1 binding sites upon MMA treatment is critical. The authors describe sites with increased binding, other with decreased binding. How does this relate globally with transcription? How does this relate with the different types of EF1 binding sites, single GGAA or microsatellites?
- 4- Peak Comparison (Figure 2): The authors compare EF1 peaks using fragment counts within selected peaks. However, the Methods section mentions the use of DiffBind for differential peak identification, yet no global differential peak analysis is presented in the main text. Indication on how peak counts were extracted and normalized between samples and treatments, and whether the DiffBind package was applied in this analysis? line 252, the author concluded: "Overall, MMA exposure favored enrichment of EWS::FLI1 binding at GGAA microsatellites vs. competitive inhibition". Such a conclusion would require a broader analysis of EF1-regulated and bound genes comparing single GGAA ETS vs GGAA repeat sites on a global scale.
- 5- Motif Analysis (Figure 3): It is unclear how the motif enrichment analysis was performed. There is no mention of the HOMER motif analysis in the Methods section. Is this a differential analysis between LCE or HCE and solvent conditions? What is the normalization process? To conclude that EF1-bound peaks in LCE treatment are more enriched for classical EF1 motifs than in HCE treatment, a differential motif enrichment analysis should be performed, comparing HCE vs. LCE or vice versa using HOMER or another motif enrichment tool.
- 6- Fig. 3c, d, e, f. The authors analyze a very limited number of genes and draw conclusions that the LCE schedule treatment reduces EF1 binding at heterotopic EF1/RUNX, EF1/AP1 or EF1/E2F3 complexes-specific binding targets. Again, proposing such a conclusion require 1- a more global, unbiased approach, not limited to what appear as opportunistic loci and 2- an experimental confirmation using ChIPseq or Cut&Tag. Line 233 states: "these data suggest that MMA disrupts binding of EWS::FLI1 containing complexes to alter transcription". With the presented data, this is clearly an overstatement.
- 7- Line 268 - 271. The authors conclude that MMA treatment increases EF1 binding at EF1 and ETV6 targets and "overcame" ETV6 suppression. However, there is no data to support this hypothesis (e.g., ETV6 + FLI1 tracks, GRO-seq results).
- 8- GRO-seq Analysis: The authors analyze GRO-seq, but there is no mention of quality controls, replicates, or methods used for comparative analyses. The GROseq signal seems to increase bidirectionally at promoters under mithramycin conditions (Fig5a, b, c, d). Sense-oriented GROseq data should be presented to define whether this is a global non-specific increase of transcription (or of background) or whether this is related to transcription of genes. In brief, specificity of the GROseq signal should be documented. On Fig 4d, the map should extend a few kb upstream of the TSS and downstream of the TES to show the difference between gene transcription and background level on intergenic regions. In the absence of such QC, the conclusions on mithramycin effects cannot be interpreted. They should be based on robust statistical analysis of high-quality data.
- 9- Figure 6: it is unclear how EF1-repressed and EF1-induced genes were defined. The authors conclude that HCE has a greater impact on RNAPII processivity as compared to LCE. However, this difference appears quite subtle, and the quantitative analyses in Figures 6b, c, d, and the Supplementary data seem insufficient to support this conclusion. Additionally, the violin plots for TSS and TES are not entirely readable. The same concern applies to Figure 6e, where LCE vs. HCE differences appear subtle. Could the authors provide profile plots to better illustrate these differences?
- 10- On Figure S1 the EF1 binding peaks are located outside of the GGAA repeat which questions the specificity of the peak.

Other Points

- Line 262: "Notably, even in the absence of opposite strand initiation, prominent EWS::FLI1 peaks were frequently associated with enrichment of nascent RNA transcripts at the actual sites of EWS::FLI1 binding at the microsatellites in addition to the transcriptional start sites." Are these data provided? If not, could the authors include this analysis?

- Figure 2, b and c, scales are missing
- Line 175, The statement “We used a FLI1 antibody bound to Tn5” could be misinterpreted, as in the Cut&Tag procedure, a secondary antibody is typically used to direct proteinA_Tn5 to the chromatin. Moreover, could the authors clarify the exact references for the antibodies used, as Active Motif, Cell Signaling, and Abcam offer multiple references for H3K27ac, IgG, or FLI1 antibodies?
- Line 180, The signal detected for H3K27ac cannot be described as prominent in Figure 2d and is even absent in Figure S1
- Line 223 “Fig.3 e and S3b-f” correction: S3c-f ?
- Line 248 “(Figs. 4e, 4f, and 6a-d)” correction S6a-d ?
- Line 261, The genes “AKAP7, GRK5, PRKCB, GSTM4, and GLG1 (Figs. S7a-e)” are mentioned, but these genes do not appear in the correct order in Figure S7, and GLG1 is not illustrated, whereas CACNB2 is. Could the authors correct this discrepancy and ensure the figure matches the mentioned genes?
- Figure S9 plots on TSS and TES are impossible to read / interpret
- Figure 6b, Line 314, mislabeling of plot titles?
- Figure 6f, Could the authors include tracks showing gene localization and EF1 binding to provide additional context and support for the analysis?

Reviewer #3

(Remarks to the Author)

The manuscript by Kaufman et al. describes efforts to understand whether continuous 24h dosing of the EWS-FLI1 inhibitor, mithramycin (MMA) (with this schedule in a current clinical trial), may impact the efficacy of MMA compared to once daily bolus for 7 days, with 21 days off, administration. The rationale for the altered schedule is based on an older protocol that induced responses to Ewing Sarcoma in a patient, versus the hepatotoxicity in a recent therapeutic trial that resulted in dose limitations that precluded achieving concentrations of MMA that was able to inhibit EWS-FLI1 (estimated to be about 3 times too low).

To support this new protocol, the authors investigate how MMA, a known EWS-FLI1 competitor to binding to its core recognition sites, GGA(A/T) DNA response elements, impacts EWS-FLI1 function at different treatment schedules. Using CUT&Tag with Global Run-On-Sequencing (CUT, TAG and GRO), the authors demonstrate that continuous administration at low concentration is more efficacious to inhibit the EWS-FLI1 transcriptome, leading to improved in vivo responses of MMA (at least in one ewing mouse model).

Overall, this is an interesting study that supports the rationale for an ongoing clinical trial of lower continuous MMA in, at least the Ewing Sarcoma cohort of the trial, and provides interesting insights into how MMA inhibits EWS-FLI1, and, as an extension, a window into how EWS-FLI1 functions in Ewing Sarcoma. Importantly, it has been traditionally difficult to study the function of EWS-FLI1, and the use of CUT, TAG and GRO here is innovative, robust, and supportive of the investigators' findings. The primary need is to address some of the in vivo concerns listed below, however I have included additional comments that would also be helpful to address.

Major comments

Do these models die following HCE or LCE MMA (as others have suggested at different doses/schedules (PMID: 36913323) and if so, do they die at different rates? If they do die does that process include caspase activation (as suggested by PMID: 36913323)? This is relevant as active caspases can cleave FLI1 (and perhaps EWS-FLI1).

For me, Fig 1 legend is confusing to follow. “a” is written twice, and, overall, the changes in concentrations and time, and the way that is written, makes it confusing. In addition, the time point is not indicated in the figure legend for A and B. I would suggest re-writing this figure legend and adding more information in the figure in terms of time and concentrations, so the reader can follow this a bit better.

Is 72h not included in Fig. 1D for a particular reason? It would seem for comparison's sake it should be included.

It looks to me that EWS-FLI1 is clearly shifting upwards at the significant time points or concentrations in blots A-D (It is harder to see in D but all the other blots in this particular panel look like they are trending downwards left to right, while EWS-FLI1 has the opposite pattern). Is there a modification of EWS-FLI1 that MMA is causing, and could this modification in itself cause some of the altered function of EWS-FLI1?

The goldilocks hypothesis the authors invoke is interesting but could use more supportive data. It would be interesting, for instance, to know how moving from 20nm to 100nm alters binding of some of the disparate genes (low dose versus high dose that have enhanced versus inhibited EWS-FLI1 binding). Of a similar vein, low dose (which is shown in Fig. 1C and 1D) over time versus high dose over time and how this impacts EZH2, for instance.

What does EZH2 look like at the transcript level, as in Fig. 1F and 1G?

It would seem NROB1 should be blotted for in fig. 2A.

What is being shown in S11? Is this vehicle-treated tissue or MMA-treated tissue?

Fig. 7—some statistics would be helpful here, but more critically, as this is the proof that the continuous low dose schedule is superior, this experiment should be repeated in at least one (and preferably two) more ewing in vivo models.

As the investigators are aware, dose-limiting hepatotoxicity of MMA led to dose reductions that likely were not successful at inhibiting EWS-FLI1. Thus, renal toxicity studies would seem to be helpful comparing the two doses/schedules.

Additionally, the authors have previously demonstrated severe liver toxicity by MMA, although this largely did not manifest in the trials. This would be appropriate to evaluate in the different schedules.

Minor comments

“We next examined how” should be removed from line 159

Its unclear to me how fli1-bound tagmentase leads to only specific binding to FLI1 sites. There should be a reference when this concept is introduced (just parenthetical point as is) and an explanation may be plausible in the text.

Fig S3 title should seem to be “EWS-FLI1 containing complexes to DNA”

Fig S9 should seem to read in the figure “repressed after silencing EWS-FLI1. In addition, the X axis labels blend in and is not clear (the font is too large)

Fig S10 same concern with the Y Axis (the font is too large)

Fig S12 statistics would be helpful comparing relevant measurements

The added comment about EWS-FLI1 not being cloned seems out of place in line 110-111.

-Anthony Faber

Reviewer #4

(Remarks to the Author)

The manuscript by Kaufman et al. characterizes the effects of two mithramycin A treatment regimes in Ewing sarcoma from the genomic level to mouse models. The results are interesting, particularly the dramatic relocalization of EWS::FLI1 for the LCE protocol and the implications for treatment. The seq data is presented in an unusual way and some of the interpretations seem a bit overstated or irrelevant to the data the study has produced. Given the strong interest of the main finding, that MMA can have alternate effects on EWS::FLI1 depending genome location, a revision of this manuscript would be good to see. It's also recommended the seq files be provided for expediency in clearing up some of the ambiguities mentioned below.

Major:

1. Figure 2a – The statement in the legend is that there is a reduction in binding for HCE and increase for LCE. There is some ambiguity in this interpretation because the concentration of EWS-FLI1 differs dramatically in the two samples. It would seem safer to say only that the LCE has more EWS-FLI1 associated with the chromatin fraction.
2. HCE is reported to increase EWSFLI1 binding at RUNX2 sites. While the other sites are described as uninterpretable, can the data for RUNX2 binding be shown?
3. The paper shows examples but lacks some context for the reader. To move the manuscript from becoming anecdotal in its methodology, it might be helpful to know how many EWS-FLI1 peaks in the M sample disappear or diminish in the LCE or HCE, how many increase, and how many novel peaks appear in the LCE.
4. The negative controls appear to be inconsistent. For the GRO-seq data, they are sometimes called “solvent”, Figure 2b-c, and for the rest of the paper called “M”, including Figure 4d and Supp Figs S5 – S8. Figure S6b is labeled both “M” and “solvent”. Also, there appears to be separate “M” experiments for the LCE and HCE based on the data in 2e-f, 4b-c, and 4e-f,

but only one M track is shown in figures 4, 6 and Supp Figures S5-8. If LCE and HCE have their own “M” or “solvent” controls, it should be shown that way, not lumped with one control and the peak magnitudes omitted. It may be better for the seq data to just be made available to reviewers to avoid so much back and forth to understand what the selected figures show.

5. Figure 4D, the GRO-seq tracks have no scale in the y-axis, they’re different heights, and the “Media” track looks a lot like noise. More of the flanking genomic regions need to be shown to help understand what this genome browser view is showing. Figure 6f and the supplement figures S5 to S8 also have no y-axis scale.

6. Does the mention of significance for Fig 4e & f refer to $p < 0.0001$ for either LCE or HCE? The same peaks are shown in both but for 4e, CACNB2 and FEZF1 hardly seem significantly different.

7. Since EWS::FLI1 dismissal or binding relates to single sites or microsatellites, it would be helpful to better characterize the microsatellites. This includes providing details such as length and number of GGAA repeats for the examples shown in Figure 4 and the relevant supplemental figures.

8. The relationship between EWS::FLI1 and antisense transcription described on page 15 is difficult to follow. The genome browser views in Supplemental Figures 7 and 8 appear to show typical divergent transcription, not eRNA-like transcripts, especially since the transcripts diverge from annotated TSSs in most cases, or close enough to suggest alternative start sites. It’s also unclear how phase separation can be inferred from GRO-seq data. Unless the authors provide clearer figures comparing representative examples of each transcriptional state and quantify antisense transcription levels, this argument feels out of place with the rest of the study.

9. The nonspecific increase in transcription is interesting. Can this be verified by a non-seq method, like an EU incorporation assay? It may contradict earlier literature reporting high levels of transcription in ES (PMID: 29513652) and also seems an interesting counterpoint to the possibility that transcriptional stress might contribute to ES survival.

10. Wouldn’t the increase in transcription be better described as “MMA-driven” rather than “EWS::FLI1-driven”? There appears to be a circular logic: the effects of MMA on transcription elongation are categorized into genes repressed or induced by EWS::FLI1 (Figure 6), yet role of EWS::FLI1 is defined by the gene’s response to MMA. A more direct test of whether MMA acts solely through EWS::FLI1 would be to assess RNA Pol II activity following MMA treatment in cells lacking EWS::FLI1, either through knockout, knockdown, or a different cell line. Similarly, the logic of comparing EWS::FLI1 and MMA effects on medium and long genes has limitations. Small genes, often essential and housekeeping, would typically have little expression changes, whereas long genes simply contain more MMA binding sites. The hypothesis is plausible, but needs direct evidence showing MMA’s effects in the absence of EWS::FLI1.

11. The violin plots for TES in Figure 6 and both TSS and TES in S9 are dominated by outliers so that it is impossible or nearly so to see what difference could explain the p-value. Perhaps the full y-axis could be shown by an inset or supplement figure, then the bulk of the data shown with a more reader friendly scale.

12. Figure 6f doesn’t have a gene structure shown. By comparison to 4d, it appears likely that this view is also cropped too close to the start and stop of the gene with too little flanking regions to interpret the signals contained in the gene.

Minor:

1. The term “S” in Figure 1 isn’t defined until the legend for Figure 2. The term “M” isn’t defined until the legend of Figure 4.

2. Figure 2d – The differences in peaks are quite difficult to see, likely because the figure has been stretched too far in the x-direction, or widthwise. If the peaks were not so broad either by increasing the region of the genome viewed or by narrowing the panel’s width, it could be much easier to spot the difference in relative height and the changes in Peak 3.

3. The paragraph spanning pages 12 and 13 was likely meant or should be broken into 2 or more paragraphs, to make the logic of the data interpretation easier to follow.

4. On page 13, the reference to “(Figs. 4e, 4f, and 6a-d)” likely meant to be “S6a-d”.

5. The somewhat excessive discussion in the results section can make the manuscript difficult to read. For example, the mention of “Cmax of the compound in patients” on page 17 is rather out of context in the middle of a description of GRO-seq results in cell culture. The impact of the research is diminished if one must look up too many references to get through the results section.

6. Supplement Figures S2B and S2D are in Greek font.

7. I can’t say I’ve ever seen a complaint about IACUC or any mention of it outside of a methods section in a paper before.

8. The methods discusses RNaseq and a siEWS::FLI1 treatment, which are not found in the paper.

Reviewer #5

(Remarks to the Author)

Specific comments

- The title of the paper really 'buries the lede' by focusing on the Pol2 processivity aspect (at high, clinically-intolerable doses) while ignoring the more translationally impactful finding that the low-dose/continuous exposure schedule was more effective than the bolus schedule, and thus suggests that earlier trials which failed due to hepatotoxicity during escalation could, indeed should, justify revisitation of MMA in trials as a continuous, low dose infusion schedule analogous to the Kofman case study cited as an inspiration for this project. This is hinted in the last paragraph of the discussion, but is perhaps the most revelatory finding of the study.

- Ideally equivalent dosing would have been performed between experiments. Though slight, the difference in total drug exposure between high (1800 nM*hr) and low dose (1440nM*hr) should be discussed as a potential confounder to comparing the schedules and their effects on gene expression. The absence of data comparing toxicity of the two schedules on non-tumor cells/tissues is a short coming. It is not very convincing indicator of toxicity that animals regain some weight after removing the drug, as this weight gain could also correspond to tumor growth. While this could be due to the fact that mice do not recapitulate the toxicity observed in patients, there needs to be more thorough support for these claims and a discussion on this potential limitation. Evaluation of liver function markers or histopathology of liver would be useful to assess the 'non-toxicity' of the low concentration exposure claimed at lines 372-380 and shown in Supplemental figure 12. Given that dose limiting hepatotoxicity was a major limitation of MMA in the clinical trial lead by the senior investigator 2016, the mechanism of which was focus of another paper from the group in 2019, it seems that this key toxicity would at least be mentioned since it is not evaluated.

- It is presumed, but never demonstrated by comparison that the improved specificity at low dose MMA observed in vivo is upheld in the animal model.

- The phospho-H2Ax immunohistochemistry experiment is underwhelming on several points. First, this data is not terribly convincing, as it appears to be derived from one animal harvested at a relatively early time point the tissue examined appears to have come from one animal that was a non-responder in the LCE group, and harvested at a relatively early time point, which begs whether this response is reproducible across animals, and whether DNA damage may appear later in the course of treatment. It is also not surprising that MMA does not induce non-specific DNA double-strand breaks; this has been reported by others investigating lower doses of MMA (10nM Lin, 2021) in Ewing tumor cells, higher doses (25-50nM Scroggins 2018 MMA) in lung or bladder cancer cell lines, and has even been reported to mitigate neuronal cell death induced by DSB (Marakevich, 2020). Finally, while the authors exclude DNA damage as a factor contributing to the growth-control response observed for xenografts LCE MMA, an opportunity is missed to identify a mechanism of tumor growth control. This is unfortunate.

- The experiments with MMA derivative AIT-102 seem appended to the paper, and don't really fit with the rest of the story. For example, are the doses of AIT-102 tested analogous/equivalent to either the HCE or LCE schedule for MMA? Are the mechanisms (affinity/sotchiometry, pharmacokinetics) of AIT-102 in terms of EWS:FLI1-GGAA interaction similar to MMA? It is not clear to me that this adds much to the paper as a whole, is it seems that the authors would need to study this agent in a similar manner either separately, or in parallel to MMA.

Minor comments:

-Line 156-7: A citation here regarding the SWI/SNF chromatin remodeling complexes as loading controls would be helpful.

-Line 159 – There seems to be an orphan start of a sentence orphan immediately following the paragraph heading.

-Line 160: States "To gain a deeper understanding of the impact of drug exposure..." but ignores the difference in total drug exposure being compared is about 20% lower in LCE schedule. This should be phrased more accurately to capture the study that follows.

-Line 175-177: "as is commonly done in the field" provide references qualifying this.

-Line 186: "this is the clearest example of any compound inhibiting EWS:FLI1" this claim seems unsubstantiated as the authors are not looking at multiple compounds here. Please rephrase, or cite literature to support this statement.

-Line 227: "careful examination of this binding locus...microsatellite" provide reference or figure reference.

-Line 264-265: Please provide a reference that this exceeds what is commonly observed for enhancer RNAs

-Line 277: Please provide a reference for what is determined to be a supraphysiological concentration.

-For Figure 4 b/c and 4e, it would emphasize the difference in occupancy LCE vs HCE systems if the two plots were presented on the same scale for normalized fragment count. There is almost a >10x difference between LCE and HCE.

-Much of the text in Supplemental figures 2B, 2D are in a symbol font and un-intelligible

Reviewer #6

(Remarks to the Author)

Version 1:

Reviewer comments:

Reviewer #1

(Remarks to the Author)

The authors conducted the requested analyses when possible. I do note that the authors have misinterpreted my comment regarding FLI1 and EWSR1 dependency scores in the DepMap database, there are several lines with EWSR1 and FLI1 dependency scores that are relatively modest including TC32. This suggests that TC32 is less dependent on the fusion than other lines, and perhaps mithramycin is disrupting other targets in this context that contribute to anti-neoplastic effects. However, none of the additional analyses change the initial conclusions of the paper and have no further comments or recommendations.

Reviewer #2

(Remarks to the Author)

The authors have improved their manuscript by adding technical details that were crucially lacking in the initial version and made valuable effort to include genome-wide analyses.

Yet, many messages of the manuscript are still unclear:

- 1- The effects of treatments on the Ewing signature are extremely subtle (Fig3b, FigS1) and the technical details that are provided are unclear and lack of rigor (example for FigS1 the legend indicate Log2FC while the figure reports log10(cpm + 1) (same in Fig3b) !! Are these ratios or absolute count values? How were these GROseq data normalized to enable these comparisons? this is essential to interpret such subtle (and sometime opposite) differences between LCE and HCE.
- 2- At different levels of the manuscript, it is indicated that solvent (or control) to treated ratios is reported. If this is the case the interpretation of data would be completely different as they would show that mithramycin is indeed a transcription inhibitor! Such inconsistencies along the manuscript questions the quality of the analyses that are shown.
- 3- The genome-wide analyses of Cut&Tag appear biased, focusing only on EF1 (FLI) peaks linked to EF1-induced or -repressed genes based on RNA-seq. But peak to gene annotation is inherently challenging, as different peak types (e.g., GGAA microsatellites or isolated peaks) may map to the same gene, complicating interpretation. A clearer approach would be to first present all EF1 peaks, identify differential peaks and their characteristics (e.g., motifs), and only then integrate transcriptomic data. This would better show how treatment-induced binding changes relate to the EF1 transcriptomic signature. Heatmaps and annotation profiles also need simplification and clearer legends. For example, the "gene" label is misleading when the central sites represent EF1 bound GGAA microsatellites annotated to EF1 induced or repressed genes.
- 4- The methodological choices are unclear. Although the authors indicate that a DiffBind analysis was performed, it is poorly integrated into the results. Why not use DiffBind peak statistics to illustrate variation in binding at key loci (e.g., Fig. 4d, 6d) instead of fragment counts across conditions? Does this mean the peaks of interest were not significant according to DiffBind? There should be coherence between materials/methods and results sections.
- 5- The data on ETV6, E2F, RUNX2 and AP1 are very preliminary and poorly convincing. They are primarily based on motif analyses. Co-binding of EF1 and co-factors have not been validated using cofactor ChIPseq or Cut&Tag which could be obtained from public datasets. Furthermore, it is very unclear what these data bring to the manuscript?
- 6- One of the messages of the manuscript is that mithramycin treatments increase EWS-FLI1 protein expression and, consequently, EWS-FLI1 binding to its DNA sites. This is an interesting, convincing, though unexplained, observation. The message on the consequences on the expression of EWS-FLI1 targets is much less clear.
- 7- The preclinical data are nice. The authors should simplify the rest of the manuscript, only showing robust data with clear indication of the methods that were used.

Reviewer #3

(Remarks to the Author)

I appreciate the authors transparency and thoughtful explanations. I can understand the dilemma concerning the TC252 xenografts and am satisfied with leaving it as a for reviewer only figure. The authors have satisfactorily addressed my concerns.

Reviewer #4

(Remarks to the Author)

The revised manuscript by Kaufman et al. the authors have made a thorough response to points raised by the 4 prior reviewers. The manuscript in its current form is a remarkable improvement.

The authors have responded sufficiently to each of my comments. Specifically, for comment 7 the information about the GGAA repeats described is sufficient and the extra figure isn't necessary. To point #9, I appreciate the authors efforts in performing the EU incorporation assay and agree with their decision about using the data at this time and changes to the text.

In general, I've found this manuscript both more readable and the data analysis easier to follow. I recommend the study for acceptance.

Reviewer #5

(Remarks to the Author)

Thank you for your thoughtful and detailed response to our critique. As I am reviewing this manuscript with a trainee, we

appreciate that your responses have been both deliberate and extensive as well as humble, without being defensive or combative, as some authors' rebuttals can be. This review has provided an excellent example of professionalism in responding to peer review, and we are grateful for this experience - Thank you!

We are generally satisfied with the response to our initial critique, and appreciate clarification of a few key points. This is particularly regarding clarification and corrections regarding the pH2AX staining in figure 7; we agree with the author that additional images are not necessary.

Reviewer #6

(Remarks to the Author)

Version 2:

Reviewer comments:

Reviewer #2

(Remarks to the Author)

The authors have improved their manuscript which may now be suitable for publication.

Aug 4, 2025

Dear Reviewer,

Thank you for considering this article. The suggestions overall have been very helpful and have greatly improved the manuscript. However, this required quite a bit of rewriting and so track changes was not possible in this resubmission. To orient you the resubmission, we needed to make Figs. 6 and 7 of the old version become Figs. 2 and 3 so that we could integrate the effects on RNAPII processivity and EWS::FLI1 binding throughout the manuscript. This change allowed us to make better links to transcription initiation vs. elongation. All of the comments are collected and annotated below. In this cover letter, I highlight the reviewer critiques in **bold** while our response is in normal script. Text from the manuscript is in *italics*.

Reviewer #1 (Remarks to the Author):

Kaufmann and colleagues explore the mechanism of anti-neoplastic activity of mithramycin (MMA) and a 2nd generation minor groove binder AT102 in Ewing sarcoma cells. They show that MMA inhibits expression of key genes driven by the EWS::FLI1 oncogenic transcription factor in Ewing sarcoma. By integrating CUT&Tag with Global Run-On Sequencing (CUT, Tag, and GRO), the authors demonstrate how MMA impacts TF binding including the EWS-FLI1 fusion, affecting RNA Polymerase II (RNAPII) processivity. The findings suggest that MMA alters transcription factor occupancy at specific response elements. Additionally, the study highlights AIT-102, a less-toxic second-generation analog, as a promising therapeutic alternative. There is tremendous rigor with regard to the attempt to correlate in vitro drug exposures with in animo PK studies, and the xenograft results are striking. The integration of genomics datasets with an effective, clinically viable agent in a lethal disease is intriguing and warrants publication in Nature Communications. However, several improvements can be made to the manuscript that will enhance its appeal and perhaps make it even more compelling.

We appreciate this thoughtful and favorable review. We thank you for the suggestions below and performed these analyses. They greatly strengthened the manuscript. Thank you.

Major Revisions:

- 1. Show average plots and heat maps for CUT&Tag with FLI1, AP1, RUNX2 and E2F3 in the various drug treatment conditions to go along with figures 2 and 3. These plots would help better demonstrate global effects of MMA on these factors.**

Thank you for this comment, this analysis has greatly strengthened the manuscript. Please find in this resubmission, average plots, heatmaps, and associated GROseq profile plots from TSS to TES for

GGAA microsatellites, isolated GGAA and tandem GGAA-AP1, RUNX2 and E2F sites. These are now figs. 4b, 5c, 5d, 6a, 6b, Supplementary Figs, S10a, S10b, S11a, S11b, S14a and S14b. As stated above, in the original submission we focused on established targets and looked at **every published** target for each of these transcriptional modulators/cofactors. We agree that this made the manuscript anecdotal. In this revision, these suggested plots clearly demonstrate enrichment of EWS::FLI1 binding at GGAA microsatellites and isolated GGAA and effects that favor disrupted binding at heterotypic complexes. These figures greatly strengthen the conclusions and highlight the CUT, Tag and GRO approach.

2. The authors should better integrate their data with DepMap, which TFs among the set the authors focus on are the biggest dependencies? Loss of those TFs is likely to be more consequential.

Thank you for this suggestion. I think this comment is an important question but difficult to incorporate into this manuscript because it is nuanced. We did this analysis as suggested and the DepMap did not yield a heightened sensitivity of Ewing tumors to any of these factors. In addition, the analysis of the DepMap data was difficult to interpret among Ewing tumors for reasons unique to the individual cofactors. The role of these cofactors in this tumor is somewhat nuanced. For example, RUNX2 did not favor Ewing sarcoma. Within bone, it favored osteosarcoma which makes sense given the role in osteoblast differentiation attributed to RUNX2. RUNX2 is associated with a loss of differentiation of Ewing tumors along the osteoblast differentiation trajectory (PMID: 20665663). While the cell of origin of Ewing tumor is not known, most investigators believe it comes from a cell or various cells along the mesenchymal to neural crest lineage. Therefore, the association with osteoblast progenitors is likely found in only a subset of models. Consistent with this, the DepMap analysis favored osteosarcoma. Nevertheless, the association did support the proposed mechanism even though this was less than 3% of targets in our motif analysis (see Supplementary Figs 8a and 9a). Similarly, AP1 has multiple family members expressed at different levels in different models so an analysis of one AP1 family member does not select Ewing sarcoma lines. Additionally, the cellular phenotypes associated with this cofactor include things like inflammation and matrix interactions that would not necessarily be evident on the DepMap (PMID: 39201282). E2F plays a role in cell cycle progression but again has multiple family members with different functions and challenging to pinpoint on the DepMap. Nevertheless, these cofactors proved, as originally intended, to be excellent cofactors to understand the mechanism of MMA as it pertains to binding to DNA and activation of RNAPII. I agree with this reviewer that understanding the role of these cofactors in Ewing is an important area of study. But this would be challenging to incorporate in this manuscript particularly in light of the space constraints. But thank you for this comment. I totally agree.

3. While the specific examples shown/chosen for TF binding sites do support the authors' conclusions, it would be interesting to show heatmaps for all TFs and identify subsets of sites that either retain or lose signal together. These can be teased out using a k-means clustering approach. It might be the case that all changes are concordant, but the finding that EWS-FLI1 fusion binding increases while other factors are lost suggests that different sites might be differentially impacted by MMA.

Thank you for the suggestion. Overall, we think the data shows that the main determinant of the impact of MMA on EWS::FLI1 binding is if EWS::FLI1 is acting alone (at isolated GGAA or microsatellites) or in concert with a co-factor allowing disruption of binding likely via the 3-side chain

**EWS:FLI1 GGAA microsatellites:
Repressed with silencing**

described by X-ray crystallography (PMID: 33275876). We did the K-means clustering analysis suggested by this reviewer **but as this reviewer suspected**, the changes were concordant. AP1, for example, showed clusters that over-represented the loss of binding, and at the GGAA microsatellites showed clusters that over-represented the gain in binding, but no cluster flipped the relationship (see left). We also did an analysis to enrich for differential binding to draw out differences but again only concordant changes were observed (see figs on left). So overall, the heat map proved to be suitable to demonstrate the described relationships and the clusters were a lot to digest in the setting of 5 different binding descriptions (GGAA microsat, isolated GGAA, AP1, E2F, RUNX2). Therefore, we chose to present the data as non-clustered heat maps. This was an interesting analysis and I think would be particularly useful for AP1 sites in the future in a study focused on understanding the phenotype of these cofactors as suggested in comment 2.

EWS::FLI1-AP1 tandem sites

**EWS::FLI1-AP1 tandem sites
Enriched for binding and
Kmeans**

4. Why were only TC32 and TC252 included in this study? In the EWS field there is increasing appreciation that different cell lines display different sensitivities to fusion knockdown/loss. For example, a quick perusal of DepMap shows that some EWS lines have scores of 0 for FLI1 and EWSR1 while others have very strong dependency scores. The authors should report IC50s for minimum 5 and closer to 10 cell lines across this spectrum that display different sensitivities to the EWS-FLI1 fusion. TC32 is also TP53 WT, is MMA similarly potent in TP53 mutant lines? This is an important point and the authors even mention TP53 in the text.

Thank you for this comment. We intentionally chose to focus on the TP53 wild-type cell lines and evaluated 2 of the 3 known TP53 wild-type lines. In the large landscape papers only 8% of Ewing sarcoma patients harbor TP53 mutations. A more recent update that is under review right now at JCO from the Dana Farber group shows that in reality it is even lower than 8% and TP53 mutations are found in less than 5% of Ewing sarcoma patients. Even in relapse, we have found around 10% of patients harbor TP53 mutations in a clinical study of 38 patients. Unfortunately, TP53 mutations are over-represented in the models and only 3 commonly used cell lines are TP53 wild-type TC32, TC252 and CHLA9. So the intention was to represent 95% of Ewing sarcoma because of the clinical implications of this manuscript. We agree, the contribution of TP53 mutation to resistance is an important area of study and a good follow-up direction for this drug.

In terms of differential sensitivity to EWS::FLI1 silencing, we agree that the differential sensitivity to the drug is important and have previously reported it in a total of 31 different cell lines in two different manuscripts (PMID: 21653923, PMID: 26979396). So we did not repeat this analysis in this manuscript. Importantly, we are working to understand the molecular determinants of differential

sensitivity for Ewing sarcoma lines, particularly in reference the rhabdoid tumor lines we have also reported (PMID: 33332735). But first we needed to describe how the drug perturbs EWS::FLI1 binding and RNAPII activity which is the goal of the current manuscript. It is true that there is quite a bit of heterogeneity in the EWS::FLI1 transcriptome, but the dependence is not questioned in the field. The heterogeneity is complicated and beyond the scope of this manuscript. A673 is the only cell line (of the more than 40 we have in our lab) that grow fine in the presence of silencing and these harbor a BRAFv600e mutation not found in patients. This reviewer's "quick perusal" of the DepMap is not necessarily accurate because it lumps Ewing with other bone tumors osteosarcoma and chordoma. The OS and chordoma cell lines do not express EWS::FLI1 and are therefore not dependent on EWS::FLI1 but will have EWSR1 and/or FLI1 sequences which may or may not impact viability. The variability in dependence in DepMap of Ewing lines may be due to the fact that the CRISPR guides targeting EWS::FLI1 in Ewing lines often target portions of EWSR1 or FLI1 that are lost in the fusion protein ie the 5' portion of FLI1 or the 3' portion of EWSR1.

5. One model is that MMA increases FLI1 binding, which outcompetes upstream adjacent nucleosome binding to reduce the nucleosomal barrier to anti-sense transcription. The authors should plot GRO-seq signal onto a list of bidirectional promoters (~1kb between up and downstream TSSs) to determine whether MMA promotes bidirectional transcription. This would allow the authors to comment more rigorously on anti-sense transcription than the somewhat anecdotal examples provided.

Thank you for this comment. We have done this analysis. Mithramycin does appear to promote bidirectional transcription at bidirectional promoters. This promotion appears over-represented in the LCE exposure. We did not include this analysis in this revision for space constraints. But this is an excellent suggestion and intriguing observation.

Bidirectional promoters: 1 kb between up and downstream TSS

Minor point: Line 159 Error "We next examined"

Thank you, this is deleted in this resubmission

Also no need to show triplicates in IGV browser track shots, please put a correlation matrix in the supplemental figures with all replicates contained within it. This should sufficiently convince readers about the high concordance and obviate the need to show more than 1 replicate per condition/ab.

Thank you for this comment, we included the correlation matrix as a supplementary figure S5A, S5B. Thank you for that suggestion. In my opinion, this consistency is very important, so thank you for suggesting this figure. In the resubmission, we favor showing one track for each exposure as this

reviewer suggested but we left all the tracks at the first instance in the text and in the supplementals where we quantitate fragment counts in the IGV both to reassure the reader of the consistency at more than one site and to show the source of the variance that is quantitated in the box plots. Thank you for this comment.

Reviewer #2 (Remarks to the Author):

In this study, the authors revisit the mechanism of action of mithramycin (MMA) on EWSR1::FLI1 (EF1) activity in two distinct administration modes of the drug. By integrating Cut&Tag and GRO-seq data, they characterize the molecular effects of two drug administration strategies, identifying a specific effect of the drug on EF1 oncogenic properties, as well as a non-specific effect on global RNA-Pol II processivity. Based on these findings, they propose recommendations for the future treatment of Ewing patients. The authors have pioneered the study of the mechanism of action of mithramycin as an inhibitor of EF1 for the treatment of Ewing sarcoma. The present manuscript uses various high throughput methods to further document this inhibitory effect. However, as they stand, the data fail to be fully convincing. Indeed, the authors do not show global analyses, neither for transcriptomic nor Cut&Tag data. They only point out at specific genes or loci which is not sufficiently convincing to support their general conclusions. These conclusions therefore frequently appear as overstatements. Finally, the authors convincingly demonstrate through in vivo xenograft experiments that long-term, low-dose treatment is more efficient in inducing regression of their Ewing model.

Thank you for this thoughtful review of this manuscript and favorable comments. We agree that the in vivo data is particularly strong. These comments are additionally echoed by other reviewers and in this revision we have included genome-wide analyses of both the transcriptome and Cut&Tag data. These have significantly strengthened the manuscript and are consistent with the described model. Thank you also for noticing the discrepancy in the labeling of supplemental figures.

Major Points

1- Showing how MMA impact on the expression level of genes, and how it impacts on known gene signatures for EF1 activity is essential. On Figure 1, the authors focus only on two EF1 target genes, which is not sufficient to document the global effect of MMA on EF1 targets.

We agree that this is an important analysis and we did not capture this clearly in the original version. Importantly, we previously published that mithramycin suppresses the transcriptional signature of EWS::FLI1 induced targets in two different manuscripts (PMID: 21653923, PMID: 26979396). Nevertheless, this reviewer is correct, we did not explicitly capture that we have published this in the text. In addition, those publications did not use the drug exposures that we explore so thoroughly in this paper. Therefore, we clarified the writing. In addition, we clarified a data set used in the manuscript, where we use siRNA targeting the breakpoint to identify a list of induced and suppressed targets ($\text{Log}_2\text{FC} \pm 1$, $P < 0.01$). We then used the GROseq assay to characterize the effect on expression of this gene signature by focusing on the statistics at the TES (because RNAseq is often polyA selected). We rewrote the section to more clearly state that it shows reversal of the EWS::FLI1 transcriptome and again added in the references to the previously published data sets showing reversal of independently generated gene signatures. Thank you very much. We agree that this is essential to make this point clearly, which we had not done in the first version.

2- On Figure 1, EF1 levels fluctuate significantly, with decrease at early time points then increase at later time points. Is this fluctuation linked to a transcriptional regulation? Does LCE or HCE treatment regulate EF1 expression at the RNA level?

Great observation! Yes it does. The mRNA expression of EWS::FLI1 decreases but protein levels remain relatively unchanged to increased. We have worked on sorting this out for many years and it is challenging. We were very careful to analyze time points that had similar levels of EWS::FLI1 expression the best we could given the biology (see Figs 1A/C). But this reviewer is correct it is likely that increased expression enhances the increased occupancy of EWS::FLI1 with LCE (as reviewer 4 also points out). So, we clarified the language around that statement and added a statement that said the increased occupancy likely stems from increased expression and stabilization. The stabilization is supported by fluorescence anisotropy data from another group. So the enhanced occupancy is at least partially due to stabilization in binding and likely is a combination of both stabilization and increased expression. At the end of the day, it doesn't change the effect, more EWS::FLI1 at these sites reverses activity which we show by CUT, Tag and GRO and is an important observation for this study. But, we did not accurately describe this combination of effects in the original version. Thank you for the suggestion.

3- MMA Effect on Chromatin: The chromatin-level effects of MMA are presented only for selected target genes. However, a broader, unbiased analysis is needed to support the conclusions. Showing a global heat map of the variation of EF1 binding sites upon MMA treatment is critical. The authors describe sites with increased binding, other with decreased binding. How does this relate globally with transcription? How does this relate with the different types of EF1 binding sites, single GGAA or microsatellites?

This comment was echoed by other reviewers and we agree was a miss in the first presentation of the data. We added these analyses and now show profile plots of average binding, heat maps of CUT&Tag occupancy and associated profile plots for all 5 conditions (EWS::FLI1 microsatellites, isolated GGAA, AP1, E2F3 and RUNX2 cofactors) for both induced and suppressed EWS::FLI1 target genes. These are now Figs. 4b, 5c, 5d, 6a, 6b, Supplementary Figs, S10a, S10b, S11a, S11b, S14a and S14b. As stated elsewhere, in the previous version, we focused on the subsets of targets and presented 100% of the published subsets of targets. So now with the inclusion of this new data, we show the effects at established targets AND genome-wide. This has greatly strengthened the manuscript. Thank you. The conclusions remain the same, there is overall stabilization in EWS::FLI1 binding by MMA that favored LCE at microsatellites and isolated GGAA and a competitive effect at co-factor sites that is consistent but a bit more subtle based on the numbers of targets. We show both overall effects and the effects at the various co-factors (which also are consistent with published XRAY crystallography data). Importantly, by combining these analyses with GROseq, we see clear demonstration of the "goldilocks" effect that just the right gene dosage of EWS::FLI1 is needed at downstream targets for productive transcription. Finally, we describe the global effects first in Fig. 2 and then focus on the EWS::FLI1 effects in the remaining figures.

4- Peak Comparison (Figure 2): The authors compare EF1 peaks using fragment counts within selected peaks. However, the Methods section mentions the use of DiffBind for differential peak identification, yet no global differential peak analysis is presented in the main text. Indication on how peak counts were extracted and normalized between samples and

treatments, and whether the DiffBind package was applied in this analysis? line 252, the author concluded: “Overall, MMA exposure favored enrichment of EWS::FLI1 binding at GGAA microsatellites vs. competitive inhibition”. Such a conclusion would require a broader analysis of EF1-regulated and bound genes comparing single GGAA ETS vs GGAA repeat sites on a global scale.

Again, agree, this should have been included. We thank all the reviewers for this comment. We have now included this data and indeed show enrichment of EWS::FLI1 at GGAA microsatellites and isolated GGAA response elements favoring LCE. Some of the methods were not totally clear in the first version so, we updated them. Here is the methods section:

Peaks were called using MACS2 with a q-value threshold of 0.05, then Fraction of Reads in Peaks (FRiP) was calculated using bedtools intersect to quantify signal enrichment. DiffBind v 3.16.0 was used to identify differentially bound peaks (padj < 0.05) in LCE,HCE,and LCE vs HCE. (HCE vs Media n=10475, LCE vs Media n=35119, HCE vs LCE n= 15952) and Homer was used to find differentially bound motifs.

5- Motif Analysis (Figure 3): It is unclear how the motif enrichment analysis was performed. There is no mention of the HOMER motif analysis in the Methods section. Is this a differential analysis between LCE or HCE and solvent conditions? What is the normalization process? To

Rank	Motif	P-value	log P-pvalue	% of Targets	% of Background	STD(Bg STD)
1		1e-1754	-4.040e+03	12.17%	0.63%	59.4bp (91.5bp)
2		1e-1244	-2.865e+03	37.71%	13.67%	49.8bp (68.2bp)
3		1e-866	-1.995e+03	12.32%	2.03%	49.6bp (60.6bp)
4		1e-125	-2.882e+02	21.26%	14.27%	55.3bp (65.3bp)
5		1e-110	-2.539e+02	5.02%	2.05%	57.0bp (74.3bp)
6		1e-94	-2.170e+02	7.09%	3.65%	58.3bp (83.0bp)
7		1e-87	-2.018e+02	30.19%	23.32%	55.7bp (62.7bp)
8		1e-84	-1.944e+02	1.69%	0.39%	49.3bp (59.2bp)
9		1e-83	-1.922e+02	24.50%	18.32%	55.8bp (65.8bp)
10		1e-76	-1.754e+02	17.42%	12.34%	54.2bp (62.0bp)

conclude that EF1-bound peaks in LCE treatment are more enriched for classical EF1 motifs than in HCE treatment, a differential motif enrichment analysis should be performed, comparing HCE vs. LCE or vice versa using HOMER or another motif enrichment tool.

Thank you for this suggestion. We did the Homer motif analysis for LCE vs. HCE and indeed, the top motif identified was the GGAA microsatellites suggesting a more specific effect at these sites with LCE which we added as Supplementary Fig. 9c.

Homer motif analysis of LCE vs HCE. The top enriched motif is the GGAA microsatellite consistent with increased specificity of LCE relative to HCE.

6- Fig. 3c, d, e, f. The authors analyze a very limited number of genes and draw conclusions that the LCE schedule treatment reduces EF1 binding at heterotopic EF1/RUNX, EF1/AP1 or EF1/E2F3 complexes-specific binding targets. Again, proposing such a conclusion require 1- a more global, unbiased approach, not limited to what appear as opportunistic loci and 2- an experimental confirmation using ChIPseq or Cut&Tag. Line 233 states: “these data suggest

that MMA disrupts binding of EWS::FLI1 containing complexes to alter transcription” . With the presented data, this is clearly an overstatement.

Yes, we agree this reviewer is correct. Although we analyzed 100% of the limited number of described targets for RUNX, AP1, and E2Fs, it is difficult to make a global conclusion based on these very limited number of loci. The field has not quite started to consider thinking about these binding events separately. Our data is consistent with published XRAY data but the overall impact on the transcriptome is limited. Nevertheless, in response to this and the other reviewers, we have addressed this in the revision. We include now genome-wide analyses which support the model. These analyses also support that there are a limited number of sites, particularly for RUNX2. As stated above, we added these analyses and now show profile plots of average binding, heat maps of CUT&Tag occupancy and associated profile plots for all 5 conditions (EWS::FLI1 microsatellites, isolated GGAAAs, AP1, E2F3 and RUNX2 cofactors) for both induced and suppressed EWS::FLI1 target genes. These are now figs. 4b, 5c, 5d, 6a, 6b, Supplementary Figs, S10a, S10b, S11a, S11b, S14a and S14b.

7- Line 268 – 271. The authors conclude that MMA treatment increases EF1 binding at EF1 and ETV6 targets and “overcame” ETV6 suppression. However, there is no data to support this hypothesis (e.g., ETV6 + FLI1 tracks, GRO-seq results).

I think this reviewer might have missed this in the first version. There are a limited number of described ETV6 sites which makes a genome wide analysis challenging. So, we again show all of the described sites from the published manuscripts. This is now supplementary figure S17 showing the intersect of EWS::FLI1 binding and GROseq at all of the described ETV6 sites. We agree, that this was easy to miss in the first version and poorly labeled. So in this version, I changed the labels of the figure to make them more clear and more consistent with the rest of the manuscript. Notably, this favors enrichment, and the anecdotal description of these sites. But with so few sites is not perfect.

8- GRO-seq Analysis: The authors analyze GRO-seq, but there is no mention of quality controls, replicates, or methods used for comparative analyses. The GROseq signal seems to increase bidirectionally at promoters under mithramycin conditions (Fig5a, b, c, d). Sense-oriented GROseq data should be presented to define whether this is a global non-specific increase of transcription (or of background) or whether this is related to transcription of genes. In brief, specificity of the GROseq signal should be documented. On Fig 4d, the map should extend a few kb upstream of the TSS and downstream of the TES to show the difference between gene transcription and background level on intergenic regions. In the absence of such QC, the conclusions on mithramycin effects cannot be interpreted. They should be based on robust statistical analysis of high-quality data.

Thank you for this comment. The methods in the first version were not clear and so are updated here in this revision to include the methods of QC. All conditions were completed with 3 biological replicates and that is also included. But yes, the data was excellent. It was robust and did show excellent signal to noise. In an attempt to make the figures more readable, we cropped a few of them too tight which could make the control data look like background. I did not think of this because I saw all of the data. But we have now added expanded views of 4f, S3 and S15. Also please see the EZH2 locus below in response to reviewer 3. It also looks great. I am happy to provide several more loci if this reviewer requests. The data is excellent. The bioinformatic tools are not as developed for

this data, though. We did use the NRSA package which many groups use (although did not include a lot of that data). There is not a readily evident non-specific increase in background transcription in the genome. I am not sure exactly how you show this, though. There is a trend towards increased transcription at EWS::FLI1 binding sites where binding is enriched. We highlight this as an anecdotal observation because the number of well-established response elements, that shows enrichment, and shows this opposite strand initiation is relatively low and so these analyses were not fulfilling. The sense oriented data was also not super fulfilling again based on the number of expected loci relative to the genome. The drug does promote bidirectional transcription at bidirectional promoters (see response to reviewer 1, above). In addition, NRSA showed, there is no difference in counts in the gene body per the NRSA analysis for control and treated overall. I do think this effect could be captured by other groups developing EWS::FLI1 stabilizers (E3 ligase inhibitors) where the enrichment of EWS::FLI1 is more robust. But for this manuscript, while these are important questions, I would argue beyond the main point of the manuscript, linking schedule to effects on targets.

9- Figure 6: it is unclear how EF1-repressed and EF1-induced genes were defined. The authors conclude that HCE has a greater impact on RNAPII processivity as compared to LCE. However, this difference appears quite subtle, and the quantitative analyses in Figures 6b, c, d, and the Supplementary data seem insufficient to support this conclusion. Additionally, the violin plots for TSS and TES are not entirely readable. The same concern applies to Figure 6e, where LCE vs. HCE differences appear subtle. Could the authors provide profile plots to better illustrate these differences?

Thank you for these comments. In this revision, we have clarified the experiment used to identify the induced and repressed targets were defined. We used siRNA to silence EWS::FLI1 using a highly specific siRNA targeting the fusion protein breakpoint that we, and others in the field, have used and published. We defined the induced and repressed targets as differentially expressed by a $\text{LOG}_2\text{FC} \pm 1$ and a P-Value < 0.01 . This resulted in 1915 EWS::FLI1 induced targets and 2661 EWS::FLI1 repressed targets that met this criteria. We agree the violin plots (Now Figs. 3b and e) are hard to see but the important thing is the statistics which are highly significant in Fig. 3b for both induced and repressed targets but not significant for repressed targets with HCE in Fig. 3e. We include expanded figure violin plots as supplementary Figs. S1 and S2. We thank the reviewer for the idea of including profile plots (now Figs. 3C and 3D). These clearly show the differences in RNAPII processivity between LCE and HCE and support the difficult to read violin plots. We also added impressive profile plots (now Fig. 3g) that shows the difference in processivity for LCE vs. HCE as a function of gene size that clearly delineates the differences between LCE and HCE. Thank you for this comment, this reviewer is correct, the profile plots were the best way to represent this data, and it is not subtle.

10- On Figure S1 the EF1 binding peaks are located outside of the GGAA repeat which questions the specificity of the peak.

This is an impressive observation. This locus is, by far, the most well established functional EWS::FLI1 binding site in the genome. It is also a site of phase-transitioned EWS::FLI1. It is not clear why there is “drop-out” of signal over the microsatellite. We believe it is because of the phase transitioned EWS::FLI1 which makes it hard for the Tn5-transposase to get through the phase transitioned EWS::FLI1 to cut the DNA in the middle of the GGAA microsatellite. To try to address this, we did Cut&Run. But the Cut&Run was worse with even more drop-out over the GGAA microsatellite. Importantly, CHIPseq is worse still because it is a repetitive sequence and hard to PCR

amplify across the microsatellite. This is one of the many challenges of studying this fusion protein. The consistency of the replicates is impressive and makes the data highly believable particularly in light of the amount of data supporting this binding site as the EWS::FLI1 response element. Additionally, although not included in this manuscript, siRNA silencing of EWS::FLI1 makes this peak go away. So it is real, but the drop out is an assay artifact and can't be avoided using another assay in our experience. These are the data. I will say, I have waited almost 15 years for this kind of high quality binding data. We tried for many years to ChIPseq EWS::FLI1 and although we could clearly ChIP EWS::FLI1 at specific loci, the sequencing failed several times.

Other Points

- Line 262: “Notably, even in the absence of opposite strand initiation, prominent EWS::FLI1 peaks were frequently associated with enrichment of nascent RNA transcripts at the actual sites of EWS::FLI1 binding at the microsatellites in addition to the transcriptional start sites.” Are these data provided? If not, could the authors include this analysis?

This was an interesting observation. But, not properly supported. We did additional analyses for this revision but they were not terribly convincing. So we removed it from the manuscript.

- Figure 2, b and c, scales are missing

We did not show scales for these. But group scaled all the peaks to the highest peak in control. This detail is added in the legend.

- Line 175, The statement “We used a FLI1 antibody bound to Tn5” could be misinterpreted, as in the Cut&Tag procedure, a secondary antibody is typically used to direct proteinA_Tn5 to the chromatin. Moreover, could the authors clarify the exact references for the antibodies used, as Active Motif, Cell Signaling, and Abcam offer multiple references for H3K27ac, IgG, or FLI1 antibodies?

Thank you, yes this language was sloppy and was corrected. As far as the references for the antibodies, we actually prepared a detailed list with all of the RRIDs for all reagents but somehow did not include it in the original submission. So we included it in this revision.

- Line 180, The signal detected for H3K27ac cannot be described as prominent in Figure 2d and is even absent in Figure S1

Thank you. Again, this was sloppy language and was rephrased.

- Line 223 “Fig.3 e and S3b-f” correction: S3c-f ?

Thank you. Now combined with other sites and now Figs. 6c-f and Supplementary Figs. 12a-f

- Line 248 “(Figs. 4e, 4f, and 6a-d)” correction S6a-d ?

S6a-d has been eliminated in the current version given the genome wide analyses. Figs 4e, f are now supplementary Figs. S7a, S7b

- Line 261, The genes “AKAP7, GRK5, PRKCB, GSTM4, and GLG1 (Figs. S7a-e)” are mentioned, but these genes do not appear in the correct order in Figure S7, and GLG1 is not illustrated, whereas CACNB2 is. Could the authors correct this discrepancy and ensure the figure matches the mentioned genes?

Thank you for pointing this out. The new figure is now supplementary figs S16a-e.

- Figure S9 plots on TSS and TES are impossible to read / interpret

I am sorry we could not fix these plots. We double checked the source files and notably these are now supplementary Figs S1 and S2. The plots are appropriately sized, but the differences are a bit challenging to see. These are violin plots which intrinsically are hard to see the differences when the counts are similar such as at the TES. Therefore, as suggested, we added profile plots as Figs. 3c and 3d to make the point another way which is much better. In addition, the important thing is the statistic at the TES for the repressed targets; either significant for LCE or not for HCE. We added clarity around the text for this point.

- Figure 6b, Line 314, mislabeling of plot titles?

We changed the text on this section to focus on the TES of induced targets only. But yes they were mislabeled, our bio-informatic scientist always refers to repressed target as induced because they are induced with EWS::FLI1 silencing so this leads to confusion. We corrected it and was clear in the text that induced targets means induced by the fusion protein and suppressed with silencing.

9.- Figure 6f, Could the authors include tracks showing gene localization and EF1 binding to provide additional context and support for the analysis?

New tracks now show both the Cut and Tag and GROseq data on one graph with additional flanking genomic space to show signal to noise as Supplementary Fig S3.

Reviewer #3 (Remarks to the Author):

The manuscript by Kaufman et al. describes efforts to understand whether continuous 24h dosing of the EWS-FLI1 inhibitor, mithramycin (MMA) (with this schedule in a current clinical trial), may impact the efficacy of MMA compared to once daily bolus for 7 days, with 21 days off, administration. The rationale for the altered schedule is based on an older protocol that induced responses to Ewing Sarcoma in a patient, versus the hepatotoxicity in a recent therapeutic trial that resulted in dose limitations that precluded achieving concentrations of MMA that was able to inhibit EWS-FLI1 (estimated to be about 3 times too low).

To support this new protocol, the authors investigate how MMA, a known EWS-FLI1 competitor to binding to its core recognition sites, GGA(A/T) DNA response elements, impacts EWS-FLI1 function at different treatment schedules. Using CUT&Tag with Global Run-On-Sequencing (CUT, TAG and GRO), the authors demonstrate that continuous administration at low concentration is more efficacious to inhibit the EWS-FLI transcriptome, leading to improved in vivo responses of MMA (at least in one ewing mouse model).

Overall, this is an interesting study that supports the rationale for an ongoing clinical trial of lower continuous MMA in, at least the Ewing Sarcoma cohort of the trial, and provides interesting insights into how MMA inhibits EWS-FLI1, and, as an extension, a window into how EWS-FLI1 functions in Ewing Sarcoma. Importantly, it has been traditionally difficult to study the function of EWS-FLI1, and the use of CUT, TAG and GRO here is innovative, robust, and supportive of the investigators' findings. The primary need is to address some of the in vivo concerns listed below, however I have included additional comments that would also be helpful to address.

Thank you for the thoughtful and thorough review. We agree that the CUT, Tag and GRO assay will be impactful for the field and for transcription factor drug development. CUT&Tag by itself, for the Ewing field has been transformative already. We tried for many years to try to generate high quality data using ChIP seq and were unsuccessful.

Major comments

Do these models die following HCE or LCE MMA (as others have suggested at different doses/schedules (PMID: 36913323) and if so, do they die at different rates? If they do die does that process include caspase activation (as suggested by PMID: 36913323)? This is relevant as active caspases can cleave FLI1 (and perhaps EWS-FLI1).

This is an interesting point. I like this idea a lot. At the concentrations and time of drug exposure in this study used for all of our genomic data, there is limited caspase activation above background in vitro which is now shown as a new Fig, Fig. 2e. The difference is particularly striking in comparison to 5 micromol/L etoposide and 25 nmol/L lurbinectedin. We added a figure showing limited caspase activation with these drug exposures generated using a caged cleaved Caspase 3,7 fluorophore and automated counting on our CellCyte (Fig. 2e).

But to the question of if caspases could be activated to cleave EWS::FLI1 to contribute to suppression, the data presented here does not support that theory. There does not appear to be a shift down in EWS::FLI1 mobility on western blot in Figs 1A-D as one would expect if EWS::FLI1 were cleaved by caspases. This is self-fulfilling, though, because we do assays in our lab at time points prior to cell death to limit confounding effects of apoptosis. I do think this is a super interesting question, though for other agents and therapies. But I was curious about this question and dug through all of our data to see if we could find evidence of cleavage of EWS::FLI1 with any drug exposure including other compounds that we study in the lab, like lurbinectedin and trabectedin, because this would be interesting and easy to exploit therapeutically. Unfortunately, I did not find evidence of EWS::FLI1 cleavage with any drug treatment but I would bet there are agents that could achieve this. One last point, there is an interesting line of investigation looking at MMA + radiation in Ewing which I really like. I wonder if this effect contributes to that cooperativity because radiation would potentiate the apoptosis?! And may cause this effect?! The tumor was described to be radiosensitive by James Ewing over 100 years ago.

For me, Fig 1 legend is confusing to follow. "a" is written twice, and, overall, the changes in concentrations and time, and the way that is written, makes it confusing. In addition, the time point is not indicated in the figure legend for A and B. I would suggest re-writing this figure

legend and adding more information in the figure in terms of time and concentrations, so the reader can follow this a bit better.

Thank you. This was poorly worded and has been rewritten.

Is 72h not included in Fig. 1D for a particular reason? It would seem for comparison's sake it should be included.

Yes, 72 hours was not included because the kinetics of the effects were slightly different and occurred earlier in TC252 cells than in TC32 (see fig 1C vs. 1D 48 hours) so the cells were pretty sick at 72 hours and therefore that time point was not shown.

It looks to me that EWS-FLI1 is clearly shifting upwards at the significant time points or concentrations in blots A-D (It is harder to see in D but all the other blots in this particular panel look like they are trending downwards left to right, while EWS-FLI1 has the opposite pattern). Is there a modification of EWS-FLI1 that MMA is causing, and could this modification in itself cause some of the altered function of EWS-FLI1?

Thank you for this comment. I believe there is a PTM and this may contribute to the change in binding but we have tried for years to figure this out without success. We believe it stabilizes EWS::FLI1 expression consistent with reviewer 1's observation and questions about EWS::FLI1 mRNA levels. It is a tougher question to answer than one would think due to properties of EWS::FLI1. If there is a modification, then it may contribute to the change in binding in a manner analogous to the slight increase in expression may contribute to the increased binding of EWS::FLI1. But for this paper, we clearly show occupancy does indeed increase at GGAA microsatellites and isolated GGAA sequences, particularly with the new data but disrupted by complexes. This stabilization of GGAA repeats and disruption in binding is supported by independent fluorescent anisotropy data and X-ray crystallography data, respectively, obtained with recombinant FLI1 that does not have PTMs. Our data supports these findings. But there may be other factors in cells that further contribute such as PTMs. So, I think this reviewer is correct, there is a likely a PTM and that PTM likely contributes to stability in expression and may contribute to binding. But to establish these relationships would require a dedicated manuscript which we continue to actively pursue but have not been successful. At the end of the day, it does not change the conclusion that there is enrichment in binding at these microsatellites. We do soften the language around this in this revision in response to this reviewers and another reviewer's comments and say that it is likely due to increased expression and stabilization in binding. But it is super cool that our genomic data supports in vitro X-ray crystallography and fluorescent anisotropy data.

The goldilocks hypothesis the authors invoke is interesting but could use more supportive data. It would be interesting, for instance, to know how moving from 20nm to 100nm alters binding of some of the disparate genes (low dose versus high dose that have enhanced versus inhibited EWS-FLI1 binding). Of a similar vein, low dose (which is shown in Fig. 1C and 1D) over time versus high dose over time and how this impacts EZH2, for instance. What does EZH2 look like at the transcript level, as in Fig. 1F and 1G?

I am sorry, I am not sure I totally understand what the reviewer is asking here. Does he want a time course of CUT&Tag? This would be challenging and expensive and I am not sure what we are asking. Or is he asking how things change at EZH2? It seems like he wants more supportive data for the goldilocks hypothesis. So in this revision, we rearranged the paper to put increased and decreased binding at *NR0B1* and *RCOR1* right next to each other in the text Figs. 4c-f, supplementary figs. 6a and 6b. In addition, we now provide genome wide analyses which support

both increased and decreased binding poisoning transcription Figs. 5c, 5d, 6b. In terms of EZH2, pictured here is the CUT, Tag and GRO of the EZH2 locus, it show less pronounced enrichment with HCE than LCE but enrichment with both exposures that translates to suppression of transcription. Additionally, we

tried to identify disparate binding in our genome wide-analysis and used the Kmeans approach suggested by the other reviewers but overall the changes tend to be concordant (see graphs in comments from reviewer #1 above). I hope this speaks to the question but would be happy to further clarify.

It would seem NROB1 should be blotted for in fig. 2A.

Thank you for this comment. In this revision, we did not repeat this experiment because this is the identical concentration and time of mithramycin as shown in Figs 1a, 1c. Those data have all been repeated with at least 3 independent experiments (and actually many more because we trained a student with this assay) by three lab members. So there is no reason to believe the total effect on NR0B1 would be any different if probed in this blot. Additionally, NR0B1 is an orphan nuclear receptor and so one would expect it is most cytoplasmic. I am happy to repeat this experiment and add in NR0B1 if there is a specific question that I am not seeing that we are missing. But we are just using NR0B1 as a marker of EWS::FLI1 activity and not making any claims about it as a disease or EWS::FLI1 modulator so I am not sure what it would add to the figure. So I would ask the reviewer to consider the need for this repeat in light of all the new data and experiments in this revision.

What is being shown in S11? Is this vehicle-treated tissue or MMA-treated tissue

Yes, sorry this was not clear in the first submission. It is hard to write a figure legend as confusing as I did in the first submission. I clarified in this version that it is HCE MMA treated tissue. Sorry about that.

Fig. 7—some statistics would be helpful here, but more critically, as this is the proof that the continuous low dose schedule is superior, this experiment should be repeated in at least one (and preferably two) more ewing in vivo models

This comment is fair but has been a challenge in the time constraints of this resubmission. The justification for not doing the experiment in another line in the first version of the manuscript is that we showed the differences with mithramycin and an analog at three different doses in three different experiments. The challenge with addressing this comment is that we moved our lab to a new institution over the past year and at the time of submission had not yet established the technically advanced capabilities for surgery, anesthesia, sterile pump preparation etc. to complete this experiment. So when we got this comment, we scrambled to re-establish these capabilities. We have now been able to get IACUC approval, obtain a new anesthesia machine, buy new animals and initiate a pilot experiment with TC252 to confirm that the effect is consistent. Additionally, we had challenges around the source of mithramycin. Nevertheless, we did complete a pilot experiment of a TC252 xenograft. This cell line was more sensitive to the effects of MMA in vitro and accordingly was more sensitive in vivo and demonstrated stunning tumor regressions with continuous infusion of rather large tumors. As you can see in the data below, continuous infusion MMA at LCE caused striking tumor regressions even with the massive tumors in the biggest two animals. These animals got sick because the new MMA was a little more potent from this new supplier and so we do not feel comfortable publishing the below pilot experiment although it is totally consistent showing striking responses. If this reviewer feels we should complete a treatment cohort, we are happy to and will include it. This will require another dose optimization pilot given the new source of mithramycin and another treatment cohort and so will take several more months. But we just haven't had time to complete this within the window of resubmission. We would prefer to keep this out and add it to the next manuscript that we are working on focused on the spectrum of activity across different Ewing and non-Ewing models as one of the reviewers above alludes to. But again, we could do a treatment cohort now that we have re-established these capabilities. The effect is and will be the same.

Pilot study of mice bearing TC252 xenografts, treated with 4.2 mg/kg or 0.6 mg/kg/day x 7 days continuous infusion mithramycin. The grey box indicates the days of treatment.

As the investigators are aware, dose-limiting hepatotoxicity of MMA led to dose reductions that likely were not successful at inhibiting EWS-FLI1. Thus, renal toxicity studies would seem to be helpful comparing the two doses/schedules. Additionally, the authors have previously demonstrated severe liver toxicity by MMA, although this largely did not manifest in the trials. This would be appropriate to evaluate in the different schedules.

In this revision, to respond this comment, we have now added clinical chemistry data from serum from the animals treated in the original submission (supplementary Fig. S20). The data show limited hepatic and no renal toxicity. This is an important point, so we appreciate this comment as it has strengthened the conclusions.

Minor comments

“We next examined how” should be removed from line 159

Thank you. This has been deleted.

Its unclear to me how fli1-bound tagmentase leads to only specific binding to FLI1 sites. There should be a reference when this concept is introduced (just parenthetical point as is) and an explanation may be plausible in the text.

This is known and widely used. FLI1 is not expressed except in a few cell lines. It has been suggested that there is a FLI1-EWSR1 fusion but this is not widely accepted and even this would not have the 3' FLI1 antigen recognized by the antibody. I added references in the text.

Fig S3 title should seem to be “EWS-FLI1 containing complexes to DNA”

This has been changed.

Fig S9 should seem to read in the figure “repressed after silencing EWS-FLI1. In addition, the X axis labels blend in and is not clear (the font is too large)

This is now S1 and S2. The labels have been clarified and changed to LCE and HCE which is more readable and more consistent. Fig S10 same concern with the Y Axis (the font is too large)

Fig S12 statistics would be helpful comparing relevant measurements

Statistics have now been included.

The added comment about EWS-FLI1 not being cloned seems out of place in line 110-111.

Thank you. Agreed that this is awkward wording. I changed cloned to discovered. The point was simply that mithramycin was tested in patients long before EWS::FLI1 was even known to exist much less known to be the oncogenic driver of the disease.

-Anthony Faber

Thank you for signing the review. It helps with the interpretation of the comments to know where the reviewer is coming from. I am going to do this more as well.

Reviewer #4 (Remarks to the Author):

The manuscript by Kaufman et al. characterizes the effects of two mithramycin A treatment regimes in Ewing sarcoma from the genomic level to mouse models. The results are interesting, particularly the dramatic relocalization of EWS::FLI1 for the LCE protocol and the implications for treatment. The seq data is presented in an unusual way and some of the interpretations seem a bit overstated or irrelevant to the data the study has produced. Given the strong interest of the main finding, that MMA can have alternate effects on EWS::FLI1 depending genome location, a revision of this manuscript would be good to see. It's also recommended the seq files be provided for expediency in clearing up some of the ambiguities mentioned below.

Thank you for the review. We agree and your comments are echoed in other reviewers. So we included new analyses that speak to the genome wide effects with more traditional CUT&Tag analyses and respond to all of your comments below. The Seq files have been uploaded to GEO.

Major:

1. Figure 2a – The statement in the legend is that there is a reduction in binding for HCE and increase for LCE. There is some ambiguity in this interpretation because the concentration of EWS-FLI1 differs dramatically in the two samples. It would seem safer to say only that the LCE has more EWS-FLI1 associated with the chromatin fraction.

Thank you for this comment. This is in agreement with reviewer 1's comments. We updated the legend to say enrichment of EWS::FLI1 in the chromatin fraction with LCE as suggested. We changed the text to say that there is increased occupancy likely secondary to both an increase in expression and stabilization (which is supported by independent fluorescent anisotropy data.) In addition, we changed HCE accordingly in the legend. We also added a volcano plot (Fig. 4b) to better show the change in occupancy as a function of significance in a more quantitative manner.

2. HCE is reported to increase EWSFLI1 binding at RUNX2 sites. While the other sites are described as uninterpretable, can the data for RUNX2 binding be shown?

In the previous version, we reported decreased binding of EWS::FLI1 at RUNX2 sites, CNN1 and SPP1 with HCE (Figs. 6d, Supplementary Fig S12a). In the current version, we have included that data in the resubmission but added genome wide analyses of the data. Here we show reduced binding with HCE in particular for EWS::FLI1 induced targets (Fig S11a, 11b). The data for repressed targets did not look good presumably because there are so few RUNX2 cofactor sites. I do not believe this interaction with RUNX2 to be that important for this particular model. There are a limited number of publications on this association. However, for the drug, another group has published X-ray crystallography data of a FLI1-RUNX2 mithramycin structure. So we needed to evaluate this heterotypic complex in our analysis.

3. The paper shows examples but lacks some context for the reader. To move the manuscript from becoming anecdotal in its methodology, it might be helpful to know how many EWS-FLI1 peaks in the M sample disappear or diminish in the LCE or HCE, how many increase, and how many novel peaks appear in the LCE.

Thank you for this comment, yes, this is the major change in this revision. In the previous version, we analyzed all of the well-established target genes for these cofactors and many of the most well-established microsatellites. We have retained that analysis in this revision but have added numerous genome-wide analyses and now show profile plots of average binding, heat maps of CUT&Tag occupancy, and associated profile plots for all 5 conditions (EWS::FLI1 microsatellites, isolated GGAA, AP1, E2F3 and RUNX2 cofactors) for both induced and suppressed EWS::FLI1 target genes. These are now figs. 4b, 5c, 5d, 6a, 6b, Supplementary Figs, S10a, S10b, S11a, S11b, S14a and S14b. Probably the best figure that speaks to context is the new volcano plot which is Fig. 4b.

4. The negative controls appear to be inconsistent. For the GRO-seq data, they are sometimes called “solvent”, Figure 2b-c, and for the rest of the paper called “M”, including Figure 4d and Supp Figs S5 – S8. Figure S6b is labeled both “M” and “solvent”. Also, there appears to be separate “M” experiments for the LCE and HCE based on the data in 2e-f, 4b-c, and 4e-f, but only one M track is shown in figures 4, 6 and Supp Figures S5-8. If LCE and HCE have their own “M” or “solvent” controls, it should be shown that way, not lumped with one control and the peak magnitudes omitted. It may be better for the seq data to just be made available to reviewers to avoid so much back and forth to understand what the selected figures show.

The control for the GROseq is solvent. But for the purposes of the data, the solvent is unlikely to interfere with any assay. Mithramycin is a natural product and highly soluble in water. We standardly make up a 1 mg/mL solution in PBS and so the solvent in any analysis at the higher 100 nM concentration is 0.01% (0.0001 v/v) PBS/Media. This was not stated anywhere in the manuscript but I want the reviewer to know this to diminish concerns that there is any interference with any assay by the solvent. The labeling is inconsistent and we fixed most of the instances of this. I did just notice a few inconsistencies, which we will fix for the final version if this paper is accepted for publication. But for the purposes of the data. The solvent is unlikely to interfere with any assay.

5. Figure 4D, the GRO-seq tracks have no scale in the y-axis, they’re different heights, and the “Media” track looks a lot like noise. More of the flanking genomic regions need to be shown to help understand what this genome browser view is showing. Figure 6f and the supplement figures S5 to S8 also have no y-axis scale.

The tools for integrating this data are not well developed. We show the IGV tracks for much of the CUT&Tag data but that does not look great. Therefore, we integrated the data from deeptools with other data in R. However, while the figures looked great, it does not export with track scales. So we could pull the scales and manually insert them, but instead in this revision, we just specified that they were group scaled to the highest peak in the control.

6. Does the mention of significance for Fig 4e & f refer to $p < 0.0001$ for either LCE or HCE? The same peaks are shown in both but for 4e, CACNB2 and FEZF1 hardly seem significantly different.

Thank you for the careful review of the data. Yes, these individual peaks do not look markedly different. But this is an ANOVA for the means for all the peaks to speak to the question of if the drug impacts binding of EWS::FLI1 to microsatellite genes in a panel of well-established targets. We also added genome wide impact on binding at GGAA microsatellites and call outs of microsatellites in the

volcano plot which also speak to the statistical significance of binding at the microsatellite genes (Fig 4b). It is clear looking at all of these analyses, that LCE impacts binding to the Microsats.

7. Since EWS::FLI1 dismissal or binding relates to single sites or microsatellites, it would be helpful to better characterize the microsatellites. This includes providing details such as length and number of GGAA repeats for the examples shown in Figure 4 and the relevant supplemental figures.

We added the relevant details as this reviewer suggested. Microsatellites were defined as greater than 6 GGAA repeats which has been shown by the Lessnick group to be the minimal length required to function as an EWS::FLI1 response element. We also did an analysis of length of GGAA as a function of binding by each schedule. The data is interesting and shown to the left. There does appear to be a trend towards longer GGAA's being bound by EWS::FLI1 and also differentially bound by drug. But, in my opinion, I am not sure the groups are distinct enough to include. I do think it is real

and the stats suggest significant. But genomic data like this can be controversial. Happy to include it if the reviewer thinks it adds to the manuscript.

8. The relationship between EWS::FLI1 and antisense transcription described on page 15 is difficult to follow. The genome browser views in Supplemental Figures 7 and 8 appear to show typical divergent transcription, not eRNA-like transcripts, especially since the transcripts diverge from annotated TSSs in most cases, or close enough to suggest alternative start sites. It's also unclear how phase separation can be inferred from GRO-seq data. Unless the authors provide clearer figures comparing representative examples of each transcriptional state and quantify antisense transcription levels, this argument feels out of place with the rest of the study.

Thank you. We agree that this argument was not well explored. We simply don't have the space to address it properly and sort of expected this comment. We removed the phase transitioned statement because that was clearly overstated. But this is the first publication of GROseq data in Ewing to show this and so I do think there is value capturing this observation even anecdotally. Reviewer 1 noticed it as well. In this revision, I added the word "anecdotally" so as not to overstate the findings. But it is really an interesting observation. We did look at bidirectional promoters see comments to reviewer 1 but there is not a place to add this concisely, either, in my opinion. Happy to take this out if this reviewer insists but as other groups are working on developing EWS::FLI1 stabilizers, this will be cool

to know about and look for when there is an approach to more robustly enrich EWS::FLI1 genome-wide.

9. The nonspecific increase in transcription is interesting. Can this be verified by a non-seq method, like an EU incorporation assay? It may contradict earlier literature reporting high levels of transcription in ES (PMID: 29513652) and also seems an interesting counterpoint to the possibility that transcriptional stress might contribute to ES survival.

Thank you for this comment. This is an interesting point. In response to this comment, we developed the EU incorporation assay. We used the conditions from the above publication which were slightly different than the manufacturers recommendations. The data, shown below, is consistent and shows increased transcription with HCE but not LCE. However, in its current form, I am not sure the assay is suitable for publication although notably we have repeated the result 3 times with similar results. We prefer, high quality, highly reproducible data with non-subtle differences among groups. With this assay, there are a lot of challenges that leads to a lot of noise. Presumably, this is why the assay is minimized in the paper the author alludes to above. The assay requires the cells to be fixed which is not ideal. There is a lot of background and the signal to noise is not great. These may be solvable problems and we could try not fixing the cells, changing the fluorophore, and perhaps using a biotin-streptavidin interaction to cut down on the noise. As of now, we are including it here and happy to add it, if this reviewer feels inclined. The GROseq is essentially the same thing with BrdU instead of EU and is in non-fixed cells so probably better. We could also try the GROseq conditions for the EU. So below is the data.

EU incorporation assay showing increased global transcription with HCE but not LCE. ($P < 0.01$ for HCE vs. solvent). But the violin plots demonstrate the noise in the assay. We also used automated counting to eliminate bias which is different than the cited publication which counted the foci manually.

As an aside, I totally agree that this is an interesting and unexpected observation. To be clear, this is a drug-induced increase in transcription that is dose dependent with more increase with 100 nM than in 20 nM (see new fig 2b). So it may or may not be related to the fundamental biology of EWS::FLI1. As this reviewer knows, the drug is not a perfect substitute for EWS::FLI1 silencing. There are differences. I pulled the paper suggested above and the authors in that paper don't actually show the impact of EWS::FLI1 silencing on EU incorporation only a comparison with IMR90 and the impact of

etoposide on EU incorporation so it does make one wonder if indeed silencing of EWS::FLI1 increases transcription as this reviewer suggests. It is known that there are more targets induced with silencing than suppressed which would in fact contradict these published observations. My interpretation of the results of our manuscript is that the drug favors GC and GA rich sequences. So the induction of transcription could be due to binding at GC sequences, known to be in the proximal promoter that favor HCE relative to LCE which favors binding at GA rich sequences. Consistent with this idea, we published that the combination of mithramycin and CDK9 inhibitors is synergistic and lowers the amount of drug needed to inhibit EWS::FLI1. But in the setting of this drug, my interpretation would be that this increase in transcription increases the stress and likely contributes to the activity and probably toxicity at higher concentrations (HCE) but this was not something we were thinking about. Thank you.

10. Wouldn't the increase in transcription be better described as "MMA-driven" rather than "EWS::FLI1-driven"? There appears to be a circular logic: the effects of MMA on transcription elongation are categorized into genes repressed or induced by EWS::FLI1 (Figure 6), yet role of EWS::FLI1 is defined by the gene's response to MMA. A more direct test of whether MMA acts solely through EWS::FLI1 would be to assess RNA Pol II activity following MMA treatment in cells lacking EWS::FLI1, either through knockout, knockdown, or a different cell line. Similarly, the logic of comparing EWS::FLI1 and MMA effects on medium and long genes has limitations. Small genes, often essential and housekeeping, would typically have little expression changes, whereas long genes simply contain more MMA binding sites. The hypothesis is plausible, but needs direct evidence showing MMA's effects in the absence of EWS::FLI1.

Yes, we agree, we think this is MMA driven and not EWS::FLI1 driven. We think it contributes to dose limiting toxicity as well as stated above. There are general effects of the drug that are totally independent of EWS::FLI1. In addition, we do not claim that these effects are solely due to effects on EWS::FLI1 and in fact the observations about the differences in activity based on gene size is novel in this disease context. Having said that, we agree this was not clear in the first submission. This is the major reason why we flipped the paper around to first talk about effects on transcription and then talk about the effects on EWS::FLI1 binding. This allows us to more clearly speak to effects on EWS::FLI1 that alter transcription initiation which are specific to EWS::FLI1 relative to elongation which are a generalized effect of the drug (and gene size as this reviewer points out). We also added additional profile plots first of the overall effect on elongation for ALL genes which is the new Fig 2b (which was Fig 5). In addition, we included profile plots of EWS::FLI1 target genes in Figs 3c, 3d, 3g. 3g in particular shows the intersect of EWS::FLI1 effects and elongation effects of the drug. We also clarified the text:

In contrast, the suppression of EWS::FLI1 with HCE resulted from a combination of effects on initiation and impaired transcriptional elongation (Fig. 3D). The generalized non-specific induction of transcription described above (Fig. 2a) was evident but.....

This comment was challenging but greatly improved the manuscript:

11. The violin plots for TES in Figure 6 and both TSS and TES in S9 are dominated by outliers so that it is impossible or nearly so to see what difference could explain the p-value. Perhaps

the full y-axis could be shown by an inset or supplement figure, then the bulk of the data shown with a more reader friendly scale.

We agree that the difference is hard to see. We tried a lot of different views to draw out the differences but it is just the nature of the plot. We include expanded sizes of the graph as supplemental S1 and S2. But to make it more clear, we added profile plots to the main manuscript, Figs. 3c, 3d. In addition, we focused only on the TES and point out that it really just the statistics that matter at the TES for repressed targets for HCE. There is not a significant difference with HCE while there is with LCE. The directionality of LCE is obvious in the profile plot 3c. So this is really the best we could do.

12. Figure 6f doesn't have a gene structure shown. By comparison to 4d, it appears likely that this view is also cropped too close to the start and stop of the gene with too little flanking regions to interpret the signals contained in the gene.

A similar comment was made by the other reviewer. Space and figure numbers (and supps) has been a challenge. We made new figures and added them. The signal to noise of this assay was excellent. We cropped them tight to make them bigger but that led to a loss of things like the gene structure. For this specific figure, we had to move it to supplementary figs to allow for the profile plots. We now include an expanded view of *PRKCB* which includes flanking regions and the intersect with the CUT&Tag and the gene structure. It is supplementary Fig. S3. The signal to noise of this assay was excellent.

Minor:

1. The term "S" in Figure 1 isn't defined until the legend for Figure 2. The term "M" isn't defined until the legend of Figure 4.

Thank you we corrected this.

2. Figure 2d – The differences in peaks are quite difficult to see, likely because the figure has been stretched too far in the x-direction, or widthwise. If the peaks were not so broad either by increasing the region of the genome viewed or by narrowing the panel's width, it could be much easier to spot the difference in relative height and the changes in Peak 3.

An expanded view of this figure and the *RCOR1* locus is now shown as Fig 4f. The differences in peak height are quite obvious in this group scaled image. The quantitation is retained as Supplementary Fig. 6.

3. The paragraph spanning pages 12 and 13 was likely meant or should be broken into 2 or more paragraphs, to make the logic of the data interpretation easier to follow.

Agreed. Thank you. Yes, we broke it up at E2Fs in this resubmission. It was a seriously long paragraph.

4. On page 13, the reference to "(Figs. 4e, 4f, and 6a-d)" likely meant to be "S6a-d".

Thank you. These have all changed in the revision. 4e, 4f are now Supplementary Figs S7a, S7b, and we did not point out the 6a-d was indeed S6a-d. But these are now deleted due to space constraints.

5. The somewhat excessive discussion in the results section can make the manuscript difficult to read. For example, the mention of “Cmax of the compound in patients” on page 17 is rather out of context in the middle of a description of GRO-seq results in cell culture. The impact of the research is diminished if one must look up too many references to get through the results section.

Agreed. It was a bit chatty. We edited it. There is still a lot of transitions.

6. Supplement Figures S2B and S2D are in Greek font.

Wow! This was quite surprising! I am not sure how this happened. Our most recent version of the figs still had it in standard script. So there must have been something in the document conversion process that changed it. Thank you for pointing this out.

7. I can't say I've ever seen a complaint about IACUC or any mention of it outside of a methods section in a paper before.

Thank you. To be clear, I am not complaining and I think this is appropriate treatment of animals to limit second surgeries. We are very passionate in our lab about limiting pain and distress to animals and even try to limit the number of experiments and the number of animals. In this article, we simply wanted to make the point, that the animals could tolerate more treatment but we couldn't administer it due to appropriate restrictions. However, patients could be retreated and likely the tumor regressions would likely, or at the very least, could be, sustained. But the point is that the limitation in the response is the model not the drug or toxicity. We are sorry if it comes across aggressively towards IACUC. We added the word appropriate to voice our support for IACUC. Thank you for pointing this out. I would not have thought of this interpretation and certainly don't want to minimize the importance of the IACUC and the ethical treatment of animals.

8. The methods discusses RNAseq and a siEWS::FLI1 treatment, which are not found in the paper.

Thank you. All the methods have been updated. We included references for the siRNA targeting EWS::FLI1. While we have published this experiment, it was a new experiment for this paper (actually for a different paper but we used it here and will have to reference it there). The reason we repeated this is we have a new 3-cycle suppression rule in the lab. That is a 3-cycle suppression of expression of the target in the actual RNA that is sequenced as measured by RT-qPCR is required for knockdown to be considered adequate.

Reviewer #5 (Remarks to the Author):

Specific comments

- The title of the paper really 'buries the lede' by focusing on the Pol2 processivity aspect (at high, clinically-intolerable doses) while ignoring the more translationally impactful finding that the low-dose/continuous exposure schedule was more effective than the bolus schedule, and thus suggests that earlier trials which failed due to hepatotoxicity toxicity during escalation

could, indeed should, justify revisitation of MMA in trials as a continuous, low dose infusion schedule analogous to the Kofman case study cited as an inspiration for this project. This is hinted in the last paragraph of the discussion, but is perhaps the most revelatory finding of the study.

I could not agree more! This is music to my ears!!!! This is the reason we did the study, and the reason why **we also** find it interesting. We did the study to better understand why our clinical trial failed and the Kofman one was a success. This is the second time we have done this. With trabectedin, we showed the opposite, threshold concentration matters and we have now translated this into bona fide clinical responses in Ewing sarcoma with a drug that previously failed in the disease (presented orally at ASCO and *manuscript in preparation*). The current study, ironically, shows the opposite that for mithramycin, continuous infusion matters. So the current study could be highly impactful as this reviewer suggests!!! But notably we discuss this in the last paragraph of the intro on page 14, in the first paragraph of the results section, in the results on page 24, and is explicitly discussed in the second paragraph of the discussion. We believe, as a translational lab (as this reviewer clearly does as well) that the coolest aspect of this study is that we link the actual activity of RNAPII and its processivity across the genome to clinically relevant conclusions.

- Ideally equivalent dosing would have been performed between experiments. Though slight, the difference in total drug exposure between high (1800 nM*hr) and low dose (1440nM*hr) should be discussed as a potential confounder to comparing the schedules and their effects on gene expression. The absence of data comparing toxicity of the two schedules on non-tumor cells/tissues is a short coming. It is not very convincing indicator of toxicity that animals regain some weight after removing the drug, as this weight gain could also correspond to tumor growth. While this could be due to the fact that mice do not recapitulate the toxicity observed in patients, there needs to be more thorough support for these claims and a discussion on this potential limitation. Evaluation of liver function markers or histopathology of liver would be useful to assess the 'non-toxicity' of the low concentration exposure claimed at lines 372-380 and shown in Supplemental figure 12. Given that dose limiting hepatotoxicity was a major limitation of MMA in the clinical trial lead by the senior investigator 2016, the mechanism of which was focus of another paper from the group in 2019, it seems that this key toxicity would at least be mentioned since it is not evaluated.

Again, totally agree. We tried very hard to match suppression and did a lot of experiments around this. If you look at 1F the suppression and induction is really close!! If we moved the exposures, the suppression became mismatched. So the obvious criticism if we would have done that and compared those exposures would be that the observed effects have nothing to do with schedule and instead has to do with simple differences in target suppression. So there was not a perfect solution and we chose to match suppression instead of exactly matching the exposure in large part because it made for a more challenging molecular question. But, we added a comment acknowledging this limitation as this reviewer suggested As far as toxicity goes, we have now included chemistries from the animals treated with the two schedules. This is how the toxicity would be evaluated in patients. We see limited toxicity and limited liver toxicity. But this reviewer is right that this could be due to the fact that mouse liver is not as sensitive as human. This comment has strengthened the manuscript

- It is presumed, but never demonstrated by comparison that the improved specificity at low dose MMA observed in vivo is upheld in the animal model.

Yes, this is true. We have previously shown this using immunofluorescence in two different papers (PMID: 21653923, PMID: 26979396). Therefore, in this manuscript, we wrote it as using the molecular mechanisms to justify the in vivo comparison of the schedules which we show. In 2025, I don't think the former immunofluorescence is an adequate assay.

- The phospho-H2Ax immunohistochemistry experiment is underwhelming on several points. First, this data is not terribly convincing, as it appears to be derived from one animal harvested at a relatively early time point the tissue examined appears to have come from one animal that was a non-responder in the LCE group, and harvested at a relatively early time point, which begs whether this response is reproducible across animals, and whether DNA damage may appear later in the course of treatment. It is also not surprising that MMA does not induce non-specific DNA double-strand breaks; this has been reported by others investigating lower doses of MMA (10nM Lin, 2021) in Ewing tumor cells, higher doses (25-50nM Scroggins 2018 MMA) in lung or bladder cancer cell lines, and has even been reported to mitigate neuronal cell death induced by DSB (Marakevich, 2020). Finally, while the authors exclude DNA damage as a factor contributing to the growth-control response observed for xenografts LCE MMA, an opportunity is missed to identify a mechanism of tumor growth control. This is unfortunate.

Thank you for the comment. This reviewer is clearly an expert in mithramycin. The premise of the paper is that reversal of EWS::FLI1 activity leads to growth control which we show throughout the paper. It does not appear that this reviewer is in the Ewing sarcoma field. But in the field inhibiting EWS::FLI1 is the 'holy grail' of therapeutic development. In the manuscript, we attempt to exclude more common mechanisms erroneously attributed to the drug, DNA damage, and non-specific global transcription inhibition. We clearly show these are less important in the context of Ewing sarcoma cells and an EWS::FLI1 oncogenic driver mutation. Nevertheless, to address the concern about the underwhelming phosphor-H2AX, we have more clearly highlighted which animal it was taken from. It was **not** from the non-responding animal as stated in this reviewer's comments. This is evident because in the previous version that tumor growth curve continues on past the arrow which would be impossible if the animal was sacrificed to collect that tumor. We agree, though, that in the previous version, it was an unfortunate placement of the arrow which did indeed suggest that we collected from the tumor from the least affected animal as this reviewer astutely noticed; the arrow was right over that curve. In this revision, we show that it is clearly at the inflection point of tumor regression, highlight what animal it came from in red, and show that all other animals regress tumors beyond this point. In addition, we did additional staining of tissue from a later time point on day 4 which I show below. There is not an increase in γ H2AX even as these tumor regress two days later. We are happy to include it if the reviewer wants but we already have 20 supplemental figures and I believe the more important staining is at the inflection point. We tried to stain for γ H2AX on tumors from animals collected on day 8 but there was no visible tumor cells in the section from the animal on that day consistent with the goal of therapy, eliminate the tumor. So the mechanism of tumor regression in Ewing sarcoma cells specifically at these exposures, is not due to non-specific DNA damage.

γH2AX staining of tumor tissue collected on day 4 (two days after the collection in Fig 7c) from animals treated with LCE and HCE.

- The experiments with MMA derivative AIT-102 seem appended to the paper, and don't really fit with the rest of the story. For example, are the doses of AIT-102 tested analogous/equivalent to either the HCE or LCE schedule for MMA? Are the mechanisms (affinity/sotchiometry, pharmacokinetics) of AIT-102 in terms of EWS:FLI1-GGAA interaction similar to MMA? It is not clear to me that this adds much to the paper as a whole, is it seems that the authors would need to study this agent in a similar manner either separately, or in parallel to MMA.

Again, totally agree. There is much to learn about AIT-102. The point of including it was simple and practical because it is very important for the clinical translation of AIT-102 to patients. The point is simply that similar in vivo activity can be demonstrated as a continuous infusion of AIT-102; a drug which is less toxic than MMA. So would MMA work in Ewings sarcoma if the trial is repeated as a continuous infusion? My belief, based on this publication is, yes! We will of course pursue this but there are practical challenges around drug supply and community enthusiasm which make the realization of this clinical trial in a rare disease like Ewing sarcoma less likely. But the less toxic analog, AIT-102, has traction. I think this simple in vivo data supports that if AIT-102 is introduced to patients, it should be administered as a continuous infusion. It is making its way through the NCI-NEXT program with this in mind. We are careful not to make claims about MOA for this drug but this is certainly our next direction looking at this and other analogs and the impact on EWS::FLI1 and other targets as well as the determinants of differential sensitivity. These studies will use many of the same techniques and now will incorporate some of the transcriptional stress suggestions of reviewer #4.

Minor comments:

-Line 156-7: A citation here regarding the SWI/SNF chromatin remodeling complexes as loading controls would be helpful.

Thank you we have added a reference from a world expert on SWI/SNF to verify that these subunits are chromatin bound.

-Line 159 – There seems to be an orphan start of a sentence orphan immediately following the paragraph heading.

Thank you we have fixed this.

-Line 160: States “To gain a deeper understanding of the impact of drug exposure...” but ignores the difference in total drug exposure being compared is about 20% lower in LCE schedule. This should be phrased more accurately to capture the study that follows Thank you as we stated above, we had to make a decision to pair suppression or exact exposure.

We chose expression because the paper describes a molecular mechanism. But it is notable that the lower exposure, LCE, is actually more effective at blocking EWS::FLI1 activity.

-Line 175-177: “as is commonly done in the field” provide references qualifying this

Thank you we have included references in this version.

-Line 186: “this is the clearest example of any compound inhibiting EWS::FLI1” this claim seems unsubstantiated as the authors are not looking at multiple compounds here. Please rephrase, or cite literature to support this statement.

We eliminated this statement. But I do think this intersection of CUT&Tag with GROseq is a true strength of the paper and the new gold standard assay for transcription factor targets.

-Line 227: “careful examination of this binding locus...microsatellite” provide reference or figure reference.

Thank you. This was in a previous publication from another group. We did not remember this when we did the analysis and discovered the microsatellite. But we reference the other pub and say (consistent with a prior publication).

-Line 264-265: Please provide a reference that this exceeds what is commonly observed for enhancer RNAs

We eliminated this sentence.

-Line 277: Please provide a reference for what is determined to be a supraphysiological concentration.

We show in our clinical trial the maximum tolerated concentration in patients is around 14-18 nmol/L. We have added this reference to this sentence. 800 nmol/L is clearly far in excess of these concentrations.

-For Figure 4 b/c and 4e, it would emphasize the difference in occupancy LCE vs HCE systems if the two plots were presented on the same scale for normalized fragment count. There is almost a >10x difference between LCE and HCE.

These were two different experiments and so the magnitude is different between LCE and HCE. To more directly compare, we did a motif analysis as suggested by a reviewer above and you can see the top differentially bound, highly significant motif is the GGAA microsatellite (P = 1e-1754) which we added as Supplementary Fig. 9c.

Rank	Motif	P-value	log P-pvalue	% of Targets	% of Background	STD(Bg STD)
1		1e-1754	-4.040e+03	12.17%	0.63%	59.4bp (91.5bp)
2		1e-1244	-2.865e+03	37.71%	13.67%	49.8bp (68.2bp)
3		1e-866	-1.995e+03	12.32%	2.03%	49.6bp (60.6bp)
4		1e-125	-2.882e+02	21.26%	14.27%	55.3bp (65.3bp)
5		1e-110	-2.539e+02	5.02%	2.05%	57.0bp (74.3bp)
6		1e-94	-2.170e+02	7.09%	3.65%	58.3bp (83.0bp)
7		1e-87	-2.018e+02	30.19%	23.32%	55.7bp (62.7bp)
8		1e-84	-1.944e+02	1.69%	0.39%	49.3bp (59.2bp)
9		1e-83	-1.922e+02	24.50%	18.32%	55.8bp (65.8bp)
10		1e-76	-1.754e+02	17.42%	12.34%	54.2bp (62.0bp)

Homer motif analysis of differentially bound sequences LCE vs. HCE.

The highest represented motif is the GGAA microsatellite consistent with it being favored by LCE.

-Much of the text in Supplemental figures 2B, 2D are in a symbol font and un-intelligible

Thank you. We are not sure how this happened. Even our table in the submitted files folder is normal. In this version, it is in normal script. We will verify it, after we submit it here. Thank you.

Reviewer #6 (Remarks to the Author):

I co-reviewed this manuscript with one of the reviewers who provided the listed reports. This is part of the NatureCommunications initiative to facilitate training in peer review and to

provide appropriate recognition for Early Career Researchers who co-review manuscripts.

Oct 16, 2025

Dear Reviewer,

Thank you for considering this article. The suggestions overall have been very helpful and have greatly improved the manuscript. In this revision, we address the comments of reviewer 2. The other 5 reviewers were satisfied with the last version. To address reviewer 2's comments required us to move some of the figures around. For example, while the impact of MMA on DNA binding of RUNX2-EWS::FLI1 has been studied by X-ray crystallography, the motif is present in only 2% of peaks. Therefore, the genome wide analysis is not that informative and so this was moved to the supplementals. In contrast, AP-1-EWS::FLI1 motif is found in up to 10% of the differentially bound motifs which made the analysis of the effects of the drug on heterotypic complex binding more evident. So these were moved to the main body of the text. All of the comments are collected and annotated below. In this cover letter, I highlight the reviewer critiques in **bold** while our response is in normal script. Text from the manuscript is in *italics*. We provide both a tracked version and a clean version to make the changes easier to check for this reviewer and the editor (although we did not track the changes in the legends).

Reviewer #1 (Remarks to the Author):

The authors conducted the requested analyses when possible. I do note that the authors have misinterpreted my comment regarding FLI1 and EWSR1 dependency scores in the DepMap database, there are several lines with EWSR1 and FLI1 dependency scores that are relatively modest including TC32. This suggests that TC32 is less dependent on the fusion than other lines, and perhaps mithramycin is disrupting other targets in this context that contribute to anti-neoplastic effects. However, none of the additional analyses change the initial conclusions of the paper and have no further comments or recommendations.

Thank you! Thank you also for this clarification. This is very interesting and certainly worthy of further study!!!

Reviewer #2 (Remarks to the Author):

The authors have improved their manuscript by adding technical details that were cruelly lacking in the initial version and made valuable effort to include genome-wide analyses. Yet, many messages of the manuscript are still unclear:

Overall, I have to say, this is an impressive, detail-oriented review. There are some labeling errors that this reviewer caught and two instances in the text where it says control to treated instead of treated to control. In addition, I do agree that the clearest data should be presented so I moved some figures to the supplementals and vice versa. I disagree that the data is subtle. For some of the associations, such as RUNX2, there are a limited number of peaks (approx. 2% in the motif analysis) associated with EWS::FLI1. However, this data still needs to be reported because the x-ray crystallography data published by another group shows that the drug

disrupts binding of this specific heterotypic complex via a direct interaction with the binding partner. Indeed we do see disruption of binding at the only two well-established RUNX2-EWS::FLI1 sites but this clearly is not going to translate into an impressive genome wide association or heat map given the limited number of binding events. However, it speaks to differential effects at heterotypic complexes vs. GGAA microsatellites and the importance of this association to the biology of the disease.

1- The effects of treatments on the Ewing signature are extremely subtle (Fig3b, FigS1) and the technical details that are provided are unclear and lack of rigor (example for FigS1 the legend indicate Log2FC while the figure reports log10(cpm + 1) (same in Fig3b) !! Are these ratios or absolute count values? How were these GROseq data normalized to enable these comparisons? this is essential to interpret such subtle (and sometime opposite) differences between LCE and HCE.

There were a limited number of errors in the last version. We have corrected the labeling errors and appreciate this reviewer's highly focused review of the manuscript. The GROseq normalization is standard CPM or counts per million reads normalization thus the violin plots compare normalized counts. In this comment, the reviewer highlights a simple labeling issue in the legend of figure 3b, S1. These are log10(cpm + 1) as in the figure. We have made those corrections. In terms of 3b, the effect is highly significant ($1e-41$) but visualized as a log-transformed violin plot which often does not appear impressive in genomic studies (in contrast to the statistic); particularly because it is log transformed. To address this concern, I would like to point out that the result is supported by multiple other pieces of data. Fig 3a, in particular, shows that the vast majority of targets with LCE. In addition, the heat map clearly shows uniform suppression or induction of targets with LCE while HCE is unable to induce repressed targets despite increased initiation with HCE. Figure 3c, 3d, 3f, 3g all make the same point collectively. Again, figures 3c and 3g do not appear subtle and show that high drug exposures impact RNA polymerase processivity and the ability to complete productive transcription. Collectively, these effects show, using different analyses and representation of the data, the same effects thus even the $1e-41$ does not stand alone. Additionally, it follows that higher concentration of the drug would have a greater effect on RNAPII processivity and this validates in vitro run on data that is over 30 years old.

2- At different levels of the manuscript, it is indicated that solvent (or control) to treated ratios is reported. If this is the case the interpretation of data would be completely different as they would show that mithramycin is indeed a transcription inhibitor! Such inconsistencies along the manuscript questions the quality of the analyses that are shown.

Thank you for this comment. Yes, in the text, at two separate spots, it did say control to treated which clearly is wrong. Thank you for pointing this out. I have read this manuscript many times and missed that. We double checked all other instances throughout the text and in the legends and it is correct everywhere else. As to the concern of the quality of the analyses, these were done using standard tools which we have clarified in the methods in this version. The only explanation is the sheer number of changes in the last version. We hope this satisfies any concerns.

3- The genome-wide analyses of Cut&Tag appear biased, focusing only on EF1 (FLI) peaks linked to EF1-induced or -repressed genes based on RNA-seq. But peak to gene annotation is inherently challenging, as different peak types (e.g., GGAA microsatellites or isolated peaks) may map to the same gene, complicating interpretation. A clearer approach would be to first present all EF1 peaks, identify differential peaks and their characteristics (e.g. motifs) and only then integrate transcriptomic data. This would better show how treatment-induced binding changes relate to the EF1 transcriptomic signature. Heatmaps and annotation profiles also need simplification and clearer legends. For example, the "gene" label is misleading when the central sites represent EF1 bound GGAA microsatellites annotated to EF1 induced or repressed genes.

Thank you for the comment. Yes, we agree that we were not totally clear in either our labeling or methods section. We clarified the labeling as suggested. In addition, we reorganized the methods to increase clarity. We also agree that peak annotation is a challenge but used Homer to annotate it in the most unbiased way possible. In addition, we did do the analysis exactly as the reviewer described above in the previous version. Unfortunately, the methods were not clear and the motif analysis was even listed in a different section. We first presented all EWS::FLI1 peaks that were differentially bound and did a motif analysis to characterize the motifs associated with the peaks. Then we used homer to annotate peak to gene to limit bias and subsetted on induced or suppressed targets. This left reasonably high numbers of motifs to visualize EWS::FLI1 binding. We added the number of motifs in each analysis to the figure legends. Additionally, we reorganized the methods section. We added the motif analysis to the cut and tag section so it immediately follows the methods for peak calling, QC etc. and changed the section heading to **CUT&Tag bioinformatics and motif analysis**. We added italic headers to delineate the steps in the differential binding analysis and we redid all the figure legends so that they all are written the same way. We eliminated the distracting “genes” label and replaced it with labels for GGAA microsats, isolated GGAA, tandem GGAA-RUNX2, GGAA-AP1 or, GGAA-E2Fs.

4- The methodological choices are unclear. Although the authors indicate that a DiffBind analysis was performed, it is poorly integrated into the results. Why not use DiffBind peak statistics to illustrate variation in binding at key loci (e.g., Fig. 4d, 6d) instead of fragment counts across conditions? Does this mean the peaks of interest were not significant according to DiffBind? There should be coherence between materials/methods and results sections.

The approach we employed (post feedback in the first revision) was to do the analysis this reviewer suggested genome wide and additionally show examples of well-established loci. But we felt like we couldn't just show the loci of well-established targets without quantitating the fragment counts which we recognize is not always done. The loci selected were chosen not based on the DiffBind but on the confidence that this is an actual target of an EWS::FLI1 containing cofactor complex. This is challenging because there are very few established binding sites of these cofactors in the literature. Therefore, we presented **all of them**. Admittedly, the effect at NR0B1 is subtle but this is the most well-established **binding loci linked to expression**. It is known that NR0B1, while highly specific for EWS::FLI1 with a clear loci where binding is linked to expression, has a small magnitude change in expression with EWS::FLI1 silencing relative to longer range enhancers. However, longer range enhancers are harder to link to targets and there are none that are tightly linked to expression in a manner analogous to NR0B1. We tried to present many well-established targets as a complement to the diff bind analysis which we also did. Additionally, we did present the volcano plot as a statistical analysis of some of the most well-established targets in the literature. The volcano plot clearly shows the relationship between binding and statistical significance (see fig 4b). So three different approaches were used to establish differences in binding of a given cofactor and two different approaches to show statistical significance at high confidence sites.

Nevertheless, as suggested by this reviewer, we did pull all of the loci that we highlight in DiffBind. All of the loci show highly significant changes in binding. For simple sharp well-defined peaks like SPP1, the peaks line up perfectly to the coordinates in IGV and the P-value is $1e-29$. However, for more complex peaks like UPP1, there are 10 different fragments in the area, 4 of which are highly significant. DiffBind employs algorithms to account for complex binding patterns, but these make it difficult to link the DiffBind peaks back to IGV. Therefore, I would prefer not to present what ends up being complicated data. To be clear every one of the sites that we described as being significantly different, when viewed in the DiffBind export, yielded statistically significant differences in binding in the region. Sometimes, however, the fragments in the complex binding patterns that are significant did not line up perfectly with the loci in IGV.

5- The data on ETV6, E2F, RUNX2 and AP1 are very preliminary and poorly convincing. They are primarily based on motif analyses. Co-binding of EF1 and co-factors have not been validated using

cofactor ChIPseq or Cut&Tag which could be obtained from public datasets. Furthermore, it is very unclear what these data bring to the manuscript?

Throughout the manuscript, we acknowledge the limitation in this data. However, we need to include it because it has been suggested by another group that the drug has different effects at heterotypic EWS::FLI1 complexes than at GGAA microsatellites. This has been supported by fluorescent anisotropy data of a GGAA repeat and an x-ray crystal structure that shows disruption in binding of a FLI1-RUNX2 complex via a specific interaction between the side chain of MMA and RUNX2 but not EWS::FLI1. A strength of our manuscript that is really quite amazing is that we can show genome-wide data that supports both models even though the thought in the field has been that one has to be wrong. Given the low frequency of RUNX2 interactions in the genome, the data is not perfect as there are a limited number of sites. So, we support the observations by looking at other heterotypic complexes. The idea of using published data sets proposed by this reviewer is a good one, but we are interested in only those that intersect with EWS::FLI1 which would be impossible to deduce from these data sets (since EWS::FLI1 is only expressed in Ewing sarcoma cells). Additionally, the sites may or may not be the same as in other contexts because EWS::FLI1 has neomorphic properties. My personal opinion, supported by the new presentation of these data, is that RUNX2 is not that important of a cofactor in this cell line and probably not in the tumor. So to make the data stronger and address comment 7, we focused more on the AP-1 site in this revision. AP-1 consistently appears in the literature as an EWS::FLI1 cofactor(s), and was the most highly represented co-factor in our motif analysis. The data we present clearly shows that LCE disrupts binding at GGAA-AP-1 sites, particularly when subsetted only on differentially bound peaks using DiffBind and LCE.

6- One of the messages of the manuscript is that mithramycin treatments increase EWS-FLI1 protein expression and, consequently, EWS-FLI1 binding to its DNA sites. This is an interesting, convincing, though unexplained, observation. The message on the consequences on the expression of EWS-FLI1 targets is much less clear.

I am not sure what the reviewer is asking here. There is not a specific figure that they highlight in this comment. This is essentially the point of the manuscript so we are hopeful that with the additional clarifications particularly in the methods, and figure legends, that we satisfy this reviewer. In addition, it is worth pointing out that we address the increase in the text:

“.....(the increased binding) likely reflected the combination of increased expression (see Fig. 1c) and stabilization of EWS::FLI1 binding at the GGAA response element. Stabilization of binding by MMA has previously been demonstrated in vitro for the FLI1 DNA binding domain at a GGAA hexameric repeat oligonucleotide using fluorescent anisotropy^{26, 29}.”

We do not get into the mechanism of increased expression as this is beyond the scope of the manuscript and quite complicated.

7- The preclinical data are nice. The authors should simplify the rest of the manuscript, only showing robust data with clear indication of the methods that were used.

Thank you for this comment. On reflection to this comment and comment 5, we agree that the most robust data should be presented in the main body of the manuscript and the less robust data as supplementals. In the original version, we focused on the RUNX2 interaction because that is the interaction supported by X-ray crystallography data. However, our own motif data said that the RUNX2 motif is under-represented in EWS::FLI1 peaks and found only in 2% of the called peaks. Therefore, the genome-wide analysis is never going to be informative of this interaction other than to say it is not widely represented in the genome. Therefore, we focused the main body of the text in this version on AP-1. This is well represented in the motif analysis and widely recognized in the literature. The data, we believe, is convincing and therefore we moved it

to the main body and included both the genome-wide analysis and the DiffBind to clearly demonstrate a loss of binding at this heterotypic complex. We retained the RUNX2 and E2F in the supplementals because the low frequency is, in fact, important and interesting to a Ewing focused person. But to make the main point of the manuscript, as this reviewer suggested, we moved AP-1 forward. Thank you for this comment.

Reviewer #3 (Remarks to the Author):

I appreciate the authors transparency and thoughtful explanations. I can understand the dilemma concerning the TC252 xenografts and am satisfied with leaving it as a for reviewer only figure. The authors have satisfactorily addressed my concerns.

Thank you for this and the previous constructive comments.

Reviewer #4 (Remarks to the Author):

The revised manuscript by Kaufman et al. the authors have made a thorough response to points raised by the 4 prior reviewers. The manuscript in its current form is a remarkable improvement.

The authors have responded sufficiently to each of my comments. Specifically, for comment 7 the information about the GGAA repeats described is sufficient and the extra figure isn't necessary. To point #9, I appreciate the authors efforts in performing the EU incorporation assay and agree with their decision about using the data at this time and changes to the text.

In general, I've found this manuscript both more readable and the data analysis easier to follow. I recommend the study for acceptance.

Thank you very much for the positive feedback and the constructive comments.

Reviewer #5 (Remarks to the Author):

Thank you for your thoughtful and detailed response to our critique. As I am reviewing this manuscript with a trainee, we appreciate that your responses have been both deliberate and extensive as well as humble, without being defensive or combative, as some authors' rebuttals can be. This review has provided an excellent example of professionalism in responding to peer review, and we are grateful for this experience - Thank you!

We are generally satisfied with the response to our initial critique, and appreciate clarification of a few key points. This is particularly regarding clarification and corrections regarding the pH2AX staining in figure 7; we agree with the author that additional images are not necessary.

Thank you! Yes, responding to the sheer number of comments was challenging but this is definitely a better manuscript now and so a credit to the peer review process at this journal overall.

Reviewer #6 (Remarks to the Author):

I co-reviewed this manuscript with one of the reviewers who provided the listed reports. This is part of the Nature Communications initiative to facilitate training in peer review and to provide appropriate recognition for Early Career Researchers who co-review manuscripts.